# Power-Boosted Granger-Causal Discovery for Large Heterogeneous Panel Data

**Yiheng Gu** [1]  **Xiufan Yu** [1]

## Abstract

This paper proposes a power-enhanced panel Granger causality test (PE-PGCT) for assessing the Granger non-causality in heterogeneous and potentially high-dimensional panel data. Building on any existing panel Granger non-causality test, we show, both theoretically and empirically, that the proposed PE-PGCT boosts its power substantially. The power gains are particularly significant in situations of high-dimensional panels when the cross-sectional dimension exceeds the time dimension, as well as under sparse alternatives when the signals are sparsely distributed across panel units. We establish rigorous theoretical guarantees on the asymptotic behavior of the proposed power enhancement component, demonstrating attractive power enhancement properties that it induces negligible size distortion under the null hypothesis while delivering significant power gain under the alternatives. The empirical performances are illustrated via extensive simulation studies, as well as a real-world application.

## 1. Introduction

Granger causality, originally introduced by Granger (1969), has become a foundational tool for uncovering dynamic relationships and directional predictability in time series. It defines causality in a predictive sense (Granger, 1980; 2003; 2004): one series is said to Granger-cause another if its past values improve the forecast of the dependent variable beyond the information contained in the target's own past. With the growth of large-scale longitudinal data, panel Granger causality test (PGCT), which extends Granger causality to panel settings by exploiting both cross-sectional and temporal dimensions, has attracted increasing attention.

[1]Department of Applied and Computational Mathematics and Statistics, University of Notre Dame, Notre Dame, IN 46556, United States. Correspondence to: Xiufan Yu <xiufan.yu@nd.edu>.

*Proceedings of the 43rd International Conference on Machine Learning*, Seoul, South Korea. PMLR 306, 2026. Copyright 2026 by the author(s).

Early developments of PGCT are closely linked to the literature on dynamic panel models and panel vector autoregressions. The seminal work of Holtz-Eakin et al. (1988) assumes homogeneous dynamics across cross-sectional units, and examines Granger causality by pooling data to estimate a common autoregressive structure. Recognizing such homogeneity assumption may be too restrictive in practical applications, subsequent research (Kónya, 2006; Emirmahmutoglu & Kose, 2011) has evolved toward more flexible approaches that allow coefficients to vary across units, thereby accounting for cross-sectional heterogeneity.

One major methodological breakthrough in PGCT is the Dumitrescu-Hurlin (DH) test (Dumitrescu & Hurlin, 2012), which has become one of the most widely used methods among practitioners. The DH test estimates unit-specific autoregressive models, computes individual Wald statistics for the null hypothesis of no causality, and then averages these statistics across the panel. Under suitable regularity conditions, the standardized average converges in distribution to a normal limit under a sequential asymptotic regime when the time dimension $T \to \infty$ first, followed by the cross-sectional dimension $N \to \infty$. This asymptotic property makes the DH appealing for panels with large $T$, while limiting its ability to control Type I error in panels with small or moderate $T$, especially when $T < N$.

To address this limitation, Juodis et al. (2021) propose the Half-Panel Jackknife (HPJ) test, which challenges the necessity of averaging unit-level statistics. Their key insight is that under the null hypothesis of non-causality, the Granger-causal coefficients are identically zero for all units, rendering the parameter of interest homogeneous under the null, even when other model parameters remain heterogeneous. By exploiting this structure, HPJ employs a pooled least-squares estimator and a quadratic-form statistic, achieving a faster convergence rate and improving finite-sample performance in panels with large $N$. Several subsequent studies have further facilitated practical implementation (Lopez & Weber, 2017; Xiao et al., 2023) and extended the PGCT methodology in various directions (Minorics et al., 2022; Nazlioglu & Karul, 2024).

Despite their popularity, existing PGCT methods face intrinsic limitations. The DH test aggregates evidence by averaging individual Wald statistics across panel units, while

the HPJ test relies on a quadratic-form aggregation of the estimated Granger-causal coefficients. Such aggregation schemes are well-suited to situations with dense signals, but may substantially dilute strong signals when only a small subset of panel units exhibits nonzero Granger-causal effects, resulting in severe power loss under such alternatives. Consistent with the terminology in hypothesis testing literature, we refer to this type of alternative as the "**sparse alternatives**" (Fan et al., 2015; Yu et al., 2024a;b), when the number of signal-bearing coordinates is small. In modern applications involving large panels with heterogeneous dynamics, e.g., firm-level financial data or user-level behavior logs, sparse alternatives are prevalent. Hence, this limitation becomes increasingly problematic, motivating the need for methods that boost power under sparse alternatives.

In this work, we propose a new PE-PGCT, that is specifically designed for large-scale heterogeneous panels and delivers substantial power gains, particularly under sparse alternatives. Recognizing the fundamental limitation that informative signals are often obscured by cross-sectional aggregation through simple averaging or quadratic-form aggregation, our approach boosts the detection power by selectively amplifying informative marginal signals. We construct a data-driven screening component, named "PE component", and integrate it with an existing PGCT. As a result, the resulting PE-PGCT adaptively accumulates evidence from individual units exhibiting potential Granger-causal effects for power boosting.

The proposed PE-PGCT enjoys several appealing theoretical and practical properties. We establish rigorous guarantees on its validity, showing that it achieves substantial power gains under sparse alternatives while maintaining correct Type I error control under the null. Moreover, PE-PGCT is provably at least as powerful as the baseline PGCT, ensuring no loss of power in any circumstance. Moreover, our framework is flexible and can be combined with a broad family of baseline PGCTs. On a separate note, existing PGCTs, such as DH and HPJ, exhibit markedly different performance across panel regimes; see an empirical illustration in Figure 1. These differences are rooted in the distinct asymptotic regimes under which the two tests are valid. The proposed PE-PGCT can be built on any existing PGCT and consistently boosts its power, since the PE strategy itself remains valid regardless of the choice of baseline PGCT. Additionally, we provide practical guidelines for selecting a data-adaptive baseline PGCT, which is shown to deliver appealing performance across a wide range of panel regimes.

## 2. Methods

### 2.1. Model Setup

Consider a panel with $N$ units observed over $T$ time periods. Let $\{x_{i,t}\}$ and $\{y_{i,t}\}$ denote the covariate and response of

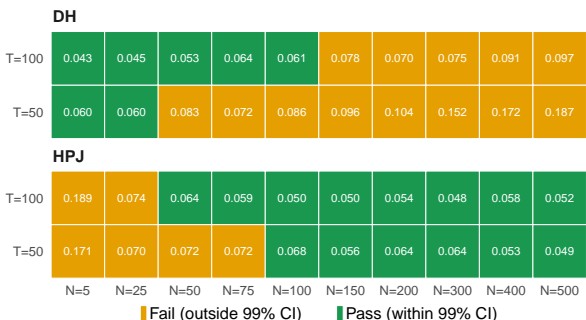

*Figure 1.* Failure map of Type I error control for DH and HPJ (inside vs. outside the 99% confidence intervals) based on 1,000 simulations. The empirical results shows DH is reliable for large $T$ and small $N$ but over-rejects when $N$ is large, whereas HPJ is reliable for large $N$ and small $T$ but over-rejects when $N$ is small.

individual $i$ at time point $t$, $i = 1, \ldots, N$, $t = 1, \ldots, T$. Here, we allow the cross-sectional dimension $N$ to be very large, even larger than the time dimension $T$. Consider a linear dynamic panel model

$$y_{i,t} = \alpha_i + \sum_{j=1}^{K} \gamma_{ij} y_{i,t-j} + \sum_{j=1}^{K} \beta_{ij} x_{i,t-j} + \epsilon_{i,t}, \quad (1)$$

where $\alpha_i$ is a panel-specific heterogeneous effect that is time-invariant but allowed to take different values across panel units, $\gamma_{ij}$ and $\beta_{ij}$ are heterogeneous coefficients that are also allowed to differ across panel units, $\epsilon_{i,t}$ is the idiosyncratic errors that are assumed to be independent with mean 0 and variance $\sigma_i^2$. This model can accommodate cross-sectional heteroscedasticity by allowing $\sigma_i^2$ to take different values for $i = 1, \ldots, N$.

$K$ is the lag order. Throughout this work, we assume $K$ is small and finite. Though it is not necessary that the lag orders for $\{x_{i,t}\}$ and $\{y_{i,t}\}$ are identical, we follow common practice by setting them to take equal values (Holtz-Eakin et al., 1988). Here, we assume $K$ is known. The selection of the lag length $K$ has been well studied in the literature, e.g., Lütkepohl (1985); Ivanov & Kilian (2005); Bruns & Stern (2019); Shojaie & Fox (2022).

Under the formulation of (1), the problem of testing Granger non-causality of $\{x_{i,t}\}$ on $\{y_{i,t}\}$ becomes testing

$$H_0 : \boldsymbol{\beta}_i = (\beta_{i1}, \ldots, \beta_{iK})^\top = \mathbf{0} \text{ for all } i = 1, \ldots, N$$
$$\text{vs } H_a : \text{at least one } \boldsymbol{\beta}_i \text{ is not zero} \quad (2)$$

**Notation.** Before proceeding, we first introduce some necessary notation. We use $c$ or $C$ to denote generic positive constants, which may vary from line to line. For any integer $n$, we use $[n]$ to denote the set $\{1, \ldots, n\}$. For two sequences $\{a_n\}$ and $\{b_n\}$, we use $a_n \lesssim b_n$ to denote $a_n \leq Cb_n$ for some universal constant $C > 0$ that does not depend on $n$, and we use $a_n \asymp b_n$ to denote $a_n \lesssim b_n$ and $b_n \lesssim a_n$.

For any random variable $U \in \mathbb{R}$, the sub-Gaussian norm is defined as $\|U\|_{\psi_2} = \inf \left\{ c > 0 : \mathbb{E}[\exp(U^2/c^2)] \leq 2 \right\}$. For any random vector $\mathbf{V} \in \mathbb{R}^d$, the sub-Gaussian norm is defined as $\|\mathbf{V}\|_{\psi_2} = \sup_{\boldsymbol{\nu} \in \mathcal{S}^{d-1}} \|\boldsymbol{\nu}^\top \mathbf{V}\|_{\psi_2}$, where $\mathcal{S}^{d-1} = \{\boldsymbol{\nu} \in \mathbb{R}^d : \|\boldsymbol{\nu}\|_2 = 1\}$ is the unit sphere in $\mathbb{R}^d$.

## 2.2. Power Enhancement (PE)

Let $T_{pivotal}$ denote any existing pivotal panel Granger-causality test that attains correct asymptotic size. In what follows, we propose PE-PGCT, a power-enhanced panel Granger-causality test, which substantially boosts the statistical power relative to $T_{pivotal}$, while incurring asymptotically negligible impact on Type I error control.

To achieve this, we propose a power enhancement component $J_{PE}$, tailored to the panel Granger-causality setting. This PE component is constructed to take the value zero under $H_0$ to maintain proper Type I error control, but to diverge rapidly under $H_a$ to amplify power. Though we adopt the concept of "power enhancement" to relate to existing literature, the proposed technique introduces new methodological aspects beyond prior works. Specifically, directly adapting the power enhancement method of Fan et al. (2015) to the panel Granger causality test presents several challenges, while our approach overcomes those limitations. A detailed comparison is provided in Appendix C.3 to more explicitly demonstrate our contribution and innovation.

We begin by extracting marginal signal information from each panel unit. We resort to the individual Wald statistic, denoted by $W_i$, for testing $H_{0i} : \boldsymbol{\beta}_i = \mathbf{0}$, respectively, for each $i \in [N]$. The popular Dumitrescu-Hurlin (DH) test (Dumitrescu & Hurlin, 2012) aggregates these statistics by averaging, which provides a natural and stable way to pool information across multiple panel units, whereas such averaging may dilute strong but localized signals, thereby weakening the test power when Granger causality is present in only a subset of units. To more effectively detect the presence of such signals, we focus on the extreme behavior of the individual statistics and consider their maximum:

$$M_{Wald} = \max_{1 \leq i \leq N} W_i. \quad (3)$$

Equipped with $M_{Wald}$, we define the PE component as

$$J_{PE} = \sqrt{N} \cdot M_{Wald} \cdot I\{M_{Wald} > \delta_{N,T}\} \quad (4)$$

where $\delta_{N,T}$ is a carefully-chosen threshold to separate estimation noise from genuine signals. It plays a critical role in balancing Type-I error control and power enhancement. Intuitively, $\delta_{N,T}$ should slightly exceed the stochastic order of $M_{Wald}$ under $H_0$. The indicator function in (4) further ensures that $J_{PE}$ is activated only when $M_{Wald}$ exceeds the noise level implied by the null distribution.

*Remark* 2.1 (Strength of $J_{PE}$ under sparse alternatives). $J_{PE}$ is particularly effective under sparse alternatives, where Granger causality is present in only a small subset of panel units. Intuitively, $M_{Wald}$ acts as a signal detector. By construction, $M_{Wald}$ is highly sensitive to strong marginal signals even when signals are rare across panels. Under sparse alternatives, $J_{PE}$ diverges at the rate $\sqrt{N}$ as long as one marginal signal is captured by $M_{Wald}$; see Theorem 3.8 for formal characterization of high-power regions of $J_{PE}$. This makes $J_{PE}$ an effective complement to existing panel Granger-causality tests, which primarily rely on averaging strategy and favor dense alternatives.

Although under some regularity conditions, each individual $W_i$ converges in distribution to a $\chi^2_K$ distribution as $T \to \infty$ under the respective null hypothesis $H_{0i}$, the asymptotic behavior of their maximum is considerably more delicate, especially when $N$ diverges with $T$ and even more so in modern settings when $N$ may exceed $T$. As we show in Proposition 3.5, characterizing the asymptotic distribution of $\max_{1 \leq i \leq N} W_i$ and its deviation to $\max_{1 \leq i \leq N} \chi^2_{K,i}$ becomes technically challenging in such regimes.

After carefully examining the asymptotic distribution of $M_{Wald}$, we define the threshold

$$\delta_{N,T} = (K + 2\sqrt{K \log N} + 2 \log N) \cdot C(T), \quad (5)$$

where $C(T)$ is a slowly diverging correction factor given by

$$C(T) = \max\{1.5, \log(\log(T))\}, \quad (6)$$

which accounts for finite-sample deviations and ensures uniform control of the null behavior as $N, T \to \infty$. The constant 1.5 in $C(T)$ is a practical calibration choice to safeguard the finite-sample performance, aiming to provide a simple, stable, and broadly applicable implementation rule. Theoretically, any choice of $C(T) > 1$ such that $C(T) \to \infty$ as $T \to \infty$ (albeit arbitrarily slowly) would suffice for establishing the asymptotic theoretical guarantees. However, in finite samples with small or moderate $T$, $\log \log(T)$ takes small values, potentially leading to under-thresholding and hence distorted Type I error. The lower bound of 1.5 is therefore imposed to ensure a minimum level of thresholding that stabilizes the test performance in small and moderate samples.

In Theorem 3.8, we prove that this construction guarantees that the PE component $J_{PE}$ is asymptotically negligible under $H_0$, but diverges under alternatives whenever at least one panel unit exhibits strong Granger causality.

It is important to note that the PE component $J_{PE}$ itself does not work as a test statistic. Rather, its role is to boost the testing power over an existing test after being added to a pivotal test $T_{pivotal}$. The asymptotic null distribution of the PE-PGCT is governed by $T_{pivotal}$, not by $J_{PE}$. We define

the test statistic for the proposed Power-Enhanced Panel Granger Causality Test (PE-PGCT) as

$$T_{PE-PGCT} = T_{pivotal} + J_{PE}. \tag{7}$$

Given any significance level $\alpha$, let $q^{\alpha}_{pivotal}$ denote the critical value of the test $T_{pivotal}$, that is, $P(T_{pivotal} \geq q^{\alpha}_{pivotal} \mid H_0) \leq \alpha$. The theoretical analysis in Section 3 proves that $\mathbb{P}(J_{PE} = 0 \mid H_0) \to 1$. As a result, we note that the addition of $J_{PE}$ does not affect the asymptotic null distribution, and the limiting null distribution of $T_{PE-PGCT}$ coincides with that of $T_{pivotal}$. Consequently, PE-PGCT rejects $H_0$ if

$$T_{PE-PGCT} \geq q^{\alpha}_{pivotal}. \tag{8}$$

To provide a ready-to-use tool for practitioners, we offer recommendations for the choice of the pivotal test below. This recommendation is based on a careful assessment of the relative strengths and limitations of existing approaches in the literature. As shown in Figure 1, DH performs well when $N < T - K$, but exhibits severely inflated Type I error when $N \geq T - K$, whereas HPJ shows the opposite pattern. Remark 2.2 is therefore designed to ensure that the baseline remains a valid pivotal test in each regime.

*Remark* 2.2 (The choice of the pivotal test $T_{pivotal}$). Methodologically, $T_{pivotal}$ can be instantiated using any existing pivotal panel Granger-causality test with correct asymptotic size. After carefully weighing the relative advantages and limitations of available approaches in the literature, we adopt a data-adaptive strategy and recommend different pivotal tests depending on the dimensionality of the panel. Specifically, we propose using

$$T_{pivotal} = \begin{cases} |T_{DH}|, & \text{if } N < T - K \\ T_{HPJ}, & \text{if } N \geq T - K \end{cases} \tag{9}$$

where the first regime corresponds to the traditional panel setting and the second corresponds to the high-dimensional panel regime. Mathematical details of $T_{pivotal}$ are in Appendix B. This data-adaptive choice leverages existing methods in the literature within their respective regimes of strength, thereby ensuring robustness and validity of the proposed procedure across a broad range of panel settings.

## 3. Theoretical Properties

In this section, we explore the theoretical properties of the proposed PE-PGCT. Proofs of all propositions and theorems are presented in Appendix A.

**Assumption 3.1.** For $i \in [N]$ and $t \in [T]$, let $\mathcal{F}_{i,t-1} := \sigma(\{(y_{i,s}, x_{i,s})_{s \leq t-1}\})$ denote the $\sigma$-algebra generated by the past history $(y_{i,s}, x_{i,s})$ up to time $t - 1$. For each unit $i \in [N]$, the individual innovations $\{\epsilon_{i,t}\}$ are independent, sub-Gaussian with $\|\epsilon_{i,t}\|_{\psi_2} \leq K_\epsilon < \infty$ and satisfies:

$\epsilon_{i,t} \perp\!\!\!\perp \mathcal{F}_{i,t-1}$, $\mathbb{E}(\epsilon_{i,t}) = 0$, $\mathbb{E}(\epsilon^2_{i,t}) = \sigma^2_i$ with $c < \sigma^2_i < \infty$ for some $c > 0$, and $\mathbb{E}(\epsilon^4_{i,t}) < \infty$.

**Assumption 3.2.** For each $i \in [N]$, the process $\{(y_{i,t}, x_{i,t})_{t \in \mathbb{Z}}\}$ is stationary and ergodic.

**Assumption 3.3.** For any $i \neq j$, $i, j \in [N]$, $\sigma(\{(y_{i,t}, x_{i,t}, \epsilon_{i,t})\}_{t \in \mathbb{Z}}) \perp\!\!\!\perp \sigma(\{(y_{j,t}, x_{j,t}, \epsilon_{j,t})\}_{t \in \mathbb{Z}})$

**Assumption 3.4.** Let $\mathbf{z}_{i,t} = (y_{t-1}, \ldots, y_{t-K}, x_{t-1}, \ldots, x_{t-K})^\top \in \mathbb{R}^{2K}$. We assume $\mathbf{z}_{i,t}$ follows a sub-Gaussian distribution with $\|\mathbf{z}_{i,t}\|_{\psi_2} \leq K_z < \infty$. Without loss of generality, we assume $\mathbb{E}(\mathbf{z}_{i,t}) = \mathbf{0}$ after demeaning with respect to the individual intercept. Additionally, $\widetilde{\mathbf{Q}}_i = \mathbb{E}(\mathbf{z}_{i,t}\mathbf{z}^\top_{i,t})$ exists and is positive definite. Furthermore, there exist positive constants $c_1, c_2 > 0$, such that $c_1 \leq \lambda_{\min}(\widetilde{\mathbf{Q}}_i) \leq c_2$.

Assumption 3.1 outlines regularity conditions on the individual innovations, and Assumption 3.2 requires stationarity and ergodicity of the individual time-series processes. Stationarity is imposed at the unit level and accommodates heterogeneity across units, and ergodicity guarantees that each unit provides sufficient time-series information for consistent estimation. These conditions are standard and commonly adopted in dynamic panel models (Fernández-Val & Lee, 2013). Furthermore, the sub-Gaussian tail condition provides a relaxation to the Gaussian error assumption that is prevalent in the literature of Granger causality tests (Dumitrescu & Hurlin, 2012; Minorics et al., 2022; Shojaie & Fox, 2022). Assumption 3.3 states cross-sectional independence, which is also common in the literature of panel Granger causality tests (Emirmahmutoglu & Kose, 2011; Dumitrescu & Hurlin, 2012; Juodis et al., 2021). Assumption 3.4 concerns the regularity of the stacked lag vector $\mathbf{z}_{i,t}$, which collects past outcomes and covariates. These conditions are standard in high-dimensional regression settings and play a crucial role in the Gaussian approximation for high-dimensional maxima (Chernozhukov et al., 2013).

With these assumptions, we study the asymptotic properties of the proposed PE component $J_{PE}$ and establish the asymptotic guarantees on the proposed PE-PGCT. It is worth mentioning that, although it appears natural that each individual Wald statistic $W_i$ asymptotically converges in distribution to a $\chi^2_K$ random variable, it is substantially more challenging to characterize the asymptotic distribution of $\max_{1 \leq i \leq N} W_i$ and its deviation to $\max_{1 \leq i \leq N} \chi^2_{K,i}$, particularly when $N$ diverges. Proposition 3.5 addresses this challenge.

**Proposition 3.5.** *With Assumptions* $3.1 - 3.4$, *under the null hypothesis* $H_0$, *we have*

$$\sup_{x>0} \left| \mathbb{P}\left( \max_{1 \leq i \leq N} W_i \geq x \right) - \mathbb{P}\left( \max_{1 \leq i \leq N} \chi^2_{K,i} \geq x \right) \right|$$

$$\lesssim \frac{\log^{7/6}(NT)}{T^{1/6}} + \frac{\log^{3/2} N}{\sqrt{T}} + \frac{\log^{7/4} N}{T^{5/4}} + \frac{1}{N^2}$$

*where* $\max_{1 \leq i \leq N} \chi^2_{K,i}$ *denotes the maximum of $N$ independent* $\chi^2_K$ *random variables.*

Proposition 3.5 provides a non-asymptotic upper bound on the Kolmogorov distance between $\max_{1 \leq i \leq N} W_i$ and $\max_{1 \leq i \leq N} \chi^2_{K,i}$. The term $\frac{\log^{7/6}(NT)}{T^{1/6}}$ arises from the Gaussian approximation error, $\frac{\log^{3/2} N}{\sqrt{T}}$ results from the random design linearization error, and $\frac{\log^{7/4} N}{T^{5/4}}$ reflects the studentization error. $N^{-2}$ is a remainder term accounting for the low-probability "bad" events induced by estimators in the intermediate steps, which vanishes quickly as $N \to \infty$.

*Remark* 3.6 (A standalone max-form test based on $M_{Wald}$). Though max-form tests have been studied in panel econometrics and high-dimensional statistics, they have not yet been explored in the relatively understudied area of panel Granger-causality testing. As an add-on contribution, Proposition 3.5 can be used to derive the asymptotic distribution of the max-form statistic $M_{Wald} := \max_{1 \leq i \leq N} W_i$ for testing panel Granger-causality, which may itself serve as a standalone testing procedure. We present detailed discussions in Appendix C.3.

**Proposition 3.7.** *With Assumptions 3.1 − 3.4, under the null hypothesis $H_0$, we have*

$$\mathbb{P}(\max_{1 \leq i \leq N} W_i \geq \delta_{N,T})$$
$$\lesssim \frac{1}{\sqrt{N}} + \frac{\log^{7/6}(NT)}{T^{1/6}} + \frac{\log^{3/2} N}{\sqrt{T}} + \frac{\log^{7/4} N}{T^{5/4}}$$

Proposition 3.7 provides a non-asymptotic upper bound on the probability that $J_{PE}$ selects spurious signals under the null hypothesis. This proposition forms the theoretical foundation of PE-PGCT in ensuring the proper Type-I error rate control after adding $J_{PE}$ to the pivotal test.

**Theorem 3.8** (Asymptotic Properties of the PE-component $J_{PE}$). *With Assumptions 3.1 − 3.4, and suppose $\log N = o(T^{1/7})$, $J_{PE}$ satisfies the following properties:*

*(i) Non-negativity: $J_{PE} \geq 0$.*

*(ii) No size distortion: $\mathbb{P}_{H_0}(J_{PE} = 0) \to 1$ as $N, T \to \infty$, where $\mathbb{P}_{H_0}(\cdot) = \mathbb{P}(\cdot \mid H_0)$ denotes the probability under the null hypothesis $H_0$.*

*(iii) Fast divergence under certain alternatives: consider the alternative space $\Theta_\beta$ defined as*

$$\Theta_\beta = \{(\boldsymbol{\beta}_1, \ldots, \boldsymbol{\beta}_N) :$$
$$\max_{1 \leq i \leq N} \boldsymbol{\beta}_i^\top \boldsymbol{\Sigma}_i^{-1} \boldsymbol{\beta}_i > \frac{\eta \cdot \log \log T \log N}{T}$$
$$for \ some \ \eta > 2\}, \qquad (10)$$

*for any $\boldsymbol{\beta} \in \Theta_\beta$, we have $J_{PE} \to \infty$ as $N, T \to \infty$.*

Theorem 3.8 presents the asymptotic properties of the proposed PE component. The three theoretical properties of $J_{PE}$ are essential for guaranteeing the power enhancement

properties of the proposed PE-PGCT, formalized in Theorem 3.10. Importantly, the alternative space $\Theta_\beta$ characterizes a sufficient condition for consistent detection of $J_{PE}$: it shows tha the presence of a single panel unit with signal strength exceeding the $(\eta \log \log T \log N)/T$ threshold is enough to trigger the divergence of $J_{PE}$. Consequently, $\Theta_\beta$ defines a class of sparse alternatives under which PE-PGCT is guaranteed to achieve asymptotically consistent power.

**Theorem 3.9** (Asymptotic Null Distribution of the PE-PGCT Statistic). *With Assumptions 3.1 − 3.4, and suppose $\log N = o(T^{1/7})$, the addition of the PE component $J_{PE}$ does not affect the limiting null distribution of the pivotal test $T_{pivotal}$, i.e., under the null hypothesis,*

$$T_{PE-PGCT} \overset{d}{=} T_{pivotal}$$

Theorem 3.9 characterizes the asymptotic null distribution of the PE-PGCT statistic. This theorem is a natural result of Property (ii) of Theorem 3.8. It reflects that $T_{PE-PGCT}$ shares the same limiting null distribution as $T_{pivotal}$, thereby confirming that adding $J_{PE}$ does not distort the Type-I error control. In the subsequent theorem, we demonstrate that adding $J_{PE}$ can substantially boost the testing power under the designated alternative space.

**Theorem 3.10** (Power Enhancement Properties of PE-PGCT). *Suppose $\Theta := \{(\boldsymbol{\beta}_1, \ldots, \boldsymbol{\beta}_N) : \boldsymbol{\beta}_i \in \mathbb{R}^K, \forall i = 1, \ldots, N\}$ denotes the parameter space, and $\Theta_0 = \{(\boldsymbol{\beta}_1, \ldots, \boldsymbol{\beta}_N) : \boldsymbol{\beta}_i = \mathbf{0}, \forall i = 1, \ldots, N\}$ denotes the null parameter space. Let $\phi_{PE-PGCT}(\boldsymbol{\beta}) = \mathbb{P}_{\boldsymbol{\beta}}(T_{PE-PGCT} \geq q^\alpha_{pivotal})$ denote the power function of PE-PGCT, and let $\phi_{pivotal}(\boldsymbol{\beta}) = \mathbb{P}_{\boldsymbol{\beta}}(T_{pivotal} \geq q^\alpha_{pivotal})$ denote the power function of the pivotal test $T_{pivotal}$. The following power enhancement properties hold:*

*(i) Type-I error rate control: for any $\boldsymbol{\beta} \in \Theta_0$, $\phi_{PE-PGCT}(\boldsymbol{\beta}) \to \alpha$ as $N, T \to \infty$;*

*(ii) Monotonicity (No power loss): for any $\boldsymbol{\beta} \in \Theta$, $\phi_{PE-PGCT}(\boldsymbol{\beta}) \geq \phi_{pivotal}(\boldsymbol{\beta})$;*

*(iii) Boosted power: $\forall \boldsymbol{\beta} \in \Theta_\beta$ where $\Theta_\beta$ is the sparse alternative space defined in (10), $\phi_{PE-PGCT}(\boldsymbol{\beta}) \to 1$ as $N, T \to \infty$.*

Theorem 3.10 establishes the PE properties of the proposed PE-PGCT. Property (i) guarantees no size distortion under the null. Property (ii) ensures no power loss, i.e., PE-PGCT is at least as powerful as the pivotal test. Property (iii) implies that PE-PGCT achieves consistent power as long as one marginal signal exceeds the threshold elaborated in $\Theta_\beta$. Furthermore, we note that the three PE properties are fully aligned with the general PE principles outlined by Fan et al. (2015), the seminal work in the PE literature.

*Table 1.* Empirical Type I error under the Gaussian error setting (nominal level $\alpha = 0.05$).

| $T$ | $N$ | PE-PGCT | DH | HPJ | $T$ | $N$ | PE-PGCT | DH | HPJ | $T$ | $N$ | PE-PGCT | DH | HPJ |
|---|---|---|---|---|---|---|---|---|---|---|---|---|---|---|
| 50 | 5 | 0.060 | 0.060 | 0.171 | 100 | 5 | 0.044 | 0.043 | 0.189 | 200 | 5 | 0.061 | 0.061 | 0.139 |
| | 10 | 0.052 | 0.052 | 0.104 | | 10 | 0.041 | 0.041 | 0.105 | | 10 | 0.052 | 0.052 | 0.099 |
| | 20 | 0.062 | 0.062 | 0.087 | | 20 | 0.056 | 0.056 | 0.101 | | 20 | 0.060 | 0.059 | 0.084 |
| | 50 | 0.074 | 0.083 | 0.072 | | 50 | 0.053 | 0.053 | 0.064 | | 50 | 0.065 | 0.065 | 0.067 |
| | 100 | 0.072 | 0.086 | 0.068 | | 100 | 0.050 | 0.061 | 0.050 | | 100 | 0.059 | 0.059 | 0.052 |
| | 200 | 0.069 | 0.104 | 0.064 | | 200 | 0.056 | 0.070 | 0.054 | | 200 | 0.057 | 0.058 | 0.057 |
| | 500 | 0.052 | 0.187 | 0.049 | | 500 | 0.052 | 0.097 | 0.052 | | 500 | 0.053 | 0.054 | 0.053 |

## 4. Simulation Studies

In this section, we carry out simulations to evaluate the finite-sample performance of PE-PGCT. We adopt the same data-generating process as in Dumitrescu & Hurlin (2012) and begin with $K = 1$. For $i = 1, \ldots, N$, we generate $\{x_{i,t}\}$ from an AR(1) process with the autoregressive coefficient $\phi_i$, and subsequently, $\{y_{i,t}\}$ following $y_{i,t} = \alpha_i + \gamma_i y_{i,t-1} + \beta_i x_{i,t-1} + \epsilon_{i,t}$, where the individual-specific parameters $\alpha_i, \phi_i, \gamma_i$ are generated from $\mathcal{U}[0.2, 0.8]$. We generate the innovations $\{\epsilon_{i,t}\}$ from different distributions, including a Gaussian error setting $\epsilon_{i,t} \sim \mathcal{N}(0, \sigma^2_{\epsilon,i})$ with each $\sigma^2_{\epsilon,i}$ randomly generated from $\mathcal{U}(0.5, 1.5)$ to mimic cross-sectional heteroscedasticity, as well as a non-Gaussian setting $\epsilon_{i,t} \sim$ standardized $t_3$. Due to space limit, non-Gaussian cases are reported in Appendix C.1. We compare PE-PGCT[1] with the Dumitrescu-Hurlin (DH) test (Dumitrescu & Hurlin, 2012) and Half-Panel Jackknife (HPJ) test (Juodis et al., 2021). A comparison of computation time is provided in Appendix C.2. Additionally, we compare our approach with the max-form test in Remark 3.6 and the method of Fan et al. (2015); the details are presented in Appendix C.3. All tests are conducted at the nominal level $\alpha = 0.05$. For each experimental design, we report the empirical rejection rate based on 1,000 replications, which corresponds to the empirical Type I error under the null and empirical power under the alternatives.

**Evaluation of Type I error control.** We set $\beta_i = 0$ for all $i \in [N]$ to simulate the null hypothesis of no Granger causality. We consider a range of panel dimensions with $T = 50, 100, 200$ and $N = 5, 10, 20, 50, 100, 200, 500$. The empirical rejection rates are summarized in Table 1. The difference in empirical Type-I error rates between PE-PGCT and DH (for $N < T - K$) or HPJ (for $N \geq T - K$) is an equivalent measurement of the null triggering rate $\widehat{\mathbb{P}}(J_{PE} > 0)$. As shown in the table, DH exhibits substantial inflated Type I errors when $T$ is small, with the size distortion becoming more severe in panels with $T \ll N$. In contrast, HPJ suffers from noticeable size distortion when $N$ is small. The proposed PE-PGCT, however, is able to

maintain the correct Type I error control across all settings considered.

**Evaluation of Power.** We consider heterogeneous alternatives by setting $\beta_i = \beta \neq 0$ for $i = 1, \ldots, N_0$ and $\beta_i = 0$, for $i = N_0 + 1, \ldots, N$, where $N_0$ represents the number of cross-sectional units with non-zero Granger-causal signals. By varying $\beta$ and $N_0$, we assess power performance along two directions: signal strength and signal sparsity. Since Table 1 shows DH and HPJ suffer from (sometimes severe) size distortions under certain configurations, we report size-corrected power for these two to ensure fair comparisons.

We first examine power under sparse alternatives with the number of signal-bearing coordinates $N_0$ set to be 1. The signal strength is varied over $\beta \in \{0.1, 0.2, \ldots, 0.9\}$. We consider the same range of $(N, T)$ configurations as in the size experiments. Figure 2 plots a representative subset of results, while the complete results are reported in Table S.2 of Appendix C. HPJ is nearly powerless under sparse alternatives. This aligns with the hypothesis testing literature that quadratic-form tests lose power under sparse alternatives (Yu et al., 2023; 2025; Li et al., 2025; Zhang et al., 2025). Evidently, PE-PGCT delivers substantial power gains. DH performs reasonably well when $T > N$, but its power deteriorates markedly when $T \leq N$. In all cases, PE-PGCT exhibits clear improvements over DH. Overall, PE-PGCT dominates the competing methods in sparse alternatives across all settings considered. These empirical findings resonate with theoretical properties established in Section 3.

Next, we evaluate power under various signal sparsity levels. We vary $N_0$ from 0 to 9, while fixing the individual signal strength at $\beta = 0.5$ for $T = 50$, 0.4 for $T = 100$, and 0.3 for $T = 200$. A representative subset of results is visualized in Figure 3, with complete results in Table S.3 of Appendix C. Once again, PE-PGCT consistently outperforms HPJ across all configurations considered. Relative to DH, PE-PGCT exhibits clear advantages when $N_0$ is small, corresponding to sparse alternatives, but slightly underperforms (or remains largely comparable) when the alternative becomes dense. This pattern is consistent with our earlier discussion: averaging-based tests such as DH are well-suited for dense alternatives but tend to suffer from

---

[1]The implementation code is available at https://github.com/yihenggu/PE-PGCT.

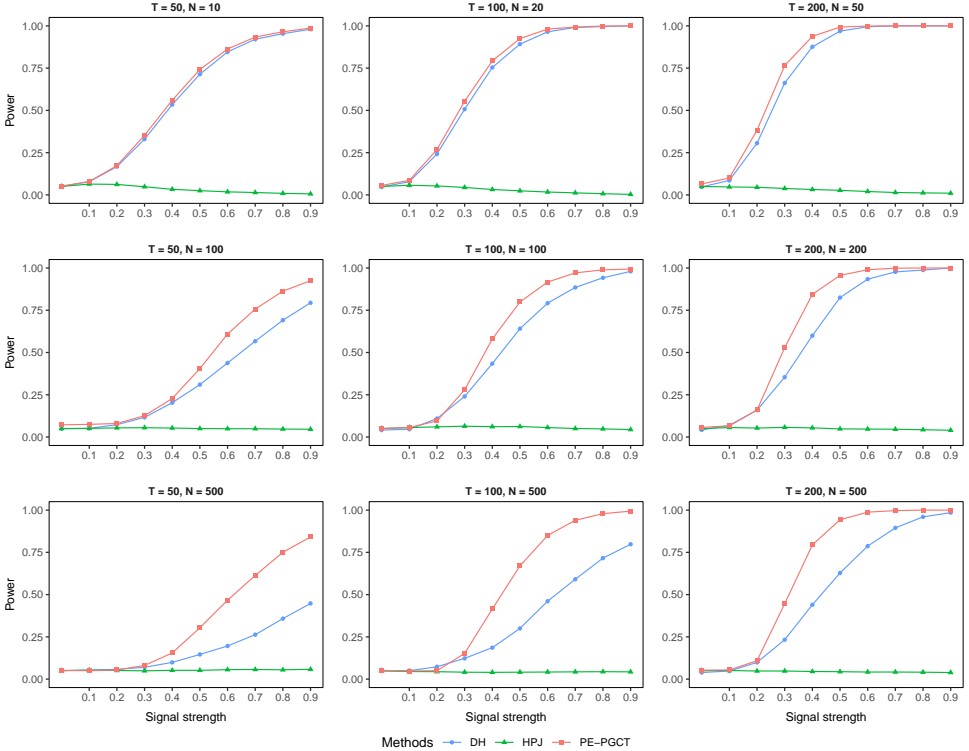

*Figure 2.* Empirical power under sparse alternatives ($N_0 = 1$) under the Gaussian error setting. Tests are conducted at level $\alpha = 0.05$.

power loss under sparse alternatives. It is also worth emphasizing that the DH results reported here are size-corrected power based on simulated null distributions. In practice, such size correction may not be feasible, and the asymptotic DH test is known to exhibit inflated Type I error when $T \leq N$ (as shown in Table 1). In contrast, PE-PGCT relies on an asymptotic distribution that does not require size correction, making it both convenient and reliable in empirical applications, an additional practical advantage.

## 5. Real Data Analysis

In this section, we investigate the Granger causal relationship between cost efficiency ($x_{i,t}$) and bank profitability ($y_{i,t}$). We use supervisory bank balance-sheet data from the Federal Deposit Insurance Corporation (FDIC) website. This dataset is publicly available and has also been used in related studies (e.g., Juodis et al. (2021)), facilitating reproducibility and comparisons across methods.

**Data Preparation.** $y_{i,t}$ is the profitability of bank $i$ at time $t$, proxied by the return on assets (ROA). ROA is calculated by the annualized net income after taxes and extraordinary items as a percent of the average total assets. $x_{i,t}$ is the operation cost efficiency of bank $i$ at time $t$. Following the standard practice of banking efficiency studies, we obtain

$x_{i,t}$ using a translog cost frontier model (Altunbas et al., 2007). Specifically, we consider

$$\log TC_{i,t} = \eta_i + \tau_t + \nu_{i,t} + \sum_{m=1}^{2} \sum_{n=1}^{3} \xi_{mn} \log Q_{m,i,t} \log P_{n,i,t}$$

$$+ \sum_{h=1}^{3} a_h \log P_{h,i,t} + \frac{1}{2} \sum_{m=1}^{3} \sum_{n=1}^{3} \delta_{mn} \log P_{m,i,t} \log P_{n,i,t}$$

$$+ \sum_{h=1}^{2} b_h \log Q_{h,i,t} + \frac{1}{2} \sum_{m=1}^{2} \sum_{n=1}^{2} \gamma_{mn} \log Q_{m,i,t} \log Q_{n,i,t}$$

where $TC_{i,t}$ refers to the total cost. $P_{1,i,t}$, $P_{2,i,t}$ and $P_{3,i,t}$ denote the capital price, labor price, and the price of loanable funds, respectively. $Q_{1,i,t}$ and $Q_{2,i,t}$ stand for the net loans and securities. The bank-specific, time-varying inefficiency is captured by $\eta_i + \tau_t$, which can be estimated using a two-way fixed effects regression. Then, the cost efficiency $x_{i,t}$ is obtained via $x_{i,t} = \exp\{\min\{\hat{\eta}_i + \hat{\tau}_t\}_{i,t} - (\hat{\eta}_i + \hat{\tau}_t)\}$.

**Panel Granger Causality Tests.** We analyze a panel consisting of $N = 300$ U.S. banking institutions observed quarterly from 2006:Q1 to 2019:Q4. Each institution is uniquely identified by its FDIC Bank Certificate Number (CERT). We record the CERTs and bank names in Appendix D. To assess panel Granger causality from a comprehensive perspective, we consider not only the full panel across all

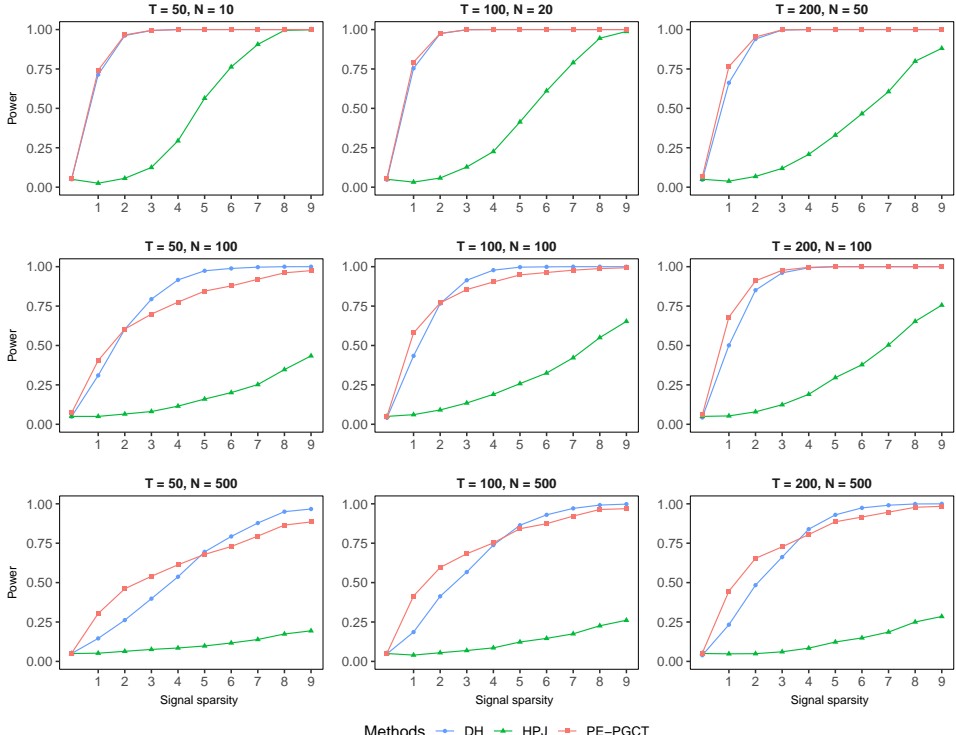

*Figure 3.* Empirical power in situations of different sparsity levels (signal number ranging from 0 to 9) under the Gaussian error setting. Tests are conducted at the nominal level $\alpha = 0.05$.

institutions and time periods, but also sub-panels defined along the time-series and cross-sectional dimensions.

Across the time dimension, we divide the panel into two sub-periods: 2006:Q1 - 2010:Q4 (named "Basel II") and 2011:Q1 - 2019:Q4 (named "DFA"). This partition is economically motivated by major changes in the regulatory regime before and after the global financial crisis. The Basel II sub-period corresponds to the regulatory environment dominated by the Basel II framework, in which capital requirements relied heavily on risk-weighted assets and internal models, increasing banks' flexibility but also their exposure to model risk and procyclicality. This period also includes the 2007–2009 global financial crisis, which exposed major weaknesses of Basel II and represents pre- and crisis-era banking behavior before comprehensive post-crisis reforms were implemented. The DFA sub-period reflects the post-crisis regulatory regime shaped primarily by the Dodd–Frank Act, which became effective from 2011 onward. This period is characterized by tighter supervision, stress testing, higher capital and liquidity standards, and enhanced oversight of systemically important banks. These reforms fundamentally changed banks' incentives and risk-taking behavior, making this sub-period a natural post-reform comparison window.

Along the cross-sectional dimension, we divide banks into two groups using the $k$-means algorithm based on their average bank size, proxied by the log total assets. This procedure yields 220 small banks and 80 large banks. Summary statistics for the two groups are reported in Table S.9.

To assess the Granger causality, we set the lag order to be 1 and employ the pivotal test defined in (9). We compare PE-PGCT with the pivotal test across the full panel and sub-panels defined above. The $p$-values are reported in Table 2, along with the banks identified by the PE component $J_{PE}$.

**Empirical Findings.** Table 2 shows PE-PGCT and the pivotal test reach consistent conclusions across all settings. We also observe that PE-PGCT is at least as powerful as the pivotal test, which is as expected by design.

The results suggest that cost efficiency Granger-causes profitability at the significance level of 1% in nearly all cases, with the only exception being that large banks temporarily decoupled profits from efficiency under Basel II but this link is restored under tighter post-crisis DFA regulation. These findings are well aligned with the mainstream empirical literature. A large body of literature has shown that operational efficiency is a primary driver of bank profitability (Berger & Mester, 1997; Berger & Humphrey, 1997; Athanasoglou et al., 2008). The absence of causality for large banks dur-

*Table 2.* $p$-values of the PE-PGCT and Pivotal tests, along with the PE component $J_{PE}$ and banks identified by $J_{PE}$.

| Group | Period | $J_{PE}$ | PE-PGCT | Pivotal | Banks identified by $J_{PE}$ |
|-------|--------|---------|---------|---------|------------------------------|
| All | Full | 489.313 | $< 10^{-16}$ | $1.1 \times 10^{-16}$ | Elevate Bank (CERT 4770), Hilltop National Bank (CERT 19184) |
| | Basel II | 0 | $< 10^{-16}$ | $< 10^{-16}$ | – |
| | DFA | 452.419 | $< 10^{-16}$ | $< 10^{-16}$ | First State Bank of San Diego (CERT 17350) |
| Small | Full | 419.023 | $< 10^{-16}$ | $1.1 \times 10^{-12}$ | First National Bank of Bellville (CERT 3099), Mid Penn Bank (CERT 9889) |
| | Basel II | 0 | $< 10^{-16}$ | $< 10^{-16}$ | – |
| | DFA | 387.428 | $< 10^{-16}$ | $4.5 \times 10^{-12}$ | Solvay Bank (CERT 13297) |
| Large | Full | 248.584 | $< 10^{-16}$ | $9.3 \times 10^{-10}$ | First National Bank of Milaca (CERT 5198) |
| | Basel II | 0 | 0.011 | 0.011 | – |
| | DFA | 0 | $1.1 \times 10^{-4}$ | $1.1 \times 10^{-4}$ | – |

ing the Basel II period aligns with evidence that large-bank profitability before the financial crisis was driven more by risk-taking, trading income, and funding advantages associated with size than by operating efficiency (DeYoung & Torna, 2013; Juodis et al., 2021; Cui et al., 2023). The re-emergence of the causal relationship in the DFA period is consistent with post-crisis regulatory reforms that reduced the "Too Big to Fail" advantage and restored the importance of managerial efficiency for profitability (Wheelock & Wilson, 2018; Cetorelli & Traina, 2021).

Moreover, by incorporating the PE component, PE-PGCT uncovers additional findings of Granger causality at the unit level, offering finer-grained insights and revealing new aspects of the underlying dynamics. For example, during the DFA period, First State Bank of San Diego (FSBSD, CERT 17350) is identified from our test among all banks. The findings support the "Bad Management Hypothesis", illustrated in the work of Berger & DeYoung (1997), that low cost efficiency Granger-causes an increase of non-performing loans, which subsequently affects the profit. As a small community bank, when the influence of significant compliance costs increased due to the DFA, marginal changes in operating expenses reacted more quickly and were a more direct and critical predictor of earning conditions. Moreover, San Diego's rising unemployment in late 2011 impacted banking and business, forcing FSBSD to manage cost controls. Meanwhile, when facing an increasing burden of post DFA era, Solvay Bank (CERT 13297) is identified among small banks during 2011-2019. Similarly, as a lean community bank, Solvay Bank is strongly focused on local lending, which enhances the predictive impact of operational control on profitability. For the full period, the Elevate Bank (previously named as First National Bank of Sedan before 2023, CERT 4770), the First National Bank of Bellville (CERT 3099), and the First National Bank of Milaca (CERT 5198) are identified among all banks, small banks, and large banks, respectively. The possible reason behind the strong signals is that this period includes the Great Recession and the post-Dodd-Frank regulatory surge, where these banks use efficiency gains or specialized lending focus to maintain profitability in spite of rising external costs.

## 6. Discussion

In this paper, we propose PE-PGCT, a powerful framework for assessing the Granger non-causality in large-scale heterogeneous panels that delivers substantial power gains, particularly under sparse alternatives. While we focus on the relatively underexplored problem of panel Granger causality testing to present the methodology in a focused and coherent manner, the proposed power enhancement technique is not restricted to PGCT. More broadly, it can be naturally adapted and applicable to a wide range of high-dimensional inference problems.

There are a few promising directions for future work. Consistent with related studies such as Emirmahmutoglu & Kose (2011); Dumitrescu & Hurlin (2012) and Juodis et al. (2021), we assume independent error terms to facilitate the theoretical analysis and obtain tractable asymptotic results. We acknowledge that this assumption may be somewhat restrictive in practical applications. To overcome this limitation, Emirmahmutoglu & Kose (2011); Dumitrescu & Hurlin (2012) and Xiao et al. (2023) discussed practical strategies using block bootstrap, while Juodis et al. (2021) discussed the possibility of modeling the cross-sectional dependence through latent factor structures. Though not reported in the paper, we conducted additional experiments under moderately correlated errors, and the test performance remains empirically robust. Nevertheless, developing a theoretical framework that rigorously accommodates such cross-sectional dependence is an important and nontrivial extension. It would require substantially different technical tools, and we shall leave it for future research.

Moreover, our current framework assumes stationarity, which is standard in panel Granger causality settings to ensure valid inference. In practice, non-stationarity can be potentially addressed by data preprocessing. Extending the methodology to non-stationary scenarios is an important direction for future research. In principle, we expect such an extension would require adapting the test to account for common stochastic trends or employing transformations to restore stationarity, yet a thorough investigation is beyond the scope of this work and shall be left for future work.

## Acknowledgements

We thank the anonymous reviewers for helpful suggestions. This work is supported in part by National Institutes of Health grant R01GM152812.

## Impact Statement

This paper advances methods for Granger causality analysis. Improved methods may have positive impacts across many domains, including economics, climate science, neuroscience, epidemiology, and other scientific fields where understanding temporal relationships is critical for decision making and hypothesis generation.

At the same time, Granger-causal analysis may be misapplied or over-interpreted, particularly when assumptions are violated. Incorrect Granger-causal conclusions could lead to flawed scientific inferences or poor downstream decisions in high-stakes settings such as policy making, healthcare, or finance. We therefore emphasize that the proposed method is intended as a statistical tool to support, rather than replace, domain expertise and careful experimental design.

We do not foresee direct negative societal impacts arising from this work. As with other advances in Granger-causal analysis and machine learning, responsible use requires transparency about assumptions, limitations, and uncertainty when applying these methods in real-world contexts.

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

# Appendices

The appendices of this paper are organized into four parts. Appendix A contains technical lemmas and the proofs of all propositions and theorems presented in Section 3. Appendix B provides additional details on the pivotal test $T_{\text{pivotal}}$, supplementing the discussion in Section 2.2. Appendix C presents further simulation results that complement those in Section 4. Finally, Appendix D provides supplementary information on the data used in the real data analysis in Section 5.

## A. Technical Lemmas and Proofs

### A.1. Lemmas

We first present some helpful lemmas and their proofs, which help facilitate the proofs of the propositions and theorems in the main context.

**Lemma A.1.** *Let* $M_{\chi^2} := \max_{1 \leq i \leq N} \chi^2_{K,i}$ *denote the max of* $N (\geq 2)$ *independent* $\chi^2_K$ *random variables. For any* $\varepsilon > 0$, *there exists* $C > 0$, *such that*

$$\sup_x \mathbb{P}\left(|M_{\chi^2_K} - x| \leq \varepsilon\right) \leq C\varepsilon. \tag{S.1}$$

*Proof of Lemma A.1.* Let $U_1, \ldots, U_N \overset{i.i.d.}{\sim} \chi^2_K$ with the cumulative density function (CDF) $F(x)$ and probability density function (PDF) $f(x)$. For convenience, we define the survival function $\bar{F}(x) = 1 - F(x)$ and the hazard function $h(x) = f(x)/\bar{F}(x)$. Then the CDF and PDF of $M_{\chi^2} := \max_{1 \leq i \leq N} U_i$ can be denoted using

$$F_{M_{\chi^2}}(x) = \mathbb{P}(M_{\chi^2} \leq x) = (F(x))^N, \quad f_{M_{\chi^2}}(x) = Nf(x)(F(x))^{N-1}.$$

For any $\varepsilon > 0$,

$$\mathbb{P}(x < M_{\chi^2} \leq x + \varepsilon) = \int_x^{x+\varepsilon} f_{M_{\chi^2}}(t)dt \leq \varepsilon \cdot \sup_t f_{M_{\chi^2}}(t).$$

Hence,

$$\sup_x \mathbb{P}\left(|M_{\chi^2_K} - x| \leq \varepsilon\right) \leq 2\varepsilon \cdot \sup_t f_{M_{\chi^2}}(t) \tag{S.2}$$

It suffices to show, there exists some positive constant $C$, such that

$$\sup_t f_{M_{\chi^2}}(t) \leq C < \infty. \tag{S.3}$$

We begin with the case $K = 1$. The CDF and PDF of $\chi^2_1$ can be rewritten as

$$F(x) = 2\Phi(\sqrt{x}) - 1, \quad f(x) = F'(x) = \frac{\phi(\sqrt{x})}{\sqrt{x}},$$

where $\Phi(x)$ and $\phi(x) = \frac{1}{\sqrt{2\pi}} \exp(-\frac{1}{2}x^2)$ are CDF and PDF of the standard normal distribution. Rewriting $f_{M_{\chi^2}}(x)$ into the hazard form,

$$f_{M_{\chi^2}}(x) = Nf(x)(F(x))^{N-1} = h(x) \cdot N\bar{F}(x)(1 - \bar{F}(x))^{N-1}.$$

Let $g(u) := Nu(1-u)^{N-1}, u \in [0, 1]$. We can show that for any $u \in [0, 1], \sup_{u \in [0,1]} g(u) = g(N^{-1}) = (1 - N^{-1})^{N-1} \leq 2e^{-1}$. Therefore,

$$\sup_x f_{M_{\chi^2}}(x) \leq 2e^{-1} \sup_x h(x). \tag{S.4}$$

For $\chi^2$ distribution with degree of freedom $K = 1$, the hazard function takes the form

$$h(x) = \frac{f(x)}{\bar{F}(x)} = \frac{\phi(\sqrt{x})/\sqrt{x}}{2 - 2\Phi(\sqrt{x})} = \frac{1}{2\sqrt{x}} \cdot \frac{\phi(\sqrt{x})}{1 - \Phi(\sqrt{x})}$$

With the Mills-ratio inequality,

$$h(x) \leq \frac{1}{2\sqrt{x}} \left(\sqrt{x} + \frac{1}{\sqrt{x}}\right) = \frac{1}{2} + \frac{1}{2x}.$$

Given this, we will bound $f_{M_{\chi^2}}(x)$ by splitting the domain into $x \in (0, 1]$ and $x \geq 1$.

For any $x \geq 1$, $h(x) \leq \frac{1}{2} + \frac{1}{2} = 1$, and hence,

$$\sup_{t \geq 1} f_{M_{\chi^2}}(t) \leq 2e^{-1} \sup_{t \geq 1} h(t) \leq 2e^{-1}. \tag{S.5}$$

For any $x \in (0, 1]$, since $\phi(x) \geq \phi(0) = (2\pi)^{-\frac{1}{2}}$, we have

$$\Phi(x) = \Phi(0) + \int_0^x \phi(t)dt \leq \Phi(0) + \int_0^x \phi(0)dt = \frac{1}{2} + x \cdot \phi(0) = \frac{1}{2} + \frac{x}{\sqrt{2\pi}}$$

and naturally,

$$F(x) = 2\Phi(\sqrt{x}) - 1 \leq 2 \cdot \left(\frac{1}{2} + \frac{x}{\sqrt{2\pi}}\right) - 1 = \frac{\sqrt{2x}}{\sqrt{\pi}}$$

and

$$f(x) = \frac{\phi(x)}{\sqrt{x}} \leq \frac{\phi(0)}{\sqrt{x}} = \frac{1}{\sqrt{2\pi x}}.$$

Thus, when $x \in (0, 1]$ and $N \geq 2$,

$$f_{M_{\chi^2}}(x) = Nf(x)(F(x))^{N-1} \leq N \cdot \frac{1}{\sqrt{2\pi x}} \cdot \left(\frac{\sqrt{2x}}{\sqrt{\pi}}\right)^N = \frac{N}{\sqrt{2\pi}} \left(\frac{2}{\pi}\right)^{\frac{N-1}{2}} x^{\frac{N-2}{2}} \leq \frac{N}{\sqrt{2\pi}} \left(\frac{2}{\pi}\right)^{\frac{N-1}{2}}$$

For any constant $a \in (0, 1)$, we have the identity $\sum_{k=0}^{\infty}(k+1)a^k = (1-a)^{-2}$. Let $a = \sqrt{2/\pi}$, we can bound $N(\sqrt{2/\pi})^{N-1}$ via $Na^{N-1} \leq \sum_{k=N-1}^{\infty}(k+1)a^k \leq (1-a)^{-2}$. As a result,

$$\sup_{0 < t < 1} f_{M_{\chi^2}}(t) \leq \frac{1}{\sqrt{2\pi}} \cdot \frac{1}{\left(1 - \sqrt{\frac{2}{\pi}}\right)^2} \tag{S.6}$$

With (S.5) and (S.6), when $K = 1$,

$$\sup_t f_{M_{\chi^2}}(t) = \max\left\{\sup_{0<t<1} f_{M_{\chi^2}}(t), \sup_{t\geq 1} f_{M_{\chi^2}}(t)\right\} \leq \frac{1}{\sqrt{2\pi}} \cdot \frac{1}{\left(1 - \sqrt{\frac{2}{\pi}}\right)^2} < \infty. \tag{S.7}$$

Next, we consider the case $K \geq 2$. Again, after rewriting $f_{M_{\chi^2}}(x)$ into the hazard form, we have (S.4). To bound $\sup_x f_{M_{\chi^2}}(x)$, it suffices to bound $\sup_x h(x)$. Note that $\chi_K^2$ is essentially a Gamma$(\alpha, \theta)$ distribution with the shape $\alpha = \frac{K}{2}$ and scale $\theta = 2$, whose density function and survival function are

$$f(x) = \frac{1}{\Gamma(\alpha)\theta^\alpha} x^{\alpha-1} \exp\left(-\frac{x}{\theta}\right), \quad \bar{F}(x) = \int_x^\infty f(t)dt$$

Since the survival function of Gamma distribution admits a recursive property that

$$\bar{F}(x) = \int_x^\infty \frac{1}{\Gamma(\alpha)\theta^\alpha} x^{\alpha-1} \exp\left(-\frac{x}{\theta}\right) dt = \frac{\theta x^{\alpha-1} e^{-\frac{x}{\theta}}}{\Gamma(\alpha)\theta^\alpha} + \frac{\theta(\alpha-1)}{\Gamma(\alpha)\theta^\alpha} \int_x^\infty t^{\theta-2} e^{-\frac{t}{\theta}} dt$$

By the non-negativity of the integral term,

$$\bar{F}(x) \geq \frac{\theta x^{\alpha-1} e^{-\frac{x}{\theta}}}{\Gamma(\alpha)\theta^\alpha} = \theta f(x).$$

Hence, for any $x > 0$,

$$h(x) = \frac{f(x)}{\bar{F}(x)} \leq \frac{1}{\theta} = \frac{1}{2}.$$

i.e., for all $K \geq 2$,

$$\sup_x f_{M_{\chi^2}}(x) \leq 2e^{-1} \sup_x h(x) \leq 2e^{-1} \cdot \frac{1}{2} = e^{-1} < \infty. \tag{S.8}$$

(S.7) and (S.8) verify (S.3). Together with (S.2), we complete the proof of (S.1). □

## A.2. Proof of Proposition 3.5

*Proof.* Recall that, for each $i = 1, \ldots, N$, $W_i$ is the Wald statistic for testing

$$H_{0i} : \boldsymbol{\beta}_i = (\beta_{i1}, \ldots, \beta_{iK})^\top = \mathbf{0} \tag{S.9}$$

in the linear panel model (1). Let $\mathbf{z}_{i,t} = (y_{t-1}, \ldots, y_{t-K}, x_{t-1}, \ldots, x_{t-K})^\top \in \mathbb{R}^{2K}$ and $\boldsymbol{\theta}_i = (\gamma_{i1}, \ldots, \gamma_{iK}, \beta_{i1}, \ldots, \beta_{iK})^\top \in \mathbb{R}^{2K}$. The model (1) becomes

$$y_{i,t} = \alpha_i + \mathbf{z}_{i,t}^\top \boldsymbol{\theta}_i + \epsilon_{i,t},$$

or equivalently,

$$\mathbf{y}_{i\cdot} = \alpha_i \mathbf{1}_T + \mathbf{Z}_i^\top \boldsymbol{\theta}_i + \boldsymbol{\varepsilon}_{i\cdot},$$

where $\mathbf{y}_{i\cdot} = (y_{i,1}, \ldots, y_{i,T})^\top \in \mathbb{R}^T$, $\mathbf{1}_T \in \mathbb{R}^T$ is a vector of 1s of length $T$, $\mathbf{Z}_i = (\mathbf{z}_{i,1}, \ldots, \mathbf{z}_{i,T})^\top \in \mathbb{R}^{T \times 2K}$, and $\boldsymbol{\varepsilon}_{i\cdot} = (\epsilon_{i,1}, \ldots, \epsilon_{i,T})^\top \in \mathbb{R}^T$. Without loss of generality, we assume $\mathbf{Z}_i$ is centered satisfying $\mathbf{1}_T^\top \mathbf{Z}_i = \mathbf{0}$. The null hypothesis (S.9) becomes

$$H_{0i} : \mathbf{R}\boldsymbol{\theta}_i = \mathbf{0} \tag{S.10}$$

with $\mathbf{R} = [\mathbf{0}_{K \times K}, \mathbf{I}_{K \times K}] \in \mathbb{R}^{K \times 2K}$. The OLS estimator of $\boldsymbol{\theta}_i$ can be obtained by $\widehat{\boldsymbol{\theta}}_i = (\mathbf{Z}_i^\top \mathbf{Z}_i)^{-1} \mathbf{Z}_i \mathbf{y}_{i\cdot}$, and the Wald statistic $W_i$ takes the form of

$$\begin{aligned} W_i &= (\mathbf{R}\widehat{\boldsymbol{\theta}}_i)^\top [\widehat{\sigma}_i^2 \mathbf{R}(\mathbf{Z}_i^\top \mathbf{Z}_i)^{-1} \mathbf{R}^\top]^{-1} (\mathbf{R}\widehat{\boldsymbol{\theta}}_i) \\ &= \boldsymbol{\varepsilon}_{i\cdot}^\top \mathbf{Z}_i (\mathbf{Z}_i^\top \mathbf{Z}_i)^{-1} \mathbf{R}^\top [\widehat{\sigma}_i^2 \mathbf{R}(\mathbf{Z}_i^\top \mathbf{Z}_i)^{-1} \mathbf{R}^\top]^{-1} \mathbf{R}(\mathbf{Z}_i^\top \mathbf{Z}_i)^{-1} \mathbf{Z}_i^\top \boldsymbol{\varepsilon}_{i\cdot}. \end{aligned} \tag{S.11}$$

Denote $\mathbf{Q}_i = \frac{1}{n} \mathbf{Z}_i^\top \mathbf{Z}_i$ and $\widetilde{\mathbf{Q}}_i = \mathbb{E}(\mathbf{z}_{it} \mathbf{z}_{it}^\top)$, then the Wald statistic $W_i$ becomes

$$W_i = \frac{1}{T \widehat{\sigma}_i^2} \boldsymbol{\varepsilon}_{i\cdot}^\top \mathbf{Z}_i \mathbf{Q}_i^{-1} \mathbf{R}^\top [\mathbf{R}\mathbf{Q}_i^{-1} \mathbf{R}^\top]^{-1} \mathbf{R}\mathbf{Q}_i^{-1} \mathbf{Z}_i^\top \boldsymbol{\varepsilon}_{i\cdot}. \tag{S.12}$$

Naturally, with Assumptions (A1)–(A4), $W_i$ asymptotically follows a $\chi_K^2$ distribution. We denote $W_i \xrightarrow{d} U_i \sim \chi_K^2$, where each $U_i$ is marginally a $\chi_K^2$ random variable, and the dependence among $(U_1, \ldots, U_N)^\top$ resembles the dependence among $(W_1, \ldots, W_N)^\top$. It suffices to show the Kolmogorov bound between $\max_{1 \leq i \leq N} W_i$ and $\max_{1 \leq i \leq N} U_i$. To this end, we further consider

$$W_i^{(0)} = \frac{1}{T \sigma_i^2} \boldsymbol{\varepsilon}_{i\cdot}^\top \mathbf{Z}_i \mathbf{Q}_i^{-1} \mathbf{R}^\top [\mathbf{R}\mathbf{Q}_i^{-1} \mathbf{R}^\top]^{-1} \mathbf{R}\mathbf{Q}_i^{-1} \mathbf{Z}_i^\top \boldsymbol{\varepsilon}_{i\cdot}. \tag{S.13}$$

and

$$\widetilde{W}_i = \frac{1}{T \sigma_i^2} \boldsymbol{\varepsilon}_{i\cdot}^\top \mathbf{Z}_i \widetilde{\mathbf{Q}}_i^{-1} \mathbf{R}^\top [\mathbf{R}\widetilde{\mathbf{Q}}_i^{-1} \mathbf{R}^\top]^{-1} \mathbf{R}\widetilde{\mathbf{Q}}_i^{-1} \mathbf{Z}_i^\top \boldsymbol{\varepsilon}_{i\cdot}. \tag{S.14}$$

Let $M_{Wald} = \max_{1 \leq i \leq N} W_i$, $M_{Wald}^{(0)} = \max_{1 \leq i \leq N} W_i^{(0)}$, $\widetilde{M}_{Wald} = \max_{1 \leq i \leq N} \widetilde{W}_i$, $M_{\chi_K^2} = \max_{1 \leq i \leq N} U_i$, and let $d(U, V) = \sup_x |\mathbb{P}(X \leq x) - \mathbb{P}(Y \leq x)|$ denote the Kolmogorov distance between the distributions of two random variables $X$ and $Y$. Then,

$$\begin{aligned} d(M_{Wald}, M_{\chi_K^2}) &= \sup_x |\mathbb{P}(M_{Wald} \leq x) - \mathbb{P}(M_{\chi_K^2} \leq x)| \\ &\leq \sup_x |\mathbb{P}(\widetilde{M}_{Wald} \leq x) - \mathbb{P}(M_{\chi_K^2} \leq x)| \\ &\quad + \sup_x |\mathbb{P}(M_{Wald}^{(0)} \leq x) - \mathbb{P}(\widetilde{M}_{Wald} \leq x)| \\ &\quad + \sup_x |\mathbb{P}(M_{Wald} \leq x) - \mathbb{P}(M_{Wald}^{(0)} \leq x)| \\ &:= d(\widetilde{M}_{Wald}, M_{\chi_K^2}) + d(M_{Wald}^{(0)}, \widetilde{M}_{Wald},) + d(M_{Wald}, M_{Wald}^{(0)}) \end{aligned} \tag{S.15}$$

where $d(M_{\chi_K^2}, \widetilde{M}_{Wald})$ is the Gaussian approximation error, $d(\widetilde{M}_{Wald}, M_{Wald}^{(0)})$ is the random design linearization error, and $d(M_{Wald}^{(0)}, M_{Wald})$ is the studentization error.

In what follows, we prove the non-asymptotic error bounds on the three error components in Lemma A.2, Lemma A.3, and Lemma A.4, respectively. Combining (S.16), (S.17), and (S.29), we obtain

$$\sup_{x>0}\left|\mathbb{P}\left(\max_{1\le i\le N}W_i\ge x\right)-\mathbb{P}\left(\max_{1\le i\le N}\chi^2_{K,i}\ge x\right)\right|\lesssim\frac{\log^{7/6}(NT)}{T^{1/6}}+\frac{\log^{3/2}N}{\sqrt{T}}+\frac{\log^{7/4}N}{T^{5/4}}+\frac{1}{N^2}.$$

This completes the proof of Proposition 3.5. $\qquad\square$

**Lemma A.2** (Non-asymptotic error bound on the Gaussian approximation error). *Under the same assumptions as in Proposition 3.5,*

$$d(\widetilde{M}_{Wald},M_{\chi^2_K})\lesssim\left(\frac{\log^7(NT)}{T}\right)^{1/6}.\tag{S.16}$$

*Proof of Lemma A.2.* We first rewrite $\widetilde{W}_i$ into

$$\widetilde{W}_i=\frac{1}{T\sigma_i^2}\boldsymbol{\varepsilon}_{i\cdot}^\top\mathbf{Z}_i\widetilde{\mathbf{Q}}_i^{-1}\mathbf{R}^\top[\mathbf{R}\widetilde{\mathbf{Q}}_i^{-1}\mathbf{R}^\top]^{-1}\mathbf{R}\widetilde{\mathbf{Q}}_i^{-1}\mathbf{Z}_i^\top\boldsymbol{\varepsilon}_{i\cdot}$$

$$=\left\|[\mathbf{R}\widetilde{\mathbf{Q}}_i^{-1}\mathbf{R}^\top]^{-1}\mathbf{R}\widetilde{\mathbf{Q}}_i^{-1}\frac{1}{\sqrt{T}\sigma_i}\sum_{t=1}^T\mathbf{z}_{i,t}\epsilon_{i,t}\right\|_2^2:=\left\|\frac{1}{\sqrt{T}}\sum_{t=1}^T\boldsymbol{\xi}_{i,t}\right\|_2^2$$

where $\boldsymbol{\xi}_{i,t}:=\sigma_i^{-1}[\mathbf{R}\widetilde{\mathbf{Q}}_i^{-1}\mathbf{R}^\top]^{-1}\mathbf{R}\widetilde{\mathbf{Q}}_i^{-1}\mathbf{z}_{i,t}\epsilon_{i,t}\in\mathbb{R}^K$ with $\mathbb{E}(\boldsymbol{\xi}_{i,t})=\mathbf{0}_K$ and $\mathrm{cov}(\boldsymbol{\xi}_{i,t})=\mathbf{I}_K$. Let $\boldsymbol{\xi}_{i,t}^G$ be the Gaussian counterpart of $\boldsymbol{\xi}_{i,t}$, satisfying $\boldsymbol{\xi}_{i,t}^G\sim\mathcal{N}(\mathbf{0}_K,\mathbf{I}_K)$, and therefore $\|\boldsymbol{\xi}_{i,t}^G\|_2^2\sim\chi^2_K$. Let $\boldsymbol{\zeta}_{i,t}=\mathbf{z}_{i,t}\epsilon_{i,t}\in\mathbb{R}^{2K}$. Assumptions 3.1 and 3.4 imply that $\boldsymbol{\zeta}_{i,t}$ follows a sub-exponential distribution, as a result, for any $k\in[2K]$, there exists $C>0$, such that $\mathbb{E}(|\boldsymbol{\zeta}_{i,t,k}|/C)\le 2$, $i\in[N]$, $t\in[T]$. Hence, the high-dimensional central limit theorem (Theorem 2.1 and Proposition 2.1, Chernozhukov et al., 2017) implies,

$$\sup_x\left|\mathbb{P}\left(\max_{1\le i\le N}\left\|\frac{1}{\sqrt{T}}\sum_{t=1}^T\boldsymbol{\xi}_{i,t}\right\|_2^2\le x\right)-\mathbb{P}\left(\max_{1\le i\le N}\left\|\frac{1}{\sqrt{T}}\sum_{t=1}^T\boldsymbol{\xi}_{i,t}^G\right\|_2^2\le x\right)\right|\le C\left(\frac{\log^7(NT)}{T}\right)^{1/6},$$

i.e.,

$$d(\widetilde{M}_{Wald},M_{\chi^2_K})=\sup_x|\mathbb{P}(\widetilde{M}_{Wald}\le x)-\mathbb{P}(M_{\chi^2_K}\le x)|\le C\left(\frac{\log^7(NT)}{T}\right)^{1/6}.$$

This completes the proof of (S.16) in Lemma A.2. $\qquad\square$

**Lemma A.3** (Non-asymptotic error bound on the random design linearization error). *Under the same assumptions as in Proposition 3.5,*

$$d(M_{Wald}^{(0)},\widetilde{M}_{Wald})\lesssim\frac{(\log N)^{3/2}}{\sqrt{T}}+\left(\frac{\log^7(NT)}{T}\right)^{1/6}+\frac{1}{N^2}.\tag{S.17}$$

*Proof of Lemma A.3.* Note that, for any $x>0$ and any $\varepsilon>0$,

$$\left|\mathbb{P}\left(\max_{1\le i\le N}W_i^{(0)}\le x\right)-\mathbb{P}\left(\max_{1\le i\le N}\widetilde{W}_i\le x\right)\right|$$

$$\le\mathbb{P}\left(\left|\max_{1\le i\le N}\widetilde{W}_i-x\right|\le\varepsilon\right)+\mathbb{P}\left(\left|\max_{1\le i\le N}W_i^{(0)}-\max_{1\le i\le N}\widetilde{W}_i\right|>\varepsilon\right)$$

$$\le\mathbb{P}\left(\left|\widetilde{M}_{Wald}-x\right|\le\varepsilon\right)+\mathbb{P}\left(\max_{1\le i\le N}\left|W_i^{(0)}-\widetilde{W}_i\right|>\varepsilon\right)$$

$$\le\mathbb{P}\left(\left|M_{\chi^2_K}-x\right|\le\varepsilon\right)+2\cdot\sup_x|\mathbb{P}(\widetilde{M}_{Wald}\le x)-\mathbb{P}(M_{\chi^2_K}\le x)|+\mathbb{P}\left(\max_{1\le i\le N}\left|W_i^{(0)}-\widetilde{W}_i\right|>\varepsilon\right).$$

Hence, for any $\varepsilon > 0$,

$$
\begin{aligned}
d(M_{Wald}^{(0)}, \widetilde{M}_{Wald}) &= \sup_x \left| \mathbb{P}\left( \max_{1 \leq i \leq N} W_i^{(0)} \leq x \right) - \mathbb{P}\left( \max_{1 \leq i \leq N} \widetilde{W}_i \leq x \right) \right| \\
&\leq \sup_x \mathbb{P}\left( |M_{\chi_K^2} - x| \leq \varepsilon \right) + 2 \cdot d(M_{\chi_K^2}, \widetilde{M}_{Wald}) + \mathbb{P}\left( \max_{1 \leq i \leq N} \left| W_i^{(0)} - \widetilde{W}_i \right| > \varepsilon \right).
\end{aligned} \tag{S.18}
$$

Let $h(\widetilde{\mathbf{Q}}_i) := \widetilde{\mathbf{Q}}_i^{-1} \mathbf{R}^\top [\mathbf{R} \widetilde{\mathbf{Q}}_i^{-1} \mathbf{R}^\top]^{-1} \mathbf{R} \widetilde{\mathbf{Q}}_i^{-1}$, $h(\mathbf{Q}_i) := \mathbf{Q}_i^{-1} \mathbf{R}^\top [\mathbf{R} \mathbf{Q}_i^{-1} \mathbf{R}^\top]^{-1} \mathbf{R} \mathbf{Q}_i^{-1}$, and $\mathbf{S}_i := \frac{1}{\sqrt{T}\sigma_i} \mathbf{Z}_i^\top \boldsymbol{\varepsilon}_{i\cdot} = \frac{1}{\sqrt{T}\sigma_i} \sum_{t=1}^T \mathbf{z}_{i,t} \epsilon_{i,t}$. The two statistics $\widetilde{W}_i$ and $W_i^{(0)}$ can therefore be rewritten as

$$
W_i^{(0)} = \mathbf{S}_i^\top h(\mathbf{Q}_i) \mathbf{S}_i, \quad \widetilde{W}_i = \mathbf{S}_i^\top h(\widetilde{\mathbf{Q}}_i) \mathbf{S}_i
$$

and their difference becomes

$$
\left| W_i^{(0)} - \widetilde{W}_i \right| = \left| \mathbf{S}_i^\top \left( h(\mathbf{Q}_i) - h(\widetilde{\mathbf{Q}}_i) \right) \mathbf{S}_i \right| \leq \| h(\mathbf{Q}_i) - h(\widetilde{\mathbf{Q}}_i) \|_{op} \cdot \| \mathbf{S}_i \|_2^2 \tag{S.19}
$$

With Assumption (A3), for any $i \in [N]$,

$$
\| h(\mathbf{Q}_i) - h(\widetilde{\mathbf{Q}}_i) \|_{op} \leq C \| \mathbf{Q}_i^{-1} - \widetilde{\mathbf{Q}}_i^{-1} \|_{op} \leq C \| \mathbf{Q}_i^{-1} \|_{op} \cdot \| \mathbf{Q}_i - \widetilde{\mathbf{Q}}_i \|_{op} \cdot \| \widetilde{\mathbf{Q}}_i^{-1} \|_{op} \leq C \| \mathbf{Q}_i - \widetilde{\mathbf{Q}}_i \|_{op}.
$$

Along with (S.19), we have

$$
\max_{1 \leq i \leq N} |W_i^{(0)} - \widetilde{W}_i| \leq C \max_{1 \leq i \leq N} \| \mathbf{Q}_i - \widetilde{\mathbf{Q}}_i \|_{op} \cdot \max_{1 \leq i \leq N} \| \mathbf{S}_i \|_2^2. \tag{S.20}
$$

Furthermore, for any $\varepsilon > 0$ and any $\eta > 0$,

$$
\mathbb{P}\left( \max_{1 \leq i \leq N} |W_i^{(0)} - \widetilde{W}_i| > \varepsilon \right) \leq \mathbb{P}\left( \max_{1 \leq i \leq N} \| \mathbf{Q}_i - \widetilde{\mathbf{Q}}_i \|_{op} > \gamma \right) + \mathbb{P}\left( \max_{1 \leq i \leq N} \| \mathbf{S}_i \|_2^2 > \frac{\varepsilon}{C\eta} \right). \tag{S.21}
$$

We now focus on $\max_{1 \leq i \leq N} \| \mathbf{Q}_i - \widetilde{\mathbf{Q}}_i \|_{op}$. Recall that $\mathbf{Q}_i = \frac{1}{T} \mathbf{Z}_i^\top \mathbf{Z}_i \in \mathbb{R}^{2K \times 2K}$ and $\widetilde{\mathbf{Q}}_i = \mathbb{E}(\mathbf{z}_{i,t} \mathbf{z}_{i,t}^\top) \in \mathbb{R}^{2K \times 2K}$. With Assumptions 3.2 and 3.4, and using the matrix Bernstein inequality for sub-Gaussian sample covariance, there exists $C > 0$, such that

$$
\mathbb{P}\left( \| \mathbf{Q}_i - \widetilde{\mathbf{Q}}_i \|_{op} \geq C \left( \sqrt{\frac{2K + u}{T}} + \frac{2K + u}{T} \right) \right) \leq 2 e^{-u}.
$$

To obtain the union bound, let $u = \log N + t$,

$$
\begin{aligned}
&\mathbb{P}\left( \max_{1 \leq i \leq N} \| \mathbf{Q}_i - \widetilde{\mathbf{Q}}_i \|_{op} \geq C \left( \sqrt{\frac{2K + \log N + t}{T}} + \frac{2K + \log N + t}{T} \right) \right) \\
&\leq \sum_{i=1}^N \mathbb{P}\left( \| \mathbf{Q}_i - \widetilde{\mathbf{Q}}_i \|_{op} \geq C \left( \sqrt{\frac{2K + u}{T}} + \frac{2K + u}{T} \right) \right) \leq N 2 e^{-\log N + t} = 2 e^{-t}
\end{aligned} \tag{S.22}
$$

Next, we derive the concentration of $\| \mathbf{S}_i \|_2^2$. Recall that $\mathbf{S}_i := \frac{1}{\sqrt{T}\sigma_i} \mathbf{Z}_i^\top \boldsymbol{\varepsilon}_{i\cdot} = \frac{1}{\sqrt{T}\sigma_i} \sum_{t=1}^T \mathbf{z}_{i,t} \epsilon_{i,t}$. With Assumption 3.1 and Assumption 3.4, $\epsilon_{i,t}$ is sub-Gaussian with $\| \epsilon_{i,t} \|_{\psi_2} \leq K_\epsilon$ and $\mathbf{z}_{i,t}$ is sub-Gaussian with $\| \mathbf{z}_{i,t} \|_{\psi_2} \leq K_z$. Given the multiplicative nature of the elements in this statistic, we adopt a two-step approach by first studying a conditional tail bound given $\mathbf{Z}_i$, followed by an additional step to obtain the unconditional tail bound. By definition,

$$
\| \mathbf{S}_i \|_2 = \sup_{\boldsymbol{\nu} \in \mathcal{S}^{2K-1}} \boldsymbol{\nu}^\top \mathbf{S}_i
$$

where $\mathcal{S}^{2K-1} = \{ \boldsymbol{\nu} \in \mathbb{R}^{2K} : \| \boldsymbol{\nu} \|_2 = 1 \}$ is the unit sphere in $\mathbb{R}^{2K}$. For any individual unit vector $\boldsymbol{\nu} \in \mathcal{S}^{2K-1}$, $\boldsymbol{\nu}^\top \mathbf{S}_i = \frac{1}{\sqrt{T}\sigma_i} \sum_{t=1}^T \boldsymbol{\nu}^\top \mathbf{z}_{i,t} \epsilon_{i,t} := \frac{1}{\sqrt{T}} \sum_{t=1}^T \xi_{i,t}$ with $\xi_{i,t} := \sigma_i^{-1} \boldsymbol{\nu}^\top \mathbf{z}_{i,t} \epsilon_{i,t}$. Conditional on $\{ \mathbf{z}_{i,t} \}_{t=1}^T$, each $\xi_{i,t}$ is

sub-Gaussian with $\|\xi_{i,t}\|_{\psi_2|Z} \le \sigma_i^{-1}|\boldsymbol{\nu}^\top \mathbf{z}_{i,t}| \cdot \|\epsilon_{i,t}\|_{\psi_2} \le \sigma_i^{-1} K_\epsilon |\boldsymbol{\nu}^\top \mathbf{z}_{i,t}|$, $\mathbb{E}(\xi_{i,t} \mid \mathbf{z}_{i,t}) = 0$, and $\{\xi_{i,t}\}_{t=1}^T$ are independent across $t$ under conditioning. Hence, a conditional Bernstein-type bound yields, for any $u > 0$,

$$\mathbb{P}\left(\left|\sum_{t=1}^T \xi_{i,t}\right| \ge CK_\epsilon \sqrt{u}\left(\sum_{t=1}^T (\boldsymbol{\nu}^\top \mathbf{z}_{i,t})^2\right)^{\frac{1}{2}} \mid \mathbf{Z}_i\right) \le 2e^{-u}.$$

and equivalently,

$$\mathbb{P}\left(|\boldsymbol{\nu}^\top \mathbf{S}_i| \ge CK_\epsilon \frac{\sqrt{u}}{\sqrt{T}}\left(\sum_{t=1}^T (\boldsymbol{\nu}^\top \mathbf{z}_{i,t})^2\right)^{\frac{1}{2}} \mid \mathbf{Z}_i\right) \le 2e^{-u}.$$

Further note that $\sum_{t=1}^T (\boldsymbol{\nu}^\top \mathbf{z}_{i,t})^2 = \|\mathbf{Z}_i \boldsymbol{\nu}\|_2^2 \le \|\mathbf{Z}_i\|_{op}^2 \cdot \|\boldsymbol{\nu}\|_2^2 = \|\mathbf{Z}_i\|_{op}^2$, then the above inequality becomes

$$\mathbb{P}\left(|\boldsymbol{\nu}^\top \mathbf{S}_i| \ge CK_\epsilon \|\mathbf{Z}_i\|_{op} \frac{\sqrt{u}}{\sqrt{T}} \mid \mathbf{Z}_i\right) \le 2e^{-u}. \tag{S.23}$$

Let $\mathcal{N}$ be a $\frac{1}{2}$-net of $\mathcal{S}^{2K-1}$ in Euclidean norm with cardinality $|\mathcal{N}| \le 5^{2K}$. By definition, $\|\mathbf{S}_i\|_2 = \sup_{\boldsymbol{\nu} \in \mathcal{S}^{2K-1}} \boldsymbol{\nu}^\top \mathbf{S}_i \le 2\max_{\boldsymbol{\nu} \in \mathcal{N}} \boldsymbol{\nu}^\top \mathbf{S}_i$. Applying the bound of each $\nu$ and considering the union bound, conditional on $\mathbf{Z}_i$,

$$\mathbb{P}\left(\|\mathbf{S}\|_i \ge 2CK_\epsilon \|\mathbf{Z}_i\|_{op} \frac{\sqrt{u}}{\sqrt{T}} \mid \mathbf{Z}_i\right) \le \mathbb{P}\left(\max_{\boldsymbol{\nu} \in \mathcal{N}} |\boldsymbol{\nu}^\top \mathbf{S}_i| \ge CK_\epsilon \|\mathbf{Z}_i\|_{op} \frac{\sqrt{u}}{\sqrt{T}} \mid \mathbf{Z}_i\right) \le |\mathcal{N}| \cdot 2e^{-u}.$$

Let $u = \log|\mathcal{N}| + t$, and note that $\log|\mathcal{N}| \asymp 2K$, the above becomes

$$\mathbb{P}\left(\|\mathbf{S}\|_i \ge 2CK_\epsilon \|\mathbf{Z}_i\|_{op} \frac{\sqrt{2K+t}}{\sqrt{T}} \mid \mathbf{Z}_i\right) \le 2e^{-t}. \tag{S.24}$$

In addition, by the sub-Gaussianity of $\mathbf{z}_{i,t}$, for any $u > 0$,

$$\mathbb{P}\left(\|\mathbf{Z}_i\|_{op} \ge CK_z(\sqrt{T} + \sqrt{2K} + \sqrt{u})\right) \le 2e^{-u}. \tag{S.25}$$

We then define two events,

$$A_{i,u} := \left\{\|\mathbf{Z}_i\|_{op} \le CK_z(\sqrt{T} + \sqrt{2K} + \sqrt{u})\right\}$$

$$B_{i,u} := \left\{\|\mathbf{S}_i\|_2 \le 2CK_\epsilon \|\mathbf{Z}_i\|_{op} \frac{\sqrt{2K+u}}{\sqrt{T}}\right\}$$

Previous results show that $\mathbb{P}(A_{i,u}^c) \le 2e^{-u}$, $\mathbb{P}(B_{i,u}^c) \mid \mathbf{Z}_i) \le 2e^{-u}$, and hence $\mathbb{P}(B_{i,u}^c) \le 2e^{-u}$ by taking expectations. On the event $A_{i,u} \cap B_{i,u}$,

$$\|\mathbf{S}_i\| \le CK_\epsilon K_z(\sqrt{T} + \sqrt{2K} + \sqrt{u}) \cdot \frac{\sqrt{2K+u}}{\sqrt{T}} \lesssim \frac{\sqrt{T} \cdot \sqrt{2K+u} + (2K+u)}{\sqrt{T}}$$

To get a cleaner form, with $2K + u \lesssim T$, we only keep the leading term, yielding

$$\mathbb{P}\left(\|\mathbf{S}_i\|_2 \ge CK_z K_\epsilon \cdot \frac{\sqrt{T \cdot 2K + T \cdot u}}{\sqrt{T}}\right) \le 4e^{-u}$$

Then, the squaring brings

$$\mathbb{P}\left(\|\mathbf{S}_i\|_2^2 \ge C\left(2K+u\right)\right) \le 4e^{-u}.$$

Let $u = \log N + t$, and we obtain the union bound

$$\mathbb{P}\left(\max_{1 \le i \le N} \|\mathbf{S}_i\|_2^2 \ge C\left(2K + \log N + t\right)\right) \le N \cdot 4e^{-\log N + t} = 4e^{-t}. \tag{S.26}$$

With (S.22) and (S.26), we have $\max_{1 \le i \le N} \|\mathbf{Q}_i - \widetilde{\mathbf{Q}}_i\|_{op} = O_p(\sqrt{\frac{\log N}{T}})$ and $\mathbf{S}_i\|_2^2 = O_p(\log N)$. Then by (S.20), $\max_{1 \le i \le N} |W_i^{(0)} - \widetilde{W}_i| = O_p(\frac{\log^{3/2} N}{\sqrt{T}})$. If we pick $t = 2 \log N$, (S.22) becomes

$$\mathbb{P}\left(\max_{1 \le i \le N} \|\mathbf{Q}_i - \widetilde{\mathbf{Q}}_i\|_{op} \ge C\sqrt{\frac{\log N}{T}}\right) \le 2e^{-2\log N} = 2N^{-2},$$

and (S.26) becomes

$$\mathbb{P}\left(\max_{1 \le i \le N} \|\mathbf{S}_i\|_2^2 \ge C \log N\right) \le 4e^{-2\log N} = 4N^{-2}.$$

With these results, we choose $\varepsilon \asymp \frac{\log^{3/2} N}{\sqrt{T}}$ and $\eta \asymp \frac{\sqrt{\log N}}{\sqrt{T}}$ in (S.21), and hence $\frac{\varepsilon}{\eta} \asymp \log N$. The inequality becomes

$$\mathbb{P}\left(\max_{1 \le i \le N} |W_i^{(0)} - \widetilde{W}_i| > \frac{C\log^{3/2} N}{\sqrt{T}}\right) \tag{S.27}$$
$$\le \mathbb{P}\left(\max_{1 \le i \le N} \|\mathbf{Q}_i - \widetilde{\mathbf{Q}}_i\|_{op} > C\frac{\sqrt{\log N}}{\sqrt{T}}\right) + \mathbb{P}\left(\max_{1 \le i \le N} \|\mathbf{S}_i\|_2^2 > C\log N\right) \lesssim N^{-2}$$

In addition, as proven in Lemma A.1, $\sup_x \mathbb{P}\left(|M_{\chi_K^2} - x| \le \varepsilon\right) \lesssim \varepsilon \lesssim \frac{\log^{3/2} N}{\sqrt{T}}$, and (S.16) implies $d(M_{\chi_K^2}, \widetilde{M}_{Wald}) \lesssim \left(\frac{\log^7(NT)}{T}\right)^{1/6}$. Then, by (S.18),

$$d(M_{Wald}^{(0)}, \widetilde{M}_{Wald}) \lesssim \frac{(\log N)^{3/2}}{\sqrt{T}} + \left(\frac{\log^7(NT)}{T}\right)^{1/6} + \frac{1}{N^2} \tag{S.28}$$

This completes the proof of (S.17) in Lemma A.3. $\qquad\square$

**Lemma A.4** (Non-asymptotic error bound on the random design linearization error)**.** *Under the same assumptions as in Proposition 3.5,*

$$d(M_{Wald}, M_{Wald}^{(0)}) \lesssim \frac{(\log N)^{7/4}}{T^{5/4}} + \frac{1}{N^2} + \left(\frac{\log^7(NT)}{T}\right)^{1/6} + \frac{(\log N)^{3/2}}{\sqrt{T}} \tag{S.29}$$

*Proof of Lemma A.4.* Revisiting (S.12) and (S.13), by construction,

$$W_i = \frac{\sigma_i^2}{\hat{\sigma}_i^2} W_i^{(0)}$$

where $\sigma_i^2 = \mathbb{E}(\epsilon_{it}^2)$ and $\hat{\sigma}_i^2 = \hat{\varepsilon}_{i\cdot}^\top \hat{\varepsilon}_{i\cdot}/(T - 2K - 1)$, where $\hat{\varepsilon}_{i\cdot} = \mathbf{y}_{i\cdot} - \hat{\alpha}_i \mathbf{1}_T - \mathbf{Z}_i^\top \hat{\boldsymbol{\theta}}_i$. With Assumption (A1), we know, for each $i \in [N]$,

$$\frac{\sigma_i^2}{\hat{\sigma}_i^2} - 1 = O_p\left(\frac{1}{\sqrt{T - 2K - 1}}\right).$$

Throughout our analysis, we consider $K$ to be a fixed and small number that satisfies $K = o(T)$, and thus, we simplifies $O_p((T - 2K - 1)^{-1/2})$ to $O_p(T^{-1/2})$. To translate the scaling error into the tail probability gap for the maxima, we consider a small positive error term $\delta_T \in (0, \frac{1}{2})$, and define an event $\mathcal{E}_\delta := \left\{\max_{1 \le i \le N} \left|\frac{\sigma_i^2}{\hat{\sigma}_i^2} - 1\right| \le \delta\right\}$. On the event $\mathcal{E}_\delta$, we have

$$(1 - \delta)M_{Wald}^{(0)} \le M_{Wald} \le (1 + \delta)M_{Wald}^{(0)}.$$

Therefore, we can show

$$\mathbb{P}(M_{Wald} \le x) \le \mathbb{P}\left(M_{Wald}^{(0)} \le \frac{x}{1 - \delta}\right) + \mathbb{P}\left(\max_{1 \le i \le N} \left|\frac{\sigma_i^2}{\hat{\sigma}_i^2} - 1\right| > \delta\right),$$

and

$$\mathbb{P}\left(M_{Wald} \leq x\right) \geq \mathbb{P}\left(M_{Wald}^{(0)} \leq \frac{x}{1+\delta}\right) - \mathbb{P}\left(\max_{1\leq i\leq N}\left|\frac{\sigma_i^2}{\hat{\sigma}_i^2} - 1\right| > \delta\right).$$

With the above two equations,

$$d(M_{Wald}, M_{Wald}^{(0)}) = \sup_x \left|\mathbb{P}(M_{Wald} \leq x) - \mathbb{P}(M_{Wald}^{(0)} \leq x)\right|$$

$$\leq \max\left\{\sup_x \left(\mathbb{P}(M_{Wald} \leq x) - \mathbb{P}(M_{Wald}^{(0)} \leq x)\right), \sup_x \left(\mathbb{P}(M_{Wald}^{(0)} \leq x) - \mathbb{P}(M_{Wald} \leq x)\right)\right\}$$

$$\leq \sup_x \mathbb{P}\left(\frac{x}{1+\delta} < M_{Wald}^{(0)} \leq \frac{x}{1-\delta}\right) + \mathbb{P}\left(\max_{1\leq i\leq N}\left|\frac{\sigma_i^2}{\hat{\sigma}_i^2} - 1\right| > \delta\right). \tag{S.30}$$

With Assumption (A1), $\epsilon_{i,t}$ is sub-Gaussian. By the Hanson-Wright concentration inequality for quadratic forms (Theorem 1.1, Rudelson & Vershynin, 2013), there exists $c > 0$, such that for any $t > 0$,

$$\mathbb{P}\left(\left|\frac{\sigma_i^2}{\hat{\sigma}_i^2} - 1\right| > t\right) \leq 2\exp\left(-c(T - 2K - 1)\min\{t^2, t\}\right) \tag{S.31}$$

Here, we consider a small $\delta \in (0, \frac{1}{2})$, leading to $\min\{\delta^2, \delta\} = \delta$. Furthermore, let $\delta = c_\delta\sqrt{\frac{\log N}{T-2K-1}} \asymp \sqrt{\frac{\log N}{T}}$, where $c_\delta$ satisfies $c_\delta > \sqrt{3c^{-1}}$. Hence,

$$\mathbb{P}\left(\max_{1\leq i\leq N}\left|\frac{\sigma_i^2}{\hat{\sigma}_i^2} - 1\right| > \delta\right) \leq N \cdot 2\exp\left(-c \cdot c_\delta^2 \log N\right) = 2N^{-(c\cdot c_\delta^2 - 1)} \leq 2N^{-2}. \tag{S.32}$$

To quantify $\sup_x \mathbb{P}\left(\frac{x}{1+\delta} < M_{Wald}^{(0)} \leq \frac{x}{1-\delta}\right)$, we consider the triangle inequality

$$\sup_x \mathbb{P}\left(\frac{x}{1+\delta} < M_{Wald}^{(0)} \leq \frac{x}{1-\delta}\right)$$

$$\leq \sup_x \mathbb{P}\left(\frac{x}{1+\delta} < M_{\chi^2} \leq \frac{x}{1-\delta}\right) + 2d(M_{Wald}^{(0)}, \widetilde{M}_{Wald}) + 2d(\widetilde{M}_{Wald}, M_{\chi^2}) \tag{S.33}$$

in which $d(M_{Wald}^{(0)}, \widetilde{M}_{Wald})$ has been established in (S.17), and $d(\widetilde{M}_{Wald}, M_{\chi^2})$ has been established in (S.16). It suffices to bound $\sup_x \mathbb{P}\left(\frac{x}{1+\delta} < M_{\chi^2} \leq \frac{x}{1-\delta}\right)$.

To obtain the anti-concentration property of $M_{\chi^2}$, recall $M_{\chi_K^2} = \max_{1\leq i\leq N} U_i$ where $U_1, \ldots, U_N \overset{i.i.d.}{\sim} \chi_K^2$. Rewriting $U_i$ into $U_i = \|\mathbf{V}_i^G\|_2^2$, where $\mathbf{V}_i^G \sim \mathcal{N}(\mathbf{0}_K, \mathbf{I}_K)$. Then $M_{\chi_K^2} = \max_{1\leq i\leq N} U_i = \max_{1\leq i\leq N} \|\mathbf{V}_i^G\|_2^2 = \left(\max_{1\leq i\leq N} \|\mathbf{V}_i^G\|_2\right)$. Let $T_G = \max_{1\leq i\leq N} \|\mathbf{V}_i^G\|_2$, using the identity $\|\mathbf{V}_i^G\|_2 = \sup_{\boldsymbol{\nu}\in\mathcal{S}^{K-1}} \boldsymbol{\nu}^\top \mathbf{V}_i^G$, then

$$T_G = \max_{1\leq i\leq N} \sup_{\boldsymbol{\nu}\in\mathcal{S}^{K-1}} \boldsymbol{\nu}^\top \mathbf{V}_i^G.$$

We approximate the sphere $\mathcal{S}^{K-1}$ by an $\tau$-net $\mathcal{N}_\tau$ with cardinality $\mathcal{N}_\tau \leq (1 - 2\tau^{-1})^K$, then $\max_{1\leq i\leq N} \max_{\boldsymbol{\nu}\in\mathcal{N}_\tau} \boldsymbol{\nu}^\top \mathbf{V}_i^G$ approximates $T_G$ up to an additive error of the order $\tau \max_i \|\mathbf{V}_i^G\|_2$. With this, applying the Gaussian-max anti-concentration inequality (Theorem 3, Chernozhukov et al., 2015) to the discretized $\max_{1\leq i\leq N} \max_{\boldsymbol{\nu}\in\mathcal{N}_\tau} \boldsymbol{\nu}^\top \mathbf{V}_i^G$ gives, for any $t > 0$,

$$\mathbb{P}\left(|T_G - t| \leq \Delta\right) \leq C\Delta\sqrt{\log(N \cdot |\mathcal{N}_\tau|)} \lesssim \Delta\sqrt{\log N}$$

For $x > 0$ and $0 < \Delta < x$,

$$\mathbb{P}\left(|M_{\chi^2} - x| \leq \Delta\right) = \mathbb{P}\left(|T_G - \sqrt{x}| \leq \sqrt{x+\Delta} - \sqrt{x-\Delta}\right)$$

$$\leq \mathbb{P}\left(|T_G - \sqrt{x}| \leq \frac{\Delta}{\sqrt{x}}\right) \leq \frac{C\Delta}{\sqrt{x}}\sqrt{\log N}.$$

If we take $\Delta = x\delta$, the above becomes

$$\mathbb{P}\left(|M_{\chi^2} - x| \le x\delta\right) \le C\sqrt{x}\delta\sqrt{\log N}. \tag{S.34}$$

It is worth mentioning that the above inequality is not a uniform bound. When taking supremum over $x$, the right-hand-side of (S.34) tends to infinity as $x \to \infty$. To obtain a uniform bound of $\sup_x \mathbb{P}\left(\frac{x}{1+\delta} < M_{\chi^2} \le \frac{x}{1-\delta}\right)$ over $x$, we split the domain using a deterministic truncation bound $B$.

For the bounded region $\{0 < x \le B\}$, note that for $\delta \in (0, \frac{1}{2})$,

$$
\begin{aligned}
\sup_{0 < x \le B} \mathbb{P}\left(\frac{x}{1+\delta} < M_{\chi^2} \le \frac{x}{1-\delta}\right) &\le \sup_{0 < x \le B} \mathbb{P}\left(|M_{\chi^2} - x| \le 2\delta x\right) \\
&\le \sup_{0 < x \le B} C\delta\sqrt{x}\sqrt{\log N} \lesssim \delta\sqrt{B}\sqrt{\log N},
\end{aligned} \tag{S.35}
$$

in which the inequality of the 2nd line is a result of (S.34).

For the tail region $\{x > B\}$,

$$\sup_{x > B} \mathbb{P}\left(\frac{x}{1+\delta} < M_{\chi^2} \le \frac{x}{1-\delta}\right) \le \sup_{x > B} \mathbb{P}\left(M_{\chi^2} \ge \frac{x}{1+\delta}\right) \le \mathbb{P}\left(M_{\chi^2} \ge \frac{B}{1+\delta}\right) \tag{S.36}$$

For some positive constant $\gamma > 0$, we define

$$B := (1+\delta)[K + 2\sqrt{3K \log N} + 6 \log N].$$

Asymptotically, $B \asymp \delta \log N$. Consistent with previous analysis, we consider $\delta \asymp \sqrt{\frac{\log N}{T}}$. With this choice of $B$, the rate in (S.35) becomes

$$\sup_{0 < x \le B} \mathbb{P}\left(\frac{x}{1+\delta} < M_{\chi^2} \le \frac{x}{1-\delta}\right) \lesssim \delta\sqrt{B}\sqrt{\log N} \asymp \frac{(\log N)^{7/4}}{T^{5/4}}. \tag{S.37}$$

The inequality (S.36) becomes

$$
\begin{aligned}
\sup_{x > B} \mathbb{P}\left(\frac{x}{1+\delta} < M_{\chi^2} \le \frac{x}{1-\delta}\right) &\le \mathbb{P}\left(M_{\chi^2} \ge \frac{B}{1+\delta}\right) \\
&= \mathbb{P}\left(M_{\chi^2} \ge K + 2\sqrt{3K \log N} + 6 \log N\right) \le Ne^{-3\log N} = N^{-2}
\end{aligned} \tag{S.38}
$$

With (S.37) and (S.38),

$$
\begin{aligned}
&\sup_x \mathbb{P}\left(\frac{x}{1+\delta} < M_{\chi^2} \le \frac{x}{1-\delta}\right) \\
&= \max\left\{\sup_{0 < x \le B} \mathbb{P}\left(\frac{x}{1+\delta} < M_{\chi^2} \le \frac{x}{1-\delta}\right), \sup_{x > B} \mathbb{P}\left(\frac{x}{1+\delta} < M_{\chi^2} \le \frac{x}{1-\delta}\right)\right\} \\
&\lesssim \max\left\{\frac{(\log N)^{7/4}}{T^{5/4}}, N^{-2}\right\} \lesssim \frac{(\log N)^{7/4}}{T^{5/4}} + N^{-2}.
\end{aligned} \tag{S.39}
$$

With (S.32), (S.33), (S.39), (S.30) becomes

$$d(M_{Wald}, M_{Wald}^{(0)}) \lesssim \frac{(\log N)^{7/4}}{T^{5/4}} + \frac{1}{N^2} + \left(\frac{\log^7(NT)}{T}\right)^{1/6} + \frac{(\log N)^{3/2}}{\sqrt{T}} \tag{S.40}$$

This completes the proof of (S.29) in Lemma A.4. $\qquad\square$

### A.3. Proof of Proposition 3.7

*Proof.* Let $U \sim \chi_K^2$ be any random variable following a $\chi_K^2$ distribution, and note that, by construction, we have $c(T) \geq 1.5$. Therefore,

$$
\begin{aligned}
\mathbb{P}(U \geq \delta_{N,T}) &= \mathbb{P}(U \geq (K + 2\sqrt{K \log N} + 2\log N) \cdot c(T)) \\
&\leq \mathbb{P}(U \geq K + 2\sqrt{K \log N \cdot c(T)} + 2\log N \cdot c(T))
\end{aligned}
\tag{S.41}
$$

By Lemma 1 of Laurent & Massart (2000),

$$
\text{(S.41)} \leq \exp(-\log(N \cdot c(T)) = N^{-c(T)}.
\tag{S.42}
$$

For $U_1, \ldots, U_N \sim \chi_K^2$, by the union bound,

$$
\mathbb{P}(\max_{1 \leq i \leq N} U_i \geq \delta_{N,T}) \leq N \cdot \max_{1 \leq i \leq N} \mathbb{P}(U_i \geq \delta_{N,T}) \leq N^{1-c(T)}.
$$

As a result,

$$
\begin{aligned}
&\mathbb{P}(\max_{1 \leq i \leq N} W_i \geq \delta_{N,T}) \\
&\leq \mathbb{P}(\max_{1 \leq i \leq N} U_i \geq \delta_{N,T}) + |\mathbb{P}(\max_{1 \leq i \leq N} W_i \geq \delta_{N,T}) - \mathbb{P}(\max_{1 \leq i \leq N} U_i \geq \delta_{N,T})| \\
&\leq N^{1-c(T)} + \sup_{x>0} \left| \mathbb{P}\left(\max_{1 \leq i \leq N} W_i \geq x\right) - \mathbb{P}\left(\max_{1 \leq i \leq N} \chi_{K,i}^2 \geq x\right) \right|
\end{aligned}
\tag{S.43}
$$

Proposition 3.5 shows that, under the null hypothesis, we have

$$
d(\max_{1 \leq i \leq N} W_i, \max_{1 \leq i \leq N} \chi_{K,i}^2) \lesssim \left( \frac{\log^7(NT)}{T} \right)^{1/6} + \frac{(\log N)^{3/2}}{\sqrt{T}} + \frac{(\log N)^{7/4}}{T^{5/4}} + \frac{1}{N^2}.
\tag{S.44}
$$

By construction, $N^{1-c(T)} \lesssim N^{-0.5}$. (S.43) becomes,

$$
\mathbb{P}(\max_{1 \leq i \leq N} W_i \geq \delta_{N,T}) \leq \frac{1}{\sqrt{N}} + \left( \frac{\log^7(NT)}{T} \right)^{1/6} + \frac{(\log N)^{3/2}}{\sqrt{T}} + \frac{(\log N)^{7/4}}{T^{5/4}}
\tag{S.45}
$$

This completes the proof of Proposition 3.7. □

### A.4. Proofs of Theorem 3.8

*Proof.* Property (i). By construction, $J_{PE} = \sqrt{N} \cdot M_{Wald} \cdot I\{M_{Wald} > \delta_{N,T}\} \geq 0$, and therefore, property (i) naturally holds.

Property (ii). By Proposition 3.7, under the null hypothesis,

$$
\mathbb{P}\left( \max_{1 \leq i \leq N} W_i \geq \delta_{N,T} \right) \leq \frac{1}{\sqrt{N}} + \left( \frac{\log^7(NT)}{T} \right)^{1/6} + \frac{(\log N)^{3/2}}{\sqrt{T}} + \frac{(\log N)^{7/4}}{T^{5/4}}
\tag{S.46}
$$

Suppose $\log N = o(T^{1/7})$, then

$$
\begin{aligned}
\frac{\log^7(NT)}{T} &= \frac{\log^7 N}{T} + \frac{\log^7 T}{T} = o(1) \\
\frac{(\log N)^{3/2}}{\sqrt{T}} &= \left( \frac{\log^7 N}{T} \right)^{1/2} \cdot \frac{1}{\log^2 N} = o(1) \\
\frac{(\log N)^{7/4}}{T^{5/4}} &= \left( \frac{\log^7 N}{T} \right)^{1/4} \cdot \frac{1}{T} = o(1).
\end{aligned}
$$

Hence, as $N, T \to \infty$,

$$\mathbb{P}(J_{PE} = 0 \mid H_0) = \mathbb{P}\left(\max_{1 \leq i \leq N} W_i \geq \delta_{N,T} \mid H_0\right) \to 0.$$

Property (iii). With Assumptions 3.1 – 3.3, under the null hypothesis $H_{0i} : \beta_i = \mathbf{0}$, $W_i \xrightarrow{d} \chi_K^2$ as $T \to \infty$. Under the alternative hypothesis $H_{ai} : \beta_i \neq \mathbf{0}$, $W_i$ converges in distribution to a non-central $\chi^2$ distribution $\chi_K^2(\lambda_i)$ with the non-centrality parameter $\lambda_i = T\beta_i^\top \Sigma_i^{-1} \beta_i$, where $\Sigma_i = T\sigma_i^2 \mathbf{R}^\top (\mathbf{Z}_i^\top \mathbf{Z}_i)^{-1} \mathbf{R}$, yielding

$$\mathbb{P}_{\beta_i} (W_i \geq \delta_{N,T}) = \mathbb{P}\left(\chi_K^2(\lambda_i) \geq \delta_{N,T}\right) + o(1).$$

To analyze the asymptotic power, we consider a widely used normal approximation for non-central $\chi_K^2(\lambda_i)$: $\chi_K^2(\lambda_i) \approx \mathcal{N}(K + \lambda_i, 2(K + 2\lambda_i))$, and with Berry-Esseen-type bound, we have

$$\mathbb{P}\left(\chi_K^2(\lambda_i) \geq \delta_{N,T}\right) = 1 - \Phi\left(\frac{\delta_{N,T} - (K + \lambda_i)}{\sqrt{2(K + 2\lambda_i)}}\right) + O(\lambda_i^{-\frac{1}{2}}) \tag{S.47}$$

For any $\beta \in \Theta_\beta = \left\{(\beta_1, \ldots, \beta_N) : \max_{1 \leq i \leq N} \beta_i^\top \Sigma_i^{-1} \beta_i > \frac{\eta \cdot \log \log T \cdot \log N}{T} \text{ for some } \eta > 2\right\}$, there exists $i_0 \in [N]$, such that $\lambda_{i_0} = T\beta_{i_0}^\top \Sigma_{i_0}^{-1} \beta_{i_0} > \eta \cdot \log \log T \cdot \log N$ for some $\eta > 2$. As a result,

$$\frac{\delta_{N,T} - (K + \lambda_i)}{\sqrt{2(K + 2\lambda_i)}} \leq \frac{(K + 2\sqrt{K \log N + 2\log N}) \cdot c(T) - (K + \eta \log N \log \log T)}{\sqrt{2(K + 2\eta \log N \log \log T)}}$$

$$= \frac{K(c(T) - 1) + 2c(T)\sqrt{K \log N} + (2c(T) - \eta \log \log T) \log N}{\sqrt{2(K + 2\eta \log N \log \log T)}}$$

$$\asymp \frac{(2c(T) - \eta \log \log T) \log N}{\sqrt{\log N \log \log T}} \to -\infty \quad \text{as } N, T \to \infty$$

and therefore,

$$\Phi\left(\frac{\delta_{N,T} - (K + \lambda_i)}{\sqrt{2(K + 2\lambda_i)}}\right) \to 0 \quad \text{as } N, T \to \infty.$$

Then,

$$\mathbb{P}_{\beta_i} (W_{i_0} \geq \delta_{N,T}) = \mathbb{P}\left(\chi_K^2(\lambda_i) \geq \delta_{N,T}\right) + o(1) = 1 - \Phi\left(\frac{\delta_{N,T} - (K + \lambda_i)}{\sqrt{2(K + 2\lambda_i)}}\right) + o(1),$$

and hence,

$$\mathbb{P}_{\beta_i} (W_{i_0} \geq \delta_{N,T}) \to 1 \quad \text{as } N, T \to \infty.$$

For any $\beta \in \Theta_\beta$,

$$\mathbb{P}_\beta(J_{PE} \geq \sqrt{T}\delta_{N,T}) = \mathbb{P}_\beta (M_{Wald} > \delta_{N,T}) \geq \mathbb{P}_{\beta_i} (W_{i_0} \geq \delta_{N,T}) \to 1 \quad \text{as } N, T \to \infty.$$

$$\square$$

## A.5. Proofs of Theorem 3.9

*Proof.* By definition, $T_{PE-PGCT} = T_{pivotal} + J_{PE}$. Then, for any $x$,

$$\begin{aligned}
&\mathbb{P}_{H_0}(T_{pivotal} \leq x) \\
=&\mathbb{P}_{H_0}(\{T_{pivotal} \leq x\} \cap \{J_{PE} = 0\}) + \mathbb{P}_{H_0}(\{T_{pivotal} \leq x\} \cap \{J_{PE} \neq 0\}) \\
\leq&\mathbb{P}_{H_0}(\{T_{pivotal} + J_{PE} \leq x\} \cap \{J_{PE} = 0\}) + \mathbb{P}_{H_0}(J_{PE} \neq 0) \\
\leq&\mathbb{P}_{H_0}(T_{PE-PGCT} \leq x) + \mathbb{P}_{H_0}(J_{PE} \neq 0).
\end{aligned} \tag{S.48}$$

On the other hand, the non-negativity of $J_{PE}$ ensures

$$\mathbb{P}_{H_0}(T_{PE-PGCT} \leq x) \leq \mathbb{P}_{H_0}(T_{pivotal} \leq x).$$

Therefore, for any $x$,

$$0 \leq \mathbb{P}_{H_0}(T_{pivotal} \leq x) - \mathbb{P}_{H_0}(T_{PE-PGCT} \leq x) \leq \mathbb{P}_{H_0}(J_{PE} \neq 0).$$

Property (ii) of Theorem 3.8 suggests, $\mathbb{P}_{H_0}(J_{PE} \neq 0) \rightarrow 0$ as $N, T \rightarrow \infty$, and hence, for any $x$,

$$\mathbb{P}_{H_0}(T_{PE-PGCT} \leq x) - \mathbb{P}_{H_0}(T_{pivotal} \leq x) \rightarrow 0,$$

i.e., under $H_0$,

$$T_{PE-PGCT} \overset{d}{=} T_{pivotal}.$$

$\square$

### A.6. Proofs of Theorem 3.10

*Proof.* Property (i). By Theorem 3.9, $T_{PE-PGCT} \overset{d}{=} T_{pivotal}$ under $H_0$. Therefore, for any $\boldsymbol{\beta} \in \Theta_0 = \{(\boldsymbol{\beta}_1, \ldots, \boldsymbol{\beta}_N) : \boldsymbol{\beta}_i = \mathbf{0}, \forall i = 1, \ldots, N\}$, as $N, T \rightarrow \infty$,

$$\phi_{PE-PGCT}(\boldsymbol{\beta} = \mathbf{0}) = \mathbb{P}_{H_0}(T_{PE-PGCT} \geq q_{pivotal}^{\alpha}) \rightarrow \mathbb{P}_{H_0}(T_{pivotal} \geq q_{pivotal}^{\alpha}) = \alpha.$$

Property (ii). The non-negativity of $J_{PE}$ ensures that $\mathbb{P}(T_{PE-PGCT} \geq x) \leq \mathbb{P}(T_{pivotal} \geq x)$. Therefore, for any $\boldsymbol{\beta} \in \Theta$,

$$\phi_{PE-PGCT}(\boldsymbol{\beta}) = \mathbb{P}_{\boldsymbol{\beta}}(T_{PE-PGCT} \geq q_{pivotal}^{\alpha}) \geq \mathbb{P}_{\boldsymbol{\beta}}(T_{pivotal} \geq q_{pivotal}^{\alpha}) = \phi_{pivotal}(\boldsymbol{\beta}).$$

Property (iii). This property is a natural result of Property (iii) in Theorem 3.8. Specifically, Theorem 3.8 shows that $\forall \boldsymbol{\beta} \in \Theta_\beta$, $J_{PE} \rightarrow \infty$ as $N, T \rightarrow \infty$. Hence, for $\forall \boldsymbol{\beta} \in \Theta_\beta$, with the fixed critical value $q_{pivotal}^{\alpha}$, as $N, T \rightarrow \infty$,

$$\phi_{PE-PGCT}(\boldsymbol{\beta}) = \mathbb{P}_{\boldsymbol{\beta}}(T_{pivotal} + J_{PE} \geq q_{pivotal}^{\alpha}) \rightarrow 1.$$

$\square$

## B. Supplemental Details of the Pivotal Test $T_{pivotal}$

As detailed in Remark 2.2 of the main context, we advocate using a data-adaptive strategy and recommend different pivotal tests depending on the dimensionality of the panel. This data-adaptive choice leverages existing methods in the literature within their respective regimes of strength, thereby ensuring robustness and validity of the proposed procedure across a broad range of panel settings. Specifically, we propose using

$$T_{pivotal} = \begin{cases} |T_{DH}|, & \text{if } N < T - K \\ T_{HPJ}, & \text{if } N \geq T - K \end{cases}$$

where the first regime corresponds to the traditional panel setting and the second corresponds to the high-dimensional panel regime. The above choice is proposed after carefully weighing the relative advantages and limitations of available approaches in the literature. As empirically demonstrated in Figure 1, DH performs well when $T$ is large but suffers from severe size distortions when $N$ is large, particularly when $N > T$. In contrast, HPJ performs well for large $N$ but exhibits distorted Type I error control when $N$ is small. These differences are rooted in the distinct asymptotic regimes under which the two tests are valid. For completion, in what follows, we present the mathematical details of the two existing PGCT tests.

### B.1. The Dumitrescu-Hurlin (DH) Test

The DH test (Dumitrescu & Hurlin, 2012) begins with the individual Wald statistics $W_i$ for testing $H_{0i} : \boldsymbol{\beta}_i = (\beta_{i1}, \ldots, \beta_{iK})^\top = \mathbf{0}$ in the linear panel model (1), for each $i \in [N]$. The explicit form of $W_i$ is given in (S.12). To conduct the Granger-causality test, the DH test considers the standardized average of individual Wald statistics, defined by

$$Z_{DH} = \sqrt{\frac{N}{2K}} \left( \frac{1}{N} \sum_{i=1}^{N} W_i - K \right).$$

Theorem 1 of Dumitrescu & Hurlin (2012) proves that under some regularity conditions, $Z_{DH} \xrightarrow{d} N(0,1)$ in a sequential asymptotic regime when $T \to \infty$ first, followed by $N \to \infty$. To improve the finite-sample performance, Dumitrescu & Hurlin (2012) also introduces a fixed-$T$ version, for any fixed $T > 5 + 2K$,

$$T_{DH} = \sqrt{\frac{p}{2K} \cdot \frac{T - 2K - 5}{T - K - 3}} \left( \frac{T - 2K - 3}{T - 2K - 1} \cdot \frac{1}{N} \sum_{i=1}^{N} W_i - K \right).$$

Proposition 3 of Dumitrescu & Hurlin (2012) proves that under some regularity conditions, $T_{DH} \xrightarrow{d} N(0,1)$ as $N \to \infty$. According to the analysis of limiting null distributions, the DH tests rejects $H_0$ at the significance level $\alpha$ if $|Z_{DH}| \geq z_{\alpha/2}$, or if using the fixed-$T$ version for better finite-sample performance, the DH tests rejects $H_0$ at the significance level $\alpha$ if $|T_{DH}| \geq z_{\alpha/2}$, where $z_{\alpha/2}$ is the upper-$\alpha/2$ quantile of the standard normal distribution.

### B.2. The Half-Panel Jackknife (HPJ) Test

The HPJ test (Juodis et al., 2021) employs a pooled least-squares estimator, that relies on a key insight that under the null hypothesis of non-causality, the Granger-causal coefficients are identically zero for all units, rendering the parameter of interest homogeneous under the null hypothesis. Assuming homogeneity in $\boldsymbol{\beta}_i$, the model (1) becomes

$$\mathbf{y}_{i\cdot} = \boldsymbol{\Xi}_i^\top \begin{pmatrix} \alpha_i \\ \gamma_i \end{pmatrix} + \mathbf{X}_i^\top \boldsymbol{\beta} + \boldsymbol{\varepsilon}_{i\cdot},$$

where $\mathbf{y}_{i\cdot} = (y_{i,1}, \dots, y_{i,T})^\top \in \mathbb{R}^T$, $\mathbf{X}_i = (\mathbf{x}_{i,1}, \dots, \mathbf{x}_{i,T})^\top \in \mathbb{R}^{T \times K}$, $\boldsymbol{\Xi}_i = (\boldsymbol{\xi}_{i,1}, \dots, \boldsymbol{\xi}_{i,T})^\top \in \mathbb{R}^{T \times (K+1)}$ with $\boldsymbol{\xi}_{i,t} = (1, y_{i,t-1}, \dots, y_{i,t-K})^\top \in \mathbb{R}^{K+1}$. The least-square estimator of $\boldsymbol{\beta}$ can be obtained by $\widehat{\boldsymbol{\beta}}_0 = (\sum_{i=1}^{N} \mathbf{X}_i^\top \mathbf{M}_i \mathbf{X}_i)^{-1} (\sum_{i=1}^{N} \mathbf{X}_i^\top \mathbf{M}_i \mathbf{y}_{i\cdot})$, where $\mathbf{M}_i = \mathbf{I}_T - \boldsymbol{\Xi}_i (\boldsymbol{\Xi}_i^\top \boldsymbol{\Xi}_i)^{-1} \boldsymbol{\Xi}_i^\top$. To remove the bias in dynamic panel settings, Juodis et al. (2021) adopts the Split-Panel Jackknife strategy (Dhaene & Jochmans, 2015) and considers

$$\widehat{\boldsymbol{\beta}} = 2\widehat{\boldsymbol{\beta}}_0 - \frac{\widehat{\boldsymbol{\beta}}_1 + \widehat{\boldsymbol{\beta}}_2}{2},$$

where $\widehat{\boldsymbol{\beta}}_1$ and $\widehat{\boldsymbol{\beta}}_2$ are the least-square estimators of $\boldsymbol{\beta}$ using the first $T_1 = T/2$ observations and the last $T_2 = T - T_1$ observations, respectively. The HPJ statistic is constructed using the quadratic form of the estimator $\widehat{\boldsymbol{\beta}}$, i.e.,

$$T_{HPJ} = \widehat{\boldsymbol{\beta}}^\top [\widehat{cov}(\widehat{\boldsymbol{\beta}})]^{-1} \widehat{\boldsymbol{\beta}},$$

where $\widehat{cov}(\widehat{\boldsymbol{\beta}})$ is an estimate of $cov(\widehat{\boldsymbol{\beta}})$, taking the explicit form of $\widehat{cov}(\widehat{\boldsymbol{\beta}}) = \frac{1}{NT} \cdot \widehat{\mathbf{J}}^{-1} \widehat{\mathbf{V}} \widehat{\mathbf{J}}^{-1}$, with $\widehat{\mathbf{J}} = \frac{1}{NT} \sum_{i=1}^{N} \mathbf{X}_i^\top \mathbf{M}_i \mathbf{X}_i$ and $\widehat{\mathbf{V}} = \frac{1}{N(T-K-1)-K} \sum_{i=1}^{N} \mathbf{X}_i^\top \mathbf{M}_i (\mathbf{y}_{i\cdot} - \mathbf{X}_i \widehat{\boldsymbol{\beta}}_0)(\mathbf{y}_{i\cdot} - \mathbf{X}_i \widehat{\boldsymbol{\beta}}_0)^\top \mathbf{M}_i \mathbf{X}_i$.

With Theorem 2 of Fernández-Val & Lee (2013) and Corollary 3.1 of Juodis et al. (2021), under some regularity conditions, $T_{HPJ} \to \chi_K^2$ as $T, N \to \infty$. The HPJ rejects $H_0$ if $T_{HPJ} \geq q_\alpha$, where $q_\alpha$ is the upper-$\alpha$ quantile of the $\chi_K^2$ distribution.

## C. Supplemental Simulation Results

### C.1. Simulation Results with non-Gaussian Errors

Besides the Gaussian error setting, we conduct simulations with non-Gaussian innovations. Specifically, errors are generated by $\epsilon_{i,t} \sim$ standardized $t_3$. Table S.1 summarizes the empirical Type-I error under the non-Gaussian error setting. The results are similar to those of the Gaussian error, demonstrating the performance of $J_{PE}$ in accommodating non-Gaussian errors. For power evaluation, we again consider situations with different signal sparsity levels and different signal strengths. We adopt the same data-generating process as those in Section 4. Again, since DH and HPJ are known to have (sometimes severe) size distortions under certain configurations, we report size-corrected power for these two to ensure fair comparisons. Figure S.1 plots a representative subset of results under sparse alternatives, with complete results reported in Table S.4. Figure S.2 visualizes a representative subset of power results under various signal sparsity levels, with complete results reported in Table S.5. The results under the non-Gaussian setting largely resemble the patterns of the Gaussian setting discussed in Section 4.

*Table S.1.* Empirical Type I error under the non-Gaussian error setting (nominal level $\alpha = 0.05$).

| $T$ | $N$ | PE-PGCT | DH | HPJ | $T$ | $N$ | PE-PGCT | DH | HPJ | $T$ | $N$ | PE-PGCT | DH | HPJ |
|---|---|---|---|---|---|---|---|---|---|---|---|---|---|---|
| 50 | 5 | 0.059 | 0.058 | 0.158 | 100 | 5 | 0.046 | 0.046 | 0.156 | 200 | 5 | 0.058 | 0.058 | 0.154 |
| | 10 | 0.047 | 0.047 | 0.105 | | 10 | 0.051 | 0.051 | 0.104 | | 10 | 0.049 | 0.049 | 0.107 |
| | 20 | 0.070 | 0.070 | 0.087 | | 20 | 0.051 | 0.049 | 0.075 | | 20 | 0.038 | 0.038 | 0.071 |
| | 50 | 0.071 | 0.079 | 0.066 | | 50 | 0.074 | 0.073 | 0.054 | | 50 | 0.045 | 0.045 | 0.061 |
| | 100 | 0.065 | 0.079 | 0.062 | | 100 | 0.061 | 0.054 | 0.060 | | 100 | 0.055 | 0.055 | 0.045 |
| | 200 | 0.056 | 0.103 | 0.054 | | 200 | 0.057 | 0.071 | 0.057 | | 200 | 0.052 | 0.063 | 0.052 |
| | 500 | 0.059 | 0.204 | 0.057 | | 500 | 0.057 | 0.093 | 0.057 | | 500 | 0.052 | 0.071 | 0.052 |

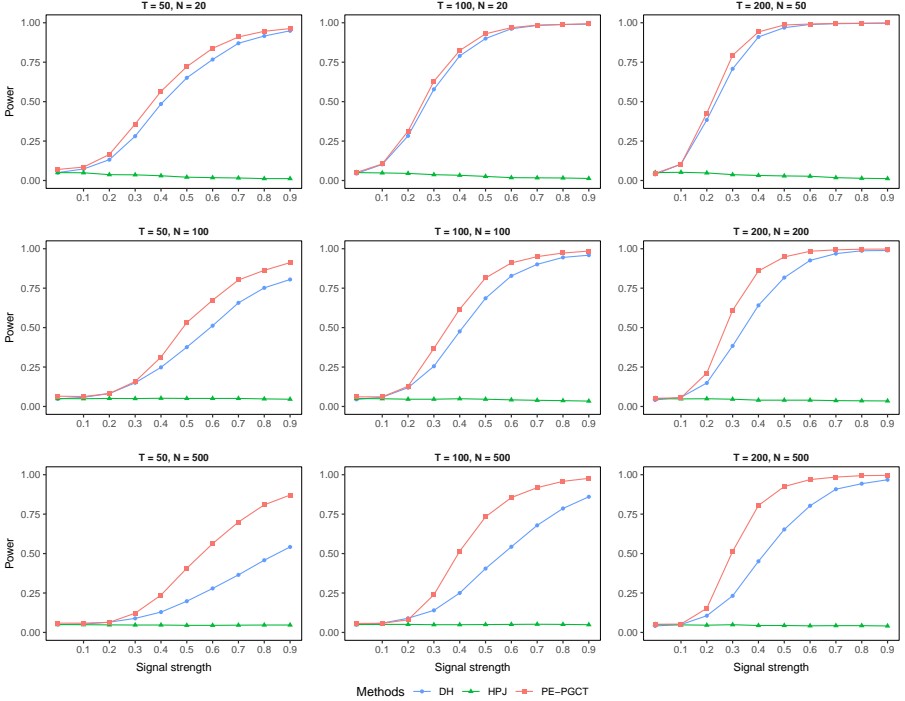

*Figure S.1.* Empirical power under sparse alternatives ($N_0 = 1$) under the non-Gaussian setting. Tests are conducted at level $\alpha = 0.05$.

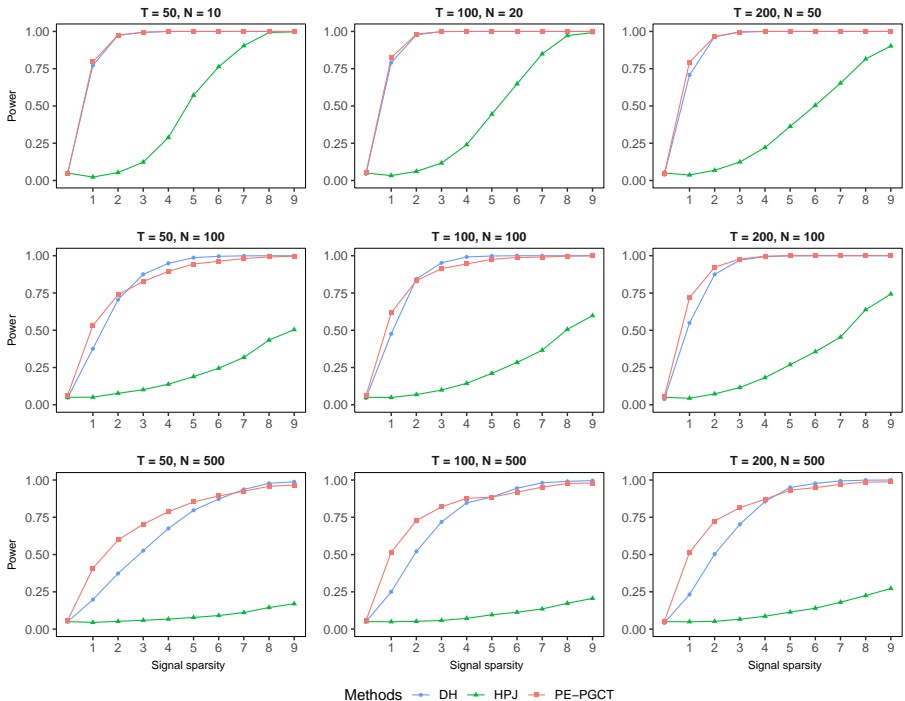

*Figure S.2.* Empirical power in situations of different sparsity levels (signal number ranging from 0 to 9) under the non-Gaussian setting. Tests are conducted at the nominal level $\alpha = 0.05$.

*Table S.2.* Empirical power under sparse alternatives ($N_0 = 1$) under the Gaussian error setting. Tests are conducted at level $\alpha = 0.05$.

| $T$ | $N$ | $\beta$ | PE-PGCT | DH | HPJ | $T$ | $N$ | $\beta$ | PE-PGCT | DH | HPJ | $T$ | $N$ | $\beta$ | PE-PGCT | DH | HPJ |
|---|---|---|---|---|---|---|---|---|---|---|---|---|---|---|---|---|---|
| 50 | 5 | 0 | 0.060 | 0.050 | 0.050 | 100 | 5 | 0 | 0.044 | 0.050 | 0.050 | 200 | 5 | 0 | 0.061 | 0.050 | 0.050 |
| | | 0.1 | 0.089 | 0.082 | 0.049 | | | 0.1 | 0.136 | 0.147 | 0.050 | | | 0.1 | 0.278 | 0.261 | 0.037 |
| | | 0.2 | 0.230 | 0.205 | 0.042 | | | 0.2 | 0.415 | 0.434 | 0.034 | | | 0.2 | 0.741 | 0.732 | 0.016 |
| | | 0.3 | 0.428 | 0.408 | 0.028 | | | 0.3 | 0.760 | 0.777 | 0.012 | | | 0.3 | 0.959 | 0.954 | 0.005 |
| | | 0.4 | 0.650 | 0.627 | 0.022 | | | 0.4 | 0.928 | 0.932 | 0.002 | | | 0.4 | 0.996 | 0.995 | 0.002 |
| | | 0.5 | 0.813 | 0.795 | 0.015 | | | 0.5 | 0.985 | 0.988 | 0.001 | | | 0.5 | 0.999 | 0.999 | 0.000 |
| | | 0.6 | 0.915 | 0.904 | 0.004 | | | 0.6 | 0.998 | 0.999 | 0.000 | | | 0.6 | 1.000 | 1.000 | 0.000 |
| | | 0.7 | 0.953 | 0.950 | 0.001 | | | 0.7 | 1.000 | 1.000 | 0.000 | | | 0.7 | 1.000 | 1.000 | 0.000 |
| | | 0.8 | 0.977 | 0.975 | 0.000 | | | 0.8 | 1.000 | 1.000 | 0.000 | | | 0.8 | 1.000 | 1.000 | 0.000 |
| | | 0.9 | 0.986 | 0.985 | 0.000 | | | 0.9 | 1.000 | 1.000 | 0.000 | | | 0.9 | 1.000 | 1.000 | 0.000 |
| 50 | 10 | 0 | 0.052 | 0.050 | 0.050 | 100 | 10 | 0 | 0.041 | 0.049 | 0.050 | 200 | 10 | 0 | 0.052 | 0.049 | 0.050 |
| | | 0.1 | 0.079 | 0.078 | 0.064 | | | 0.1 | 0.102 | 0.111 | 0.050 | | | 0.1 | 0.172 | 0.170 | 0.061 |
| | | 0.2 | 0.173 | 0.167 | 0.062 | | | 0.2 | 0.334 | 0.349 | 0.036 | | | 0.2 | 0.606 | 0.599 | 0.045 |
| | | 0.3 | 0.353 | 0.330 | 0.048 | | | 0.3 | 0.679 | 0.681 | 0.023 | | | 0.3 | 0.904 | 0.895 | 0.024 |
| | | 0.4 | 0.560 | 0.534 | 0.033 | | | 0.4 | 0.873 | 0.870 | 0.013 | | | 0.4 | 0.989 | 0.987 | 0.008 |
| | | 0.5 | 0.742 | 0.714 | 0.025 | | | 0.5 | 0.958 | 0.954 | 0.005 | | | 0.5 | 0.999 | 0.999 | 0.003 |
| | | 0.6 | 0.863 | 0.846 | 0.018 | | | 0.6 | 0.986 | 0.985 | 0.002 | | | 0.6 | 1.000 | 1.000 | 0.001 |
| | | 0.7 | 0.934 | 0.921 | 0.014 | | | 0.7 | 0.997 | 0.997 | 0.002 | | | 0.7 | 1.000 | 1.000 | 0.001 |
| | | 0.8 | 0.965 | 0.954 | 0.009 | | | 0.8 | 1.000 | 0.999 | 0.001 | | | 0.8 | 1.000 | 1.000 | 0.000 |
| | | 0.9 | 0.987 | 0.980 | 0.006 | | | 0.9 | 1.000 | 1.000 | 0.000 | | | 0.9 | 1.000 | 1.000 | 0.000 |
| 50 | 20 | 0 | 0.062 | 0.049 | 0.050 | 100 | 20 | 0 | 0.056 | 0.047 | 0.050 | 200 | 20 | 0 | 0.060 | 0.047 | 0.050 |
| | | 0.1 | 0.088 | 0.074 | 0.047 | | | 0.1 | 0.086 | 0.078 | 0.057 | | | 0.1 | 0.133 | 0.118 | 0.060 |
| | | 0.2 | 0.159 | 0.137 | 0.044 | | | 0.2 | 0.268 | 0.242 | 0.053 | | | 0.2 | 0.526 | 0.480 | 0.047 |
| | | 0.3 | 0.312 | 0.271 | 0.044 | | | 0.3 | 0.554 | 0.507 | 0.044 | | | 0.3 | 0.860 | 0.831 | 0.031 |
| | | 0.4 | 0.494 | 0.426 | 0.040 | | | 0.4 | 0.793 | 0.754 | 0.032 | | | 0.4 | 0.974 | 0.960 | 0.018 |
| | | 0.5 | 0.666 | 0.606 | 0.036 | | | 0.5 | 0.925 | 0.892 | 0.024 | | | 0.5 | 0.998 | 0.994 | 0.009 |
| | | 0.6 | 0.800 | 0.733 | 0.030 | | | 0.6 | 0.981 | 0.965 | 0.017 | | | 0.6 | 1.000 | 1.000 | 0.005 |
| | | 0.7 | 0.880 | 0.839 | 0.025 | | | 0.7 | 0.993 | 0.991 | 0.012 | | | 0.7 | 1.000 | 1.000 | 0.002 |
| | | 0.8 | 0.942 | 0.905 | 0.022 | | | 0.8 | 0.999 | 0.996 | 0.007 | | | 0.8 | 1.000 | 1.000 | 0.001 |
| | | 0.9 | 0.964 | 0.948 | 0.020 | | | 0.9 | 1.000 | 1.000 | 0.003 | | | 0.9 | 1.000 | 1.000 | 0.001 |
| 50 | 50 | 0 | 0.074 | 0.046 | 0.050 | 100 | 50 | 0 | 0.053 | 0.043 | 0.050 | 200 | 50 | 0 | 0.065 | 0.047 | 0.050 |
| | | 0.1 | 0.061 | 0.067 | 0.048 | | | 0.1 | 0.078 | 0.071 | 0.044 | | | 0.1 | 0.100 | 0.086 | 0.047 |
| | | 0.2 | 0.067 | 0.093 | 0.044 | | | 0.2 | 0.199 | 0.174 | 0.046 | | | 0.2 | 0.383 | 0.306 | 0.045 |
| | | 0.3 | 0.127 | 0.157 | 0.041 | | | 0.3 | 0.434 | 0.353 | 0.044 | | | 0.3 | 0.765 | 0.662 | 0.038 |
| | | 0.4 | 0.268 | 0.276 | 0.041 | | | 0.4 | 0.721 | 0.598 | 0.040 | | | 0.4 | 0.938 | 0.876 | 0.032 |
| | | 0.5 | 0.457 | 0.417 | 0.042 | | | 0.5 | 0.873 | 0.781 | 0.040 | | | 0.5 | 0.993 | 0.969 | 0.027 |
| | | 0.6 | 0.638 | 0.567 | 0.040 | | | 0.6 | 0.946 | 0.895 | 0.037 | | | 0.6 | 0.999 | 0.995 | 0.020 |
| | | 0.7 | 0.780 | 0.680 | 0.039 | | | 0.7 | 0.977 | 0.948 | 0.032 | | | 0.7 | 1.000 | 1.000 | 0.014 |
| | | 0.8 | 0.872 | 0.782 | 0.033 | | | 0.8 | 0.993 | 0.971 | 0.027 | | | 0.8 | 1.000 | 1.000 | 0.012 |
| | | 0.9 | 0.938 | 0.856 | 0.029 | | | 0.9 | 0.999 | 0.989 | 0.024 | | | 0.9 | 1.000 | 1.000 | 0.010 |
| 50 | 100 | 0 | 0.072 | 0.049 | 0.050 | 100 | 100 | 0 | 0.050 | 0.041 | 0.050 | 200 | 100 | 0 | 0.059 | 0.043 | 0.050 |
| | | 0.1 | 0.075 | 0.051 | 0.051 | | | 0.1 | 0.056 | 0.046 | 0.056 | | | 0.1 | 0.084 | 0.071 | 0.050 |
| | | 0.2 | 0.080 | 0.073 | 0.054 | | | 0.2 | 0.095 | 0.109 | 0.060 | | | 0.2 | 0.295 | 0.219 | 0.054 |
| | | 0.3 | 0.126 | 0.116 | 0.055 | | | 0.3 | 0.279 | 0.240 | 0.064 | | | 0.3 | 0.680 | 0.501 | 0.053 |
| | | 0.4 | 0.228 | 0.203 | 0.053 | | | 0.4 | 0.580 | 0.434 | 0.061 | | | 0.4 | 0.914 | 0.779 | 0.051 |
| | | 0.5 | 0.405 | 0.310 | 0.050 | | | 0.5 | 0.799 | 0.641 | 0.062 | | | 0.5 | 0.986 | 0.921 | 0.047 |
| | | 0.6 | 0.609 | 0.438 | 0.049 | | | 0.6 | 0.917 | 0.792 | 0.056 | | | 0.6 | 1.000 | 0.971 | 0.036 |
| | | 0.7 | 0.757 | 0.567 | 0.049 | | | 0.7 | 0.971 | 0.885 | 0.050 | | | 0.7 | 1.000 | 0.997 | 0.033 |
| | | 0.8 | 0.863 | 0.691 | 0.047 | | | 0.8 | 0.989 | 0.942 | 0.048 | | | 0.8 | 1.000 | 0.999 | 0.026 |
| | | 0.9 | 0.925 | 0.794 | 0.046 | | | 0.9 | 0.994 | 0.980 | 0.044 | | | 0.9 | 1.000 | 1.000 | 0.021 |
| 50 | 200 | 0 | 0.069 | 0.050 | 0.050 | 100 | 200 | 0 | 0.056 | 0.047 | 0.050 | 200 | 200 | 0 | 0.057 | 0.042 | 0.050 |
| | | 0.1 | 0.065 | 0.053 | 0.051 | | | 0.1 | 0.058 | 0.053 | 0.053 | | | 0.1 | 0.066 | 0.069 | 0.056 |
| | | 0.2 | 0.073 | 0.071 | 0.052 | | | 0.2 | 0.072 | 0.085 | 0.050 | | | 0.2 | 0.160 | 0.159 | 0.053 |
| | | 0.3 | 0.105 | 0.094 | 0.049 | | | 0.3 | 0.224 | 0.167 | 0.051 | | | 0.3 | 0.529 | 0.354 | 0.057 |
| | | 0.4 | 0.179 | 0.152 | 0.051 | | | 0.4 | 0.457 | 0.290 | 0.050 | | | 0.4 | 0.844 | 0.600 | 0.054 |
| | | 0.5 | 0.334 | 0.213 | 0.052 | | | 0.5 | 0.711 | 0.452 | 0.048 | | | 0.5 | 0.956 | 0.825 | 0.048 |
| | | 0.6 | 0.519 | 0.305 | 0.052 | | | 0.6 | 0.887 | 0.620 | 0.049 | | | 0.6 | 0.990 | 0.934 | 0.047 |
| | | 0.7 | 0.680 | 0.413 | 0.051 | | | 0.7 | 0.951 | 0.753 | 0.048 | | | 0.7 | 0.999 | 0.977 | 0.046 |
| | | 0.8 | 0.806 | 0.520 | 0.052 | | | 0.8 | 0.980 | 0.853 | 0.046 | | | 0.8 | 1.000 | 0.988 | 0.043 |
| | | 0.9 | 0.888 | 0.632 | 0.051 | | | 0.9 | 0.994 | 0.918 | 0.046 | | | 0.9 | 1.000 | 1.000 | 0.040 |
| 50 | 500 | 0 | 0.052 | 0.050 | 0.050 | 100 | 500 | 0 | 0.052 | 0.048 | 0.050 | 200 | 500 | 0 | 0.053 | 0.039 | 0.050 |
| | | 0.1 | 0.050 | 0.054 | 0.053 | | | 0.1 | 0.048 | 0.050 | 0.045 | | | 0.1 | 0.054 | 0.048 | 0.051 |
| | | 0.2 | 0.054 | 0.058 | 0.051 | | | 0.2 | 0.050 | 0.073 | 0.044 | | | 0.2 | 0.109 | 0.099 | 0.048 |
| | | 0.3 | 0.081 | 0.070 | 0.049 | | | 0.3 | 0.152 | 0.123 | 0.041 | | | 0.3 | 0.446 | 0.233 | 0.048 |
| | | 0.4 | 0.155 | 0.099 | 0.052 | | | 0.4 | 0.415 | 0.186 | 0.040 | | | 0.4 | 0.794 | 0.440 | 0.045 |
| | | 0.5 | 0.304 | 0.146 | 0.052 | | | 0.5 | 0.670 | 0.300 | 0.041 | | | 0.5 | 0.943 | 0.628 | 0.044 |
| | | 0.6 | 0.467 | 0.196 | 0.056 | | | 0.6 | 0.850 | 0.461 | 0.042 | | | 0.6 | 0.988 | 0.787 | 0.042 |
| | | 0.7 | 0.613 | 0.263 | 0.057 | | | 0.7 | 0.940 | 0.591 | 0.043 | | | 0.7 | 0.997 | 0.895 | 0.042 |
| | | 0.8 | 0.751 | 0.358 | 0.055 | | | 0.8 | 0.979 | 0.716 | 0.044 | | | 0.8 | 1.000 | 0.960 | 0.041 |
| | | 0.9 | 0.841 | 0.448 | 0.058 | | | 0.9 | 0.994 | 0.798 | 0.043 | | | 0.9 | 1.000 | 0.985 | 0.039 |

*Table S.3.* Empirical power in situations of different sparsity levels (signal number ranging from 0 to 9) under the Gaussian error setting. Tests are conducted at the nominal level $\alpha = 0.05$.

| $T$ | $N$ | $\beta$ | $N_0$ | PE-PGCT | DH | HPJ | $T$ | $N$ | $\beta$ | $N_0$ | PE-PGCT | DH | HPJ | $T$ | $N$ | $\beta$ | $N_0$ | PE-PGCT | DH | HPJ |
|---|---|---|---|---|---|---|---|---|---|---|---|---|---|---|---|---|---|---|---|---|
| 50 | 5 | 0.5 | 0 | 0.06 | 0.05 | 0.05 | 100 | 5 | 0.4 | 0 | 0.044 | 0.05 | 0.05 | 200 | 5 | 0.3 | 0 | 0.061 | 0.05 | 0.05 |
| | | | 1 | 0.813 | 0.795 | 0.015 | | | | 1 | 0.928 | 0.932 | 0.002 | | | | 1 | 0.959 | 0.954 | 0.005 |
| | | | 2 | 0.986 | 0.984 | 0.046 | | | | 2 | 1 | 1 | 0.027 | | | | 2 | 0.998 | 0.998 | 0.032 |
| | | | 3 | 0.998 | 0.998 | 0.201 | | | | 3 | 1 | 1 | 0.196 | | | | 3 | 1 | 1 | 0.244 |
| | | | 4 | 1 | 1 | 0.593 | | | | 4 | 1 | 1 | 0.651 | | | | 4 | 1 | 1 | 0.712 |
| | | | 5 | 1 | 1 | 1 | | | | 5 | 1 | 1 | 1 | | | | 5 | 1 | 1 | 1 |
| 50 | 10 | 0.5 | 0 | 0.052 | 0.05 | 0.05 | 100 | 10 | 0.4 | 0 | 0.041 | 0.049 | 0.05 | 200 | 10 | 0.3 | 0 | 0.052 | 0.049 | 0.05 |
| | | | 1 | 0.742 | 0.714 | 0.025 | | | | 1 | 0.873 | 0.87 | 0.013 | | | | 1 | 0.904 | 0.895 | 0.024 |
| | | | 2 | 0.966 | 0.962 | 0.056 | | | | 2 | 0.994 | 0.994 | 0.025 | | | | 2 | 0.998 | 0.998 | 0.051 |
| | | | 3 | 0.995 | 0.994 | 0.125 | | | | 3 | 0.999 | 0.999 | 0.084 | | | | 3 | 1 | 1 | 0.154 |
| | | | 4 | 1 | 1 | 0.294 | | | | 4 | 1 | 1 | 0.226 | | | | 4 | 1 | 1 | 0.404 |
| | | | 5 | 1 | 1 | 0.564 | | | | 5 | 1 | 1 | 0.52 | | | | 5 | 1 | 1 | 0.732 |
| | | | 6 | 1 | 1 | 0.763 | | | | 6 | 1 | 1 | 0.754 | | | | 6 | 1 | 1 | 0.89 |
| | | | 7 | 1 | 1 | 0.906 | | | | 7 | 1 | 1 | 0.923 | | | | 7 | 1 | 1 | 0.975 |
| | | | 8 | 1 | 1 | 0.995 | | | | 8 | 1 | 1 | 1 | | | | 8 | 1 | 1 | 1 |
| | | | 9 | 1 | 1 | 0.999 | | | | 9 | 1 | 1 | 1 | | | | 9 | 1 | 1 | 1 |
| 50 | 20 | 0.5 | 0 | 0.062 | 0.049 | 0.05 | 100 | 20 | 0.4 | 0 | 0.056 | 0.047 | 0.05 | 200 | 20 | 0.3 | 0 | 0.06 | 0.047 | 0.05 |
| | | | 1 | 0.666 | 0.606 | 0.036 | | | | 1 | 0.793 | 0.754 | 0.032 | | | | 1 | 0.86 | 0.831 | 0.031 |
| | | | 2 | 0.934 | 0.922 | 0.053 | | | | 2 | 0.976 | 0.974 | 0.058 | | | | 2 | 0.993 | 0.992 | 0.06 |
| | | | 3 | 0.985 | 0.984 | 0.103 | | | | 3 | 0.999 | 0.999 | 0.128 | | | | 3 | 1 | 1 | 0.15 |
| | | | 4 | 0.997 | 0.996 | 0.179 | | | | 4 | 1 | 1 | 0.226 | | | | 4 | 1 | 1 | 0.291 |
| | | | 5 | 1 | 1 | 0.311 | | | | 5 | 1 | 1 | 0.414 | | | | 5 | 1 | 1 | 0.511 |
| | | | 6 | 1 | 1 | 0.478 | | | | 6 | 1 | 1 | 0.611 | | | | 6 | 1 | 1 | 0.71 |
| | | | 7 | 1 | 1 | 0.67 | | | | 7 | 1 | 1 | 0.79 | | | | 7 | 1 | 1 | 0.89 |
| | | | 8 | 1 | 1 | 0.869 | | | | 8 | 1 | 1 | 0.945 | | | | 8 | 1 | 1 | 0.986 |
| | | | 9 | 1 | 1 | 0.935 | | | | 9 | 1 | 1 | 0.988 | | | | 9 | 1 | 1 | 0.997 |
| 50 | 50 | 0.5 | 0 | 0.074 | 0.046 | 0.05 | 100 | 50 | 0.4 | 0 | 0.053 | 0.043 | 0.05 | 200 | 50 | 0.3 | 0 | 0.065 | 0.047 | 0.05 |
| | | | 1 | 0.457 | 0.417 | 0.042 | | | | 1 | 0.721 | 0.598 | 0.04 | | | | 1 | 0.765 | 0.662 | 0.038 |
| | | | 2 | 0.684 | 0.746 | 0.058 | | | | 2 | 0.926 | 0.897 | 0.064 | | | | 2 | 0.953 | 0.94 | 0.068 |
| | | | 3 | 0.807 | 0.893 | 0.095 | | | | 3 | 0.98 | 0.977 | 0.128 | | | | 3 | 0.998 | 0.997 | 0.119 |
| | | | 4 | 0.878 | 0.974 | 0.15 | | | | 4 | 0.998 | 0.998 | 0.195 | | | | 4 | 1 | 1 | 0.208 |
| | | | 5 | 0.927 | 0.994 | 0.249 | | | | 5 | 1 | 1 | 0.304 | | | | 5 | 1 | 1 | 0.33 |
| | | | 6 | 0.957 | 0.999 | 0.334 | | | | 6 | 1 | 1 | 0.431 | | | | 6 | 1 | 1 | 0.466 |
| | | | 7 | 0.972 | 1 | 0.456 | | | | 7 | 1 | 1 | 0.584 | | | | 7 | 1 | 1 | 0.606 |
| | | | 8 | 0.99 | 1 | 0.61 | | | | 8 | 1 | 1 | 0.747 | | | | 8 | 1 | 1 | 0.799 |
| | | | 9 | 0.996 | 1 | 0.725 | | | | 9 | 1 | 1 | 0.852 | | | | 9 | 1 | 1 | 0.881 |
| 50 | 100 | 0.5 | 0 | 0.072 | 0.049 | 0.05 | 100 | 100 | 0.4 | 0 | 0.05 | 0.041 | 0.05 | 200 | 100 | 0.3 | 0 | 0.059 | 0.043 | 0.05 |
| | | | 1 | 0.405 | 0.31 | 0.05 | | | | 1 | 0.58 | 0.434 | 0.061 | | | | 1 | 0.68 | 0.501 | 0.053 |
| | | | 2 | 0.604 | 0.603 | 0.065 | | | | 2 | 0.77 | 0.766 | 0.091 | | | | 2 | 0.91 | 0.851 | 0.079 |
| | | | 3 | 0.699 | 0.794 | 0.081 | | | | 3 | 0.855 | 0.914 | 0.135 | | | | 3 | 0.977 | 0.961 | 0.124 |
| | | | 4 | 0.775 | 0.916 | 0.115 | | | | 4 | 0.903 | 0.978 | 0.19 | | | | 4 | 0.996 | 0.993 | 0.19 |
| | | | 5 | 0.845 | 0.974 | 0.16 | | | | 5 | 0.948 | 0.997 | 0.258 | | | | 5 | 1 | 1 | 0.296 |
| | | | 6 | 0.878 | 0.989 | 0.201 | | | | 6 | 0.963 | 0.999 | 0.325 | | | | 6 | 1 | 1 | 0.378 |
| | | | 7 | 0.92 | 0.997 | 0.252 | | | | 7 | 0.978 | 1 | 0.422 | | | | 7 | 1 | 1 | 0.503 |
| | | | 8 | 0.961 | 1 | 0.347 | | | | 8 | 0.988 | 1 | 0.55 | | | | 8 | 1 | 1 | 0.653 |
| | | | 9 | 0.975 | 1 | 0.434 | | | | 9 | 0.993 | 1 | 0.653 | | | | 9 | 1 | 1 | 0.756 |
| 50 | 200 | 0.5 | 0 | 0.069 | 0.05 | 0.05 | 100 | 200 | 0.4 | 0 | 0.056 | 0.047 | 0.05 | 200 | 200 | 0.3 | 0 | 0.057 | 0.042 | 0.05 |
| | | | 1 | 0.334 | 0.213 | 0.052 | | | | 1 | 0.457 | 0.29 | 0.05 | | | | 1 | 0.529 | 0.354 | 0.057 |
| | | | 2 | 0.492 | 0.433 | 0.063 | | | | 2 | 0.685 | 0.596 | 0.063 | | | | 2 | 0.741 | 0.708 | 0.078 |
| | | | 3 | 0.6 | 0.631 | 0.074 | | | | 3 | 0.777 | 0.801 | 0.086 | | | | 3 | 0.823 | 0.882 | 0.105 |
| | | | 4 | 0.706 | 0.793 | 0.098 | | | | 4 | 0.85 | 0.919 | 0.112 | | | | 4 | 0.886 | 0.975 | 0.136 |
| | | | 5 | 0.789 | 0.901 | 0.134 | | | | 5 | 0.907 | 0.971 | 0.157 | | | | 5 | 0.925 | 0.991 | 0.187 |
| | | | 6 | 0.841 | 0.95 | 0.162 | | | | 6 | 0.941 | 0.99 | 0.209 | | | | 6 | 0.953 | 0.999 | 0.261 |
| | | | 7 | 0.881 | 0.984 | 0.201 | | | | 7 | 0.959 | 0.999 | 0.284 | | | | 7 | 0.979 | 1 | 0.33 |
| | | | 8 | 0.924 | 0.996 | 0.249 | | | | 8 | 0.982 | 1 | 0.368 | | | | 8 | 0.991 | 1 | 0.432 |
| | | | 9 | 0.94 | 0.997 | 0.292 | | | | 9 | 0.984 | 1 | 0.447 | | | | 9 | 0.995 | 1 | 0.519 |
| 50 | 500 | 0.5 | 0 | 0.052 | 0.05 | 0.05 | 100 | 500 | 0.4 | 0 | 0.052 | 0.048 | 0.05 | 200 | 500 | 0.3 | 0 | 0.053 | 0.039 | 0.05 |
| | | | 1 | 0.304 | 0.146 | 0.052 | | | | 1 | 0.415 | 0.186 | 0.04 | | | | 1 | 0.446 | 0.233 | 0.048 |
| | | | 2 | 0.461 | 0.262 | 0.064 | | | | 2 | 0.596 | 0.413 | 0.055 | | | | 2 | 0.654 | 0.484 | 0.049 |
| | | | 3 | 0.54 | 0.398 | 0.076 | | | | 3 | 0.683 | 0.567 | 0.069 | | | | 3 | 0.727 | 0.662 | 0.061 |
| | | | 4 | 0.613 | 0.537 | 0.085 | | | | 4 | 0.752 | 0.738 | 0.086 | | | | 4 | 0.805 | 0.839 | 0.084 |
| | | | 5 | 0.679 | 0.695 | 0.098 | | | | 5 | 0.842 | 0.864 | 0.123 | | | | 5 | 0.886 | 0.93 | 0.123 |
| | | | 6 | 0.729 | 0.793 | 0.117 | | | | 6 | 0.874 | 0.93 | 0.146 | | | | 6 | 0.916 | 0.974 | 0.149 |
| | | | 7 | 0.795 | 0.878 | 0.139 | | | | 7 | 0.922 | 0.971 | 0.175 | | | | 7 | 0.947 | 0.991 | 0.186 |
| | | | 8 | 0.865 | 0.95 | 0.174 | | | | 8 | 0.964 | 0.992 | 0.226 | | | | 8 | 0.978 | 0.999 | 0.25 |
| | | | 9 | 0.885 | 0.967 | 0.194 | | | | 9 | 0.969 | 0.998 | 0.261 | | | | 9 | 0.984 | 1 | 0.285 |

*Table S.4.* Empirical power under sparse alternatives ($N_0 = 1$) under the non-Gaussian setting. Tests are conducted at level $\alpha = 0.05$.

| $T$ | $N$ | $\beta$ | PE-PGCT | DH | HPJ | $T$ | $N$ | $\beta$ | PE-PGCT | DH | HPJ | $T$ | $N$ | $\beta$ | PE-PGCT | DH | HPJ |
|---|---|---|---|---|---|---|---|---|---|---|---|---|---|---|---|---|---|
| 50 | 5 | 0 | 0.059 | 0.050 | 0.050 | 100 | 5 | 0 | 0.046 | 0.050 | 0.050 | 200 | 5 | 0 | 0.058 | 0.050 | 0.050 |
| | | 0.1 | 0.109 | 0.094 | 0.054 | | | 0.1 | 0.170 | 0.194 | 0.057 | | | 0.1 | 0.288 | 0.277 | 0.038 |
| | | 0.2 | 0.286 | 0.258 | 0.045 | | | 0.2 | 0.526 | 0.558 | 0.041 | | | 0.2 | 0.775 | 0.765 | 0.017 |
| | | 0.3 | 0.531 | 0.498 | 0.026 | | | 0.3 | 0.808 | 0.825 | 0.019 | | | 0.3 | 0.962 | 0.959 | 0.003 |
| | | 0.4 | 0.735 | 0.699 | 0.016 | | | 0.4 | 0.941 | 0.947 | 0.007 | | | 0.4 | 0.992 | 0.992 | 0.002 |
| | | 0.5 | 0.869 | 0.852 | 0.003 | | | 0.5 | 0.975 | 0.976 | 0.001 | | | 0.5 | 0.998 | 0.998 | 0.001 |
| | | 0.6 | 0.925 | 0.914 | 0.002 | | | 0.6 | 0.991 | 0.992 | 0.001 | | | 0.6 | 0.999 | 0.999 | 0.000 |
| | | 0.7 | 0.956 | 0.948 | 0.002 | | | 0.7 | 0.999 | 0.999 | 0.001 | | | 0.7 | 0.999 | 0.999 | 0.000 |
| | | 0.8 | 0.979 | 0.973 | 0.001 | | | 0.8 | 0.999 | 0.999 | 0.001 | | | 0.8 | 0.999 | 0.999 | 0.000 |
| | | 0.9 | 0.989 | 0.988 | 0.001 | | | 0.9 | 1.000 | 1.000 | 0.000 | | | 0.9 | 1.000 | 1.000 | 0.000 |
| 50 | 10 | 0 | 0.047 | 0.050 | 0.050 | 100 | 10 | 0 | 0.051 | 0.050 | 0.050 | 200 | 10 | 0 | 0.049 | 0.049 | 0.050 |
| | | 0.1 | 0.091 | 0.091 | 0.054 | | | 0.1 | 0.128 | 0.126 | 0.056 | | | 0.1 | 0.202 | 0.212 | 0.049 |
| | | 0.2 | 0.221 | 0.208 | 0.060 | | | 0.2 | 0.399 | 0.380 | 0.043 | | | 0.2 | 0.692 | 0.694 | 0.025 |
| | | 0.3 | 0.414 | 0.403 | 0.046 | | | 0.3 | 0.691 | 0.679 | 0.027 | | | 0.3 | 0.914 | 0.915 | 0.014 |
| | | 0.4 | 0.628 | 0.612 | 0.036 | | | 0.4 | 0.880 | 0.864 | 0.020 | | | 0.4 | 0.982 | 0.981 | 0.004 |
| | | 0.5 | 0.796 | 0.772 | 0.023 | | | 0.5 | 0.954 | 0.948 | 0.012 | | | 0.5 | 0.995 | 0.995 | 0.000 |
| | | 0.6 | 0.880 | 0.871 | 0.014 | | | 0.6 | 0.981 | 0.979 | 0.007 | | | 0.6 | 0.999 | 0.999 | 0.000 |
| | | 0.7 | 0.928 | 0.920 | 0.009 | | | 0.7 | 0.990 | 0.990 | 0.003 | | | 0.7 | 0.999 | 0.999 | 0.000 |
| | | 0.8 | 0.953 | 0.945 | 0.007 | | | 0.8 | 0.995 | 0.995 | 0.003 | | | 0.8 | 1.000 | 1.000 | 0.000 |
| | | 0.9 | 0.973 | 0.968 | 0.005 | | | 0.9 | 0.997 | 0.997 | 0.001 | | | 0.9 | 1.000 | 1.000 | 0.000 |
| 50 | 20 | 0 | 0.070 | 0.050 | 0.050 | 100 | 20 | 0 | 0.051 | 0.044 | 0.050 | 200 | 20 | 0 | 0.038 | 0.043 | 0.050 |
| | | 0.1 | 0.085 | 0.073 | 0.049 | | | 0.1 | 0.105 | 0.102 | 0.048 | | | 0.1 | 0.151 | 0.161 | 0.062 |
| | | 0.2 | 0.165 | 0.132 | 0.037 | | | 0.2 | 0.313 | 0.283 | 0.045 | | | 0.2 | 0.540 | 0.547 | 0.056 |
| | | 0.3 | 0.358 | 0.281 | 0.036 | | | 0.3 | 0.629 | 0.578 | 0.037 | | | 0.3 | 0.861 | 0.854 | 0.039 |
| | | 0.4 | 0.565 | 0.485 | 0.030 | | | 0.4 | 0.824 | 0.790 | 0.033 | | | 0.4 | 0.963 | 0.960 | 0.026 |
| | | 0.5 | 0.722 | 0.651 | 0.021 | | | 0.5 | 0.930 | 0.900 | 0.026 | | | 0.5 | 0.992 | 0.991 | 0.018 |
| | | 0.6 | 0.838 | 0.767 | 0.018 | | | 0.6 | 0.969 | 0.962 | 0.018 | | | 0.6 | 0.996 | 0.995 | 0.009 |
| | | 0.7 | 0.910 | 0.870 | 0.016 | | | 0.7 | 0.985 | 0.983 | 0.017 | | | 0.7 | 0.999 | 0.999 | 0.002 |
| | | 0.8 | 0.946 | 0.916 | 0.012 | | | 0.8 | 0.989 | 0.989 | 0.016 | | | 0.8 | 0.999 | 1.000 | 0.001 |
| | | 0.9 | 0.963 | 0.949 | 0.012 | | | 0.9 | 0.994 | 0.991 | 0.013 | | | 0.9 | 1.000 | 1.000 | 0.000 |
| 50 | 50 | 0 | 0.071 | 0.048 | 0.050 | 100 | 50 | 0 | 0.074 | 0.044 | 0.050 | 200 | 50 | 0 | 0.045 | 0.039 | 0.050 |
| | | 0.1 | 0.066 | 0.060 | 0.042 | | | 0.1 | 0.102 | 0.076 | 0.053 | | | 0.1 | 0.103 | 0.104 | 0.052 |
| | | 0.2 | 0.084 | 0.104 | 0.043 | | | 0.2 | 0.246 | 0.182 | 0.054 | | | 0.2 | 0.427 | 0.384 | 0.048 |
| | | 0.3 | 0.190 | 0.185 | 0.044 | | | 0.3 | 0.521 | 0.399 | 0.048 | | | 0.3 | 0.794 | 0.708 | 0.037 |
| | | 0.4 | 0.377 | 0.316 | 0.038 | | | 0.4 | 0.757 | 0.625 | 0.049 | | | 0.4 | 0.942 | 0.910 | 0.032 |
| | | 0.5 | 0.543 | 0.466 | 0.036 | | | 0.5 | 0.885 | 0.789 | 0.043 | | | 0.5 | 0.986 | 0.969 | 0.029 |
| | | 0.6 | 0.717 | 0.619 | 0.033 | | | 0.6 | 0.947 | 0.889 | 0.040 | | | 0.6 | 0.992 | 0.989 | 0.027 |
| | | 0.7 | 0.837 | 0.742 | 0.031 | | | 0.7 | 0.971 | 0.942 | 0.032 | | | 0.7 | 0.996 | 0.994 | 0.018 |
| | | 0.8 | 0.905 | 0.823 | 0.031 | | | 0.8 | 0.990 | 0.966 | 0.026 | | | 0.8 | 0.997 | 0.996 | 0.014 |
| | | 0.9 | 0.949 | 0.887 | 0.030 | | | 0.9 | 0.995 | 0.983 | 0.023 | | | 0.9 | 0.999 | 0.997 | 0.012 |
| 50 | 100 | 0 | 0.065 | 0.048 | 0.050 | 100 | 100 | 0 | 0.061 | 0.045 | 0.050 | 200 | 100 | 0 | 0.055 | 0.038 | 0.050 |
| | | 0.1 | 0.063 | 0.058 | 0.049 | | | 0.1 | 0.061 | 0.059 | 0.049 | | | 0.1 | 0.083 | 0.065 | 0.053 |
| | | 0.2 | 0.082 | 0.082 | 0.051 | | | 0.2 | 0.128 | 0.119 | 0.046 | | | 0.2 | 0.321 | 0.225 | 0.047 |
| | | 0.3 | 0.158 | 0.150 | 0.050 | | | 0.3 | 0.368 | 0.255 | 0.046 | | | 0.3 | 0.718 | 0.549 | 0.044 |
| | | 0.4 | 0.313 | 0.248 | 0.052 | | | 0.4 | 0.617 | 0.476 | 0.049 | | | 0.4 | 0.913 | 0.785 | 0.047 |
| | | 0.5 | 0.532 | 0.376 | 0.051 | | | 0.5 | 0.815 | 0.686 | 0.046 | | | 0.5 | 0.969 | 0.916 | 0.045 |
| | | 0.6 | 0.674 | 0.512 | 0.051 | | | 0.6 | 0.910 | 0.828 | 0.042 | | | 0.6 | 0.990 | 0.967 | 0.040 |
| | | 0.7 | 0.802 | 0.657 | 0.051 | | | 0.7 | 0.950 | 0.901 | 0.039 | | | 0.7 | 0.995 | 0.986 | 0.033 |
| | | 0.8 | 0.863 | 0.752 | 0.048 | | | 0.8 | 0.973 | 0.945 | 0.037 | | | 0.8 | 1.000 | 0.994 | 0.031 |
| | | 0.9 | 0.912 | 0.805 | 0.046 | | | 0.9 | 0.985 | 0.959 | 0.034 | | | 0.9 | 1.000 | 0.998 | 0.026 |
| 50 | 200 | 0 | 0.056 | 0.049 | 0.050 | 100 | 200 | 0 | 0.057 | 0.048 | 0.050 | 200 | 200 | 0 | 0.052 | 0.041 | 0.050 |
| | | 0.1 | 0.054 | 0.049 | 0.050 | | | 0.1 | 0.059 | 0.064 | 0.050 | | | 0.1 | 0.055 | 0.058 | 0.048 |
| | | 0.2 | 0.070 | 0.070 | 0.052 | | | 0.2 | 0.086 | 0.101 | 0.045 | | | 0.2 | 0.212 | 0.149 | 0.049 |
| | | 0.3 | 0.123 | 0.111 | 0.050 | | | 0.3 | 0.293 | 0.187 | 0.042 | | | 0.3 | 0.610 | 0.384 | 0.046 |
| | | 0.4 | 0.252 | 0.165 | 0.048 | | | 0.4 | 0.566 | 0.350 | 0.042 | | | 0.4 | 0.859 | 0.641 | 0.040 |
| | | 0.5 | 0.430 | 0.264 | 0.048 | | | 0.5 | 0.755 | 0.525 | 0.042 | | | 0.5 | 0.948 | 0.817 | 0.040 |
| | | 0.6 | 0.604 | 0.384 | 0.048 | | | 0.6 | 0.876 | 0.676 | 0.041 | | | 0.6 | 0.984 | 0.926 | 0.040 |
| | | 0.7 | 0.748 | 0.497 | 0.047 | | | 0.7 | 0.945 | 0.798 | 0.041 | | | 0.7 | 0.993 | 0.969 | 0.037 |
| | | 0.8 | 0.828 | 0.608 | 0.047 | | | 0.8 | 0.974 | 0.873 | 0.040 | | | 0.8 | 0.997 | 0.987 | 0.036 |
| | | 0.9 | 0.887 | 0.705 | 0.048 | | | 0.9 | 0.987 | 0.927 | 0.040 | | | 0.9 | 0.998 | 0.989 | 0.035 |
| 50 | 500 | 0 | 0.059 | 0.050 | 0.050 | 100 | 500 | 0 | 0.057 | 0.049 | 0.050 | 200 | 500 | 0 | 0.052 | 0.041 | 0.050 |
| | | 0.1 | 0.059 | 0.053 | 0.049 | | | 0.1 | 0.058 | 0.059 | 0.051 | | | 0.1 | 0.053 | 0.050 | 0.048 |
| | | 0.2 | 0.064 | 0.065 | 0.048 | | | 0.2 | 0.080 | 0.090 | 0.051 | | | 0.2 | 0.153 | 0.106 | 0.046 |
| | | 0.3 | 0.122 | 0.089 | 0.047 | | | 0.3 | 0.239 | 0.140 | 0.049 | | | 0.3 | 0.514 | 0.232 | 0.049 |
| | | 0.4 | 0.237 | 0.129 | 0.047 | | | 0.4 | 0.515 | 0.250 | 0.049 | | | 0.4 | 0.805 | 0.451 | 0.044 |
| | | 0.5 | 0.407 | 0.198 | 0.045 | | | 0.5 | 0.733 | 0.405 | 0.050 | | | 0.5 | 0.925 | 0.653 | 0.044 |
| | | 0.6 | 0.563 | 0.279 | 0.045 | | | 0.6 | 0.855 | 0.543 | 0.051 | | | 0.6 | 0.969 | 0.803 | 0.042 |
| | | 0.7 | 0.700 | 0.365 | 0.046 | | | 0.7 | 0.918 | 0.679 | 0.052 | | | 0.7 | 0.985 | 0.908 | 0.043 |
| | | 0.8 | 0.809 | 0.458 | 0.047 | | | 0.8 | 0.958 | 0.786 | 0.051 | | | 0.8 | 0.994 | 0.943 | 0.043 |
| | | 0.9 | 0.871 | 0.542 | 0.047 | | | 0.9 | 0.977 | 0.860 | 0.049 | | | 0.9 | 0.996 | 0.968 | 0.041 |

*Table S.5.* Empirical power in situations of different sparsity levels (signal number ranging from 0 to 9) under the non-Gaussian setting. Tests are conducted at the nominal level $\alpha = 0.05$.

| $T$ | $N$ | $\beta$ | $N_0$ | PE-PGCT | DH | HPJ | $T$ | $N$ | $\beta$ | $N_0$ | PE-PGCT | DH | HPJ | $T$ | $N$ | $\beta$ | $N_0$ | PE-PGCT | DH | HPJ |
|---|---|---|---|---|---|---|---|---|---|---|---|---|---|---|---|---|---|---|---|---|
| 50 | 5 | 0.5 | 0 | 0.059 | 0.050 | 0.050 | 100 | 5 | 0.4 | 0 | 0.046 | 0.050 | 0.050 | 200 | 5 | 0.3 | 0 | 0.069 | 0.063 | 0.056 |
|  |  |  | 1 | 0.869 | 0.852 | 0.003 |  |  |  | 1 | 0.941 | 0.947 | 0.007 |  |  |  | 1 | 0.931 | 0.930 | 0.005 |
|  |  |  | 2 | 0.987 | 0.985 | 0.044 |  |  |  | 2 | 0.998 | 0.998 | 0.056 |  |  |  | 2 | 0.998 | 0.998 | 0.076 |
|  |  |  | 3 | 0.999 | 0.999 | 0.225 |  |  |  | 3 | 1.000 | 1.000 | 0.288 |  |  |  | 3 | 1.000 | 1.000 | 0.342 |
|  |  |  | 4 | 1.000 | 1.000 | 0.662 |  |  |  | 4 | 1.000 | 1.000 | 0.724 |  |  |  | 4 | 1.000 | 1.000 | 0.753 |
|  |  |  | 5 | 1.000 | 1.000 | 0.992 |  |  |  | 5 | 1.000 | 1.000 | 1.000 |  |  |  | 5 | 1.000 | 1.000 | 0.999 |
| 50 | 10 | 0.5 | 0 | 0.047 | 0.050 | 0.050 | 100 | 10 | 0.4 | 0 | 0.051 | 0.050 | 0.050 | 200 | 10 | 0.3 | 0 | 0.049 | 0.049 | 0.050 |
|  |  |  | 1 | 0.796 | 0.772 | 0.023 |  |  |  | 1 | 0.880 | 0.864 | 0.020 |  |  |  | 1 | 0.914 | 0.915 | 0.014 |
|  |  |  | 2 | 0.975 | 0.974 | 0.053 |  |  |  | 2 | 0.993 | 0.993 | 0.051 |  |  |  | 2 | 0.997 | 0.997 | 0.041 |
|  |  |  | 3 | 0.994 | 0.994 | 0.123 |  |  |  | 3 | 1.000 | 1.000 | 0.147 |  |  |  | 3 | 0.999 | 0.999 | 0.113 |
|  |  |  | 4 | 1.000 | 1.000 | 0.288 |  |  |  | 4 | 1.000 | 1.000 | 0.350 |  |  |  | 4 | 1.000 | 1.000 | 0.312 |
|  |  |  | 5 | 1.000 | 1.000 | 0.572 |  |  |  | 5 | 1.000 | 1.000 | 0.684 |  |  |  | 5 | 1.000 | 1.000 | 0.667 |
|  |  |  | 6 | 1.000 | 1.000 | 0.763 |  |  |  | 6 | 1.000 | 1.000 | 0.863 |  |  |  | 6 | 1.000 | 1.000 | 0.859 |
|  |  |  | 7 | 1.000 | 1.000 | 0.904 |  |  |  | 7 | 1.000 | 1.000 | 0.962 |  |  |  | 7 | 1.000 | 1.000 | 0.968 |
|  |  |  | 8 | 1.000 | 1.000 | 0.993 |  |  |  | 8 | 1.000 | 1.000 | 0.999 |  |  |  | 8 | 1.000 | 1.000 | 1.000 |
|  |  |  | 9 | 1.000 | 1.000 | 0.997 |  |  |  | 9 | 1.000 | 1.000 | 1.000 |  |  |  | 9 | 1.000 | 1.000 | 1.000 |
| 50 | 20 | 0.5 | 0 | 0.070 | 0.050 | 0.050 | 100 | 20 | 0.4 | 0 | 0.051 | 0.044 | 0.050 | 200 | 20 | 0.3 | 0 | 0.038 | 0.043 | 0.050 |
|  |  |  | 1 | 0.722 | 0.651 | 0.021 |  |  |  | 1 | 0.824 | 0.790 | 0.033 |  |  |  | 1 | 0.861 | 0.854 | 0.039 |
|  |  |  | 2 | 0.950 | 0.932 | 0.047 |  |  |  | 2 | 0.982 | 0.977 | 0.061 |  |  |  | 2 | 0.986 | 0.986 | 0.075 |
|  |  |  | 3 | 0.991 | 0.987 | 0.084 |  |  |  | 3 | 0.999 | 0.999 | 0.117 |  |  |  | 3 | 0.998 | 0.998 | 0.171 |
|  |  |  | 4 | 0.995 | 0.994 | 0.173 |  |  |  | 4 | 1.000 | 1.000 | 0.240 |  |  |  | 4 | 1.000 | 1.000 | 0.334 |
|  |  |  | 5 | 1.000 | 1.000 | 0.300 |  |  |  | 5 | 1.000 | 1.000 | 0.445 |  |  |  | 5 | 1.000 | 1.000 | 0.591 |
|  |  |  | 6 | 1.000 | 1.000 | 0.457 |  |  |  | 6 | 1.000 | 1.000 | 0.648 |  |  |  | 6 | 1.000 | 1.000 | 0.785 |
|  |  |  | 7 | 1.000 | 1.000 | 0.659 |  |  |  | 7 | 1.000 | 1.000 | 0.849 |  |  |  | 7 | 1.000 | 1.000 | 0.929 |
|  |  |  | 8 | 1.000 | 1.000 | 0.859 |  |  |  | 8 | 1.000 | 1.000 | 0.972 |  |  |  | 8 | 1.000 | 1.000 | 0.994 |
|  |  |  | 9 | 1.000 | 1.000 | 0.944 |  |  |  | 9 | 1.000 | 1.000 | 0.992 |  |  |  | 9 | 1.000 | 1.000 | 1.000 |
| 50 | 50 | 0.5 | 0 | 0.071 | 0.048 | 0.050 | 100 | 50 | 0.4 | 0 | 0.074 | 0.044 | 0.050 | 200 | 50 | 0.3 | 0 | 0.045 | 0.039 | 0.050 |
|  |  |  | 1 | 0.543 | 0.466 | 0.036 |  |  |  | 1 | 0.757 | 0.625 | 0.049 |  |  |  | 1 | 0.794 | 0.708 | 0.037 |
|  |  |  | 2 | 0.790 | 0.826 | 0.057 |  |  |  | 2 | 0.945 | 0.917 | 0.084 |  |  |  | 2 | 0.966 | 0.964 | 0.068 |
|  |  |  | 3 | 0.881 | 0.944 | 0.087 |  |  |  | 3 | 0.995 | 0.991 | 0.145 |  |  |  | 3 | 0.996 | 0.996 | 0.124 |
|  |  |  | 4 | 0.938 | 0.993 | 0.146 |  |  |  | 4 | 1.000 | 1.000 | 0.226 |  |  |  | 4 | 1.000 | 1.000 | 0.221 |
|  |  |  | 5 | 0.969 | 0.998 | 0.241 |  |  |  | 5 | 1.000 | 1.000 | 0.350 |  |  |  | 5 | 1.000 | 1.000 | 0.363 |
|  |  |  | 6 | 0.982 | 1.000 | 0.340 |  |  |  | 6 | 1.000 | 1.000 | 0.481 |  |  |  | 6 | 1.000 | 1.000 | 0.504 |
|  |  |  | 7 | 0.993 | 1.000 | 0.423 |  |  |  | 7 | 1.000 | 1.000 | 0.618 |  |  |  | 7 | 1.000 | 1.000 | 0.653 |
|  |  |  | 8 | 0.998 | 1.000 | 0.591 |  |  |  | 8 | 1.000 | 1.000 | 0.797 |  |  |  | 8 | 1.000 | 1.000 | 0.815 |
|  |  |  | 9 | 0.999 | 1.000 | 0.699 |  |  |  | 9 | 1.000 | 1.000 | 0.882 |  |  |  | 9 | 1.000 | 1.000 | 0.902 |
| 50 | 100 | 0.5 | 0 | 0.065 | 0.048 | 0.050 | 100 | 100 | 0.4 | 0 | 0.061 | 0.045 | 0.050 | 200 | 100 | 0.3 | 0 | 0.055 | 0.038 | 0.050 |
|  |  |  | 1 | 0.532 | 0.376 | 0.051 |  |  |  | 1 | 0.617 | 0.476 | 0.049 |  |  |  | 1 | 0.718 | 0.549 | 0.044 |
|  |  |  | 2 | 0.739 | 0.705 | 0.077 |  |  |  | 2 | 0.833 | 0.844 | 0.068 |  |  |  | 2 | 0.923 | 0.875 | 0.073 |
|  |  |  | 3 | 0.827 | 0.875 | 0.101 |  |  |  | 3 | 0.913 | 0.952 | 0.099 |  |  |  | 3 | 0.978 | 0.970 | 0.116 |
|  |  |  | 4 | 0.895 | 0.949 | 0.138 |  |  |  | 4 | 0.945 | 0.992 | 0.144 |  |  |  | 4 | 0.997 | 0.994 | 0.182 |
|  |  |  | 5 | 0.944 | 0.987 | 0.189 |  |  |  | 5 | 0.976 | 0.998 | 0.211 |  |  |  | 5 | 1.000 | 0.999 | 0.269 |
|  |  |  | 6 | 0.963 | 0.996 | 0.245 |  |  |  | 6 | 0.988 | 1.000 | 0.284 |  |  |  | 6 | 1.000 | 1.000 | 0.356 |
|  |  |  | 7 | 0.981 | 0.999 | 0.318 |  |  |  | 7 | 0.991 | 1.000 | 0.366 |  |  |  | 7 | 1.000 | 1.000 | 0.454 |
|  |  |  | 8 | 0.992 | 1.000 | 0.434 |  |  |  | 8 | 0.997 | 1.000 | 0.506 |  |  |  | 8 | 1.000 | 1.000 | 0.638 |
|  |  |  | 9 | 0.996 | 1.000 | 0.504 |  |  |  | 9 | 0.999 | 1.000 | 0.598 |  |  |  | 9 | 1.000 | 1.000 | 0.743 |
| 50 | 200 | 0.5 | 0 | 0.056 | 0.049 | 0.050 | 100 | 200 | 0.4 | 0 | 0.057 | 0.048 | 0.050 | 200 | 200 | 0.3 | 0 | 0.052 | 0.041 | 0.050 |
|  |  |  | 1 | 0.430 | 0.264 | 0.048 |  |  |  | 1 | 0.566 | 0.350 | 0.042 |  |  |  | 1 | 0.610 | 0.384 | 0.046 |
|  |  |  | 2 | 0.630 | 0.536 | 0.056 |  |  |  | 2 | 0.772 | 0.692 | 0.050 |  |  |  | 2 | 0.803 | 0.730 | 0.064 |
|  |  |  | 3 | 0.746 | 0.754 | 0.078 |  |  |  | 3 | 0.859 | 0.869 | 0.069 |  |  |  | 3 | 0.879 | 0.899 | 0.092 |
|  |  |  | 4 | 0.820 | 0.878 | 0.091 |  |  |  | 4 | 0.921 | 0.959 | 0.095 |  |  |  | 4 | 0.925 | 0.974 | 0.123 |
|  |  |  | 5 | 0.898 | 0.949 | 0.116 |  |  |  | 5 | 0.958 | 0.991 | 0.137 |  |  |  | 5 | 0.967 | 0.993 | 0.172 |
|  |  |  | 6 | 0.930 | 0.975 | 0.156 |  |  |  | 6 | 0.967 | 0.996 | 0.198 |  |  |  | 6 | 0.975 | 1.000 | 0.227 |
|  |  |  | 7 | 0.955 | 0.989 | 0.215 |  |  |  | 7 | 0.983 | 0.999 | 0.271 |  |  |  | 7 | 0.987 | 1.000 | 0.306 |
|  |  |  | 8 | 0.977 | 0.999 | 0.292 |  |  |  | 8 | 0.995 | 1.000 | 0.355 |  |  |  | 8 | 0.996 | 1.000 | 0.429 |
|  |  |  | 9 | 0.983 | 0.999 | 0.340 |  |  |  | 9 | 0.998 | 1.000 | 0.419 |  |  |  | 9 | 0.998 | 1.000 | 0.528 |
| 50 | 500 | 0.5 | 0 | 0.059 | 0.050 | 0.050 | 100 | 500 | 0.4 | 0 | 0.057 | 0.049 | 0.050 | 200 | 500 | 0.3 | 0 | 0.052 | 0.041 | 0.050 |
|  |  |  | 1 | 0.407 | 0.198 | 0.045 |  |  |  | 1 | 0.515 | 0.250 | 0.049 |  |  |  | 1 | 0.514 | 0.232 | 0.049 |
|  |  |  | 2 | 0.598 | 0.373 | 0.052 |  |  |  | 2 | 0.728 | 0.521 | 0.052 |  |  |  | 2 | 0.725 | 0.503 | 0.052 |
|  |  |  | 3 | 0.703 | 0.526 | 0.059 |  |  |  | 3 | 0.820 | 0.719 | 0.058 |  |  |  | 3 | 0.814 | 0.703 | 0.066 |
|  |  |  | 4 | 0.787 | 0.675 | 0.067 |  |  |  | 4 | 0.877 | 0.847 | 0.072 |  |  |  | 4 | 0.871 | 0.856 | 0.086 |
|  |  |  | 5 | 0.853 | 0.797 | 0.078 |  |  |  | 5 | 0.885 | 0.886 | 0.096 |  |  |  | 5 | 0.932 | 0.950 | 0.114 |
|  |  |  | 6 | 0.893 | 0.872 | 0.091 |  |  |  | 6 | 0.919 | 0.945 | 0.113 |  |  |  | 6 | 0.949 | 0.977 | 0.140 |
|  |  |  | 7 | 0.925 | 0.937 | 0.111 |  |  |  | 7 | 0.952 | 0.981 | 0.135 |  |  |  | 7 | 0.971 | 0.994 | 0.180 |
|  |  |  | 8 | 0.958 | 0.978 | 0.145 |  |  |  | 8 | 0.977 | 0.991 | 0.173 |  |  |  | 8 | 0.986 | 0.999 | 0.225 |
|  |  |  | 9 | 0.966 | 0.988 | 0.170 |  |  |  | 9 | 0.980 | 0.996 | 0.206 |  |  |  | 9 | 0.988 | 1.000 | 0.272 |

## C.2. Computational Costs

To provide an empirical assessment of the computational cost, we report in Table S.6 the median implementation time per replication, computed over 100 runs. As shown in the table, when $N < T - K$, PE-PGCT adds negligible computational cost relative to the pivotal test DH, since computing $J_{PE}$ requires little additional effort once the individual statistics $W_i$ have been obtained for DH. When $N \geq T - K$, PE-PGCT incurs extra time to compute $J_{PE}$; however, the added complexity scales linearly with $N$.

*Table S.6.* Median time (in seconds) per replication (over 100 runs).

| | $T = 50$ | | | | $T = 100$ | | | | $T = 200$ | | | |
|---|---|---|---|---|---|---|---|---|---|---|---|---|
| $N$ | DH | HPJ | $J_{PE}$ | PE-PGCT | DH | HPJ | $J_{PE}$ | PE-PGCT | DH | HPJ | $J_{PE}$ | PE-PGCT |
| 5 | 0.009 | 0.002 | $1.3 \times 10^{-5}$ | 0.009 | 0.010 | 0.003 | $1.3 \times 10^{-5}$ | 0.010 | 0.011 | 0.005 | $1.4 \times 10^{-5}$ | 0.011 |
| 10 | 0.019 | 0.003 | $1.4 \times 10^{-5}$ | 0.019 | 0.018 | 0.005 | $1.4 \times 10^{-5}$ | 0.018 | 0.021 | 0.011 | $1.5 \times 10^{-5}$ | 0.021 |
| 20 | 0.035 | 0.007 | $1.4 \times 10^{-5}$ | 0.035 | 0.038 | 0.012 | $1.5 \times 10^{-5}$ | 0.038 | 0.044 | 0.027 | $1.6 \times 10^{-5}$ | 0.044 |
| 50 | 0.079 | 0.021 | 0.078 | 0.099 | 0.088 | 0.035 | $1.6 \times 10^{-5}$ | 0.088 | 0.102 | 0.072 | $1.6 \times 10^{-5}$ | 0.102 |
| 100 | 0.160 | 0.043 | 0.158 | 0.201 | 0.175 | 0.079 | 0.172 | 0.252 | 0.203 | 0.170 | $1.7 \times 10^{-5}$ | 0.203 |
| 200 | 0.316 | 0.111 | 0.313 | 0.424 | 0.346 | 0.214 | 0.344 | 0.558 | 0.405 | 0.420 | 0.403 | 0.824 |
| 500 | 0.785 | 0.436 | 0.785 | 1.221 | 0.864 | 0.818 | 0.868 | 1.687 | 1.040 | 1.747 | 1.039 | 2.786 |

Note: $J_{PE}$ reports the computation time for the PE component only,
whereas DH, HPJ, and PE-PGCT report the total runtime of their respective tests.

## C.3. Comparisons with Additional Relevant Approaches

We consider comparisons with two additional benchmarks: (i) a max-form test, whose asymptotic distribution for panel Granger-causality testing follows from Proposition 3.5 as detailed in Remark 3.6, and (ii) the method of Fan et al. (2015).

### C.3.1. A Standalone Max Test based on $M_{Wald}$

Proposition 3.5 can also be used to derive the asymptotic distribution of the max-form statistic $M_{Wald} := \max_{1 \leq i \leq N} W_i$, which may itself serve as a standalone testing procedure. Under the same conditions of Proposition 3.5 and suppose $\log N = o(T^{1/7})$ with small and finite $K$, we have, under the null hypothesis $H_0$, $P(M_{Wald} - 2 \log N - (K-2) \log \log N \leq x) \to \exp(-\exp(-x/2))$ as $N, T \to \infty$.

### C.3.2. The Power Enhancement Method of Fan et al. (2015)

Since Fan et al. (2015) considered a different testing problem, we first adapt it to the panel Granger causality framework to enable direct comparisons. Following the key insights of the paper, a power enhancement component can be constructed as the sum of marginal statistics whose absolute values exceed a given threshold. Here,

$$\widetilde{J}_{PE} = \sqrt{NK} \sum_{i=1}^{N} \sum_{j=1}^{K} \frac{|\widehat{\beta}_{ij}|}{\sqrt{\widehat{var}(\widehat{\beta}_{ij})}} I \left\{ \frac{|\widehat{\beta}_{ij}|}{\sqrt{\widehat{var}(\widehat{\beta}_{ij})}} > \widetilde{\delta}_{N,T} \right\}.$$

There are a few challenges in determining an appropriate value $\widetilde{\delta}_{N,T}$ in this setting. First, by design, $\widetilde{\delta}_{N,T}$ must dominate the maximum noise level $\max_{1 \leq i \leq N} \max_{1 \leq j \leq K} |\widehat{\beta}_{ij}| / \sqrt{\widehat{var}(\widehat{\beta}_{ij})}$ under $H_0$. Even under the assumption of cross-sectional independence across $i$, accounting for the dependence across lags $j$ remains difficult, complicating the determination of a theoretically justified threshold. Second, extending Fan et al. (2015), we expect $\widetilde{\delta}_{N,T}$ to take the form of

$$\widetilde{\delta}_{N,T} = \zeta \cdot \sqrt{\log(NK)} \cdot \log \log T \tag{S.49}$$

for some constant $\zeta > 0$. Simulations show that the empirical performance of this approach is highly sensitive to the value of $\zeta$, making it challenging to ensure reliable finite-sample performance.

### C.3.3. EMPIRICAL COMPARISONS

We adopt the same data-generating process as in Section 4. Table S.7 summarizes the comparison among PE-PGCT, MAX, and Fan2015 in terms of their empirical Type I error rates. The results show that the MAX test maintains proper type I error rate control when $T$ is moderate or large and $N$ is large. When $T$ is small but $N$ is large, the test exhibits a slight inflation in Type I error. We also observe that it tends to be conservative when $N$ is small.

For power assessment, we again consider two scenarios: (i) a sparse alternative with $N_0 = 1$ and $\beta = 0.5$, and (ii) a relatively denser alternative with $N_0 = 5$ and $\beta = 0.3$. The comparisons of empirical power are reported in Table S.8. The MAX test performs particularly well under sparse alternatives, but is less effective under dense alternatives especially when $N$ is small. This aligns with the intuition that the maximum statistic is sensitive to strong marginal signals, making it well-suited for detecting sparse signals. However, it may lose power in dense settings where signals are weak but spread across many coordinates. Comparing PE-PGCT with Fan2015, PE-PGCT slightly underperforms Fan2015 when $T$ is small, but outperforms it when $T$ is moderate or large, particularly in settings with a large $N$. Overall, the two deliver comparable performance. Methodologically, the proposed PE-PGCT offers several advantages in addressing the aforementioned challenges of threshold determination. It not only provides theoretically rigorous guarantees but also delivers tuning-free, practical recommendations for strong finite-sample performance.

*Table S.7.* Additional comparisons of empirical Type I error under the Gaussian error setting (nominal level $\alpha = 0.05$).

| $T$ | $N$ | PE-PGCT | MAX | $\widetilde{J}_{PE}$ | $T$ | $N$ | PE-PGCT | MAX | $\widetilde{J}_{PE}$ | $T$ | $N$ | PE-PGCT | MAX | $\widetilde{J}_{PE}$ |
|---|---|---|---|---|---|---|---|---|---|---|---|---|---|---|
| 50 | 5 | 0.060 | 0.026 | 0.085 | 100 | 5 | 0.044 | 0.019 | 0.048 | 200 | 5 | 0.061 | 0.025 | 0.061 |
| | 10 | 0.052 | 0.035 | 0.075 | | 10 | 0.041 | 0.035 | 0.042 | | 10 | 0.052 | 0.024 | 0.052 |
| | 20 | 0.062 | 0.049 | 0.069 | | 20 | 0.056 | 0.043 | 0.056 | | 20 | 0.060 | 0.039 | 0.060 |
| | 50 | 0.074 | 0.063 | 0.082 | | 50 | 0.053 | 0.039 | 0.053 | | 50 | 0.065 | 0.037 | 0.065 |
| | 100 | 0.072 | 0.081 | 0.077 | | 100 | 0.050 | 0.038 | 0.050 | | 100 | 0.059 | 0.030 | 0.059 |
| | 200 | 0.069 | 0.083 | 0.073 | | 200 | 0.056 | 0.059 | 0.055 | | 200 | 0.057 | 0.038 | 0.057 |
| | 500 | 0.052 | 0.129 | 0.055 | | 500 | 0.052 | 0.064 | 0.052 | | 500 | 0.053 | 0.033 | 0.053 |

Note: For the implementation of $\widetilde{J}_{PE}$, we set $\zeta = 1.5$ in (S.49).

*Table S.8.* Additional comparisons of empirical power under the Gaussian error setting (nominal level $\alpha = 0.05$).

| | | | | | (i) (a sparse alternative) $N_0 = 1$, $\beta = 0.5$ | | | | | | | | | |
|---|---|---|---|---|---|---|---|---|---|---|---|---|---|---|
| $T$ | $N$ | PE-PGCT | MAX | $\widetilde{J}_{PE}$ | $T$ | $N$ | PE-PGCT | MAX | $\widetilde{J}_{PE}$ | $T$ | $N$ | PE-PGCT | MAX | $\widetilde{J}_{PE}$ |
| 50 | 5 | 0.813 | 0.791 | 0.843 | 100 | 5 | 0.985 | 0.980 | 0.989 | 200 | 5 | 0.999 | 0.999 | 0.999 |
| | 10 | 0.742 | 0.764 | 0.779 | | 10 | 0.958 | 0.965 | 0.964 | | 10 | 0.999 | 0.999 | 0.999 |
| | 20 | 0.666 | 0.727 | 0.715 | | 20 | 0.925 | 0.962 | 0.925 | | 20 | 0.998 | 1 | 0.997 |
| | 50 | 0.457 | 0.671 | 0.547 | | 50 | 0.873 | 0.940 | 0.862 | | 50 | 0.993 | 0.999 | 0.984 |
| | 100 | 0.405 | 0.680 | 0.476 | | 100 | 0.799 | 0.928 | 0.755 | | 100 | 0.986 | 1 | 0.967 |
| | 200 | 0.334 | 0.604 | 0.384 | | 200 | 0.711 | 0.916 | 0.636 | | 200 | 0.956 | 0.995 | 0.915 |
| | 500 | 0.304 | 0.590 | 0.331 | | 500 | 0.670 | 0.906 | 0.579 | | 500 | 0.943 | 0.991 | 0.879 |
| | | | | | (ii) (a dense alternative) $N_0 = 5$, $\beta = 0.3$ | | | | | | | | | |
| $T$ | $N$ | PE-PGCT | MAX | $\widetilde{J}_{PE}$ | $T$ | $N$ | PE-PGCT | MAX | $\widetilde{J}_{PE}$ | $T$ | $N$ | PE-PGCT | MAX | $\widetilde{J}_{PE}$ |
| 50 | 5 | 0.988 | 0.849 | 0.935 | 100 | 5 | 1 | 0.994 | 1 | 200 | 5 | 1 | 1 | 1 |
| | 10 | 0.959 | 0.800 | 0.926 | | 10 | 1 | 0.989 | 1 | | 10 | 1 | 1 | 1 |
| | 20 | 0.910 | 0.755 | 0.899 | | 20 | 0.998 | 0.981 | 0.998 | | 20 | 1 | 1 | 1 |
| | 50 | 0.401 | 0.648 | 0.463 | | 50 | 0.985 | 0.948 | 0.985 | | 50 | 0.998 | 0.997 | 0.997 |
| | 100 | 0.294 | 0.564 | 0.345 | | 100 | 0.669 | 0.927 | 0.601 | | 100 | 1 | 1 | 1 |
| | 200 | 0.218 | 0.507 | 0.238 | | 200 | 0.554 | 0.891 | 0.424 | | 200 | 0.925 | 1 | 0.831 |
| | 500 | 0.147 | 0.455 | 0.161 | | 500 | 0.418 | 0.851 | 0.284 | | 500 | 0.886 | 0.999 | 0.674 |

Note: Given the practical challenge of selecting an appropriate threshold value in (S.49), we determine the cutoff for $\widetilde{J}_{PE}$ empirically and report its size-adjusted power to ensure a fair comparison.

## D. Supplemental Data Documentation for Section 5

Table S.9 reports the summary statistics for the 220 small banks and 80 large banks. Table S.10 provides detailed information about the CERTs and bank names of the $N = 300$ financial institutions used in the real data analysis in Section 5.

*Table S.9.* Summary Statistics of Bank Size by Group

| Group | $N$ | Mean | SD | Min | Mdn | Max |
|-------|-----|------|----|-----|-----|-----|
| Small | 220 | 10.455 | 0.701 | 8.726 | 10.477 | 11.630 |
| Large | 80 | 12.833 | 1.400 | 11.654 | 12.466 | 18.305 |
| All | 300 | 11.089 | 1.410 | 8.726 | 10.888 | 18.305 |

*Table S.10.* CERT Identifiers and Bank Names for the $N = 300$ Banks

| CERT | NAME | CERT | NAME | CERT | NAME |
|------|------|------|------|------|------|
| 58 | UNITED BANK | 77 | BANKSOUTH | 89 | SIGNATURE BANK OF ARKANSAS |
| 99 | BANK OF DELIGHT | 197 | ALLIANCE BANK | 275 | MORGANTOWN BANK&TRUST CO INC |
| 659 | FIVE STAR BANK | 803 | WESBANCO BANK INC | 899 | GERBER STATE BANK |
| 976 | BLISSFIELD STATE BANK | 1049 | FIRST NB OF STEELEVILLE | 1117 | AMERICAN BANK&TRUST |
| 1208 | GUARANTY BANK&TRUST N A | 1219 | WELLINGTON STATE BANK | 1391 | ARLINGTON STATE BANK |
| 1413 | PERENNIAL BANK | 1468 | BANK OF CROCKETT | 1536 | BANK OF STOCKTON |
| 1545 | FIRST NB AMES IOWA | 1562 | PEOPLES BANK | 1622 | FIRST SECURITY BANK |
| 1686 | FARMERS STATE BANK | 1745 | FARMERS&MERCHANTS BANK | 1833 | STATE BANK |
| 1865 | KANSAS STB OVERBROOK KANSAS | 1900 | FIRST INDEPENDENT BANK | 2160 | FIRST NB OF WAYNESBORO |
| 2301 | FIRST STATE BANK | 2315 | ARMSTRONG BANK | 2448 | PIONEER COMMUNITY BANK INC |
| 2608 | NATIONAL GRAND BK MARBLEHEAD | 2682 | FIRST NB OF BROOKSVILLE | 2754 | FARMERS NB OF LEBANON |
| 2767 | FIRST NB OF RUSSELL SPRINGS | 2784 | FIRST BANK OF BOAZ | 3034 | FIRST NATIONAL BANK COLORADO |
| 3099 | FIRST NB OF BELLVILLE | 3183 | NEWFIRST NATIONAL BANK | 3232 | GRANGER NATIONAL BANK |
| 3276 | AUSTIN BANK TEXAS NA | 3293 | FIRST TEXAS BANK | 3301 | FIRST LIBERTY BANK |
| 3376 | OZONA BANK | 3657 | FIRST AMERICAN BANK | 3664 | INB NATIONAL ASSN |
| 3747 | FIRST NEIGHBOR BANK NA | 3887 | FIRST NATIONAL BANK | 4222 | FIRST FARMERS NB OF WAURIKA |
| 4255 | CAMDEN NATIONAL BANK | 4297 | CAPITAL ONE NATIONAL ASSN | 4444 | FIRST NB IN CRESTON |
| 4477 | FIRST NB OF MANNING | 4506 | SECURITY NB OF SIOUX CITY IA | 4636 | FIRST HERITAGE BANK |
| 4770 | ELEVATE BANK NATIONAL ASSN | 4832 | SHORE UNITED BANK NA | 4865 | SANDY SPRING BANK |
| 4958 | FIRST NB OF ONEIDA | 5058 | SUPERIOR NATIONAL BANK | 5197 | COMMUNITY FIRST BANK |
| 5198 | FIRST NB OF MILACA | 5280 | WNB FINANCIAL N A | 5432 | FIRST NORTHEAST BANK OF NE |
| 5434 | MNB BANK | 5538 | FIRST NB OF STERLING CITY | 5589 | FANNIN BANK |
| 5628 | PIGGOTT STATE BANK | 5632 | LOGAN COUNTY BANK | 5633 | FIRST SECURITY BANK |
| 5686 | BANK OF EDISON | 5716 | FARMERS&MERCHANTS BANK | 5730 | STATE BANK OF ST JACOB |
| 5892 | KALAMAZOO COUNTY STATE BANK | 5956 | PEOPLES BANK | 6180 | BANK OF MONROE |
| 6194 | CITIZENS STATE BANK | 6354 | FIRST HOPE BK A NATL BK ASSN | 6384 | PNC BANK NATIONAL ASSN |
| 6548 | U S BANK NATIONAL ASSN | 6584 | COMMUNITY FIRST BANK NA | 6623 | LCNB NATIONAL BANK |
| 6821 | NATIONAL BANK OF BLACKSBURG | 6917 | FIRST BANK | 6954 | CAYUGA LAKE NATIONAL BANK |
| 6959 | BALLSTON SPA NATIONAL BANK | 7085 | FIRST NB OF GROTON | 7799 | CITIZENS&NORTHERN BANK |
| 7888 | FIRST NB OF PENNSYLVANIA | 7910 | MARS BANK | 8214 | FARMERS STATE BANK OF HAMEL |
| 8276 | ALTON BANK | 8317 | FIRST STATE BANK OF FORSYTH | 8499 | UNION STATE BANK |
| 8594 | MONTROSE SAVINGS BANK | 8745 | FIRST STB OF MIDDLEBURY | 8811 | HEBRON SAVINGS BANK |
| 8840 | PEOPLES STATE BANK OF WELLS | 8866 | MERCHANTS BANK NATIONAL ASSN | 8904 | NODAWAY VALLEY BANK |
| 8980 | SECURITY STB OF OKLAHOMA | 9099 | SOUTH OTTUMWA SAVINGS BANK | 9244 | GLENNVILLE BANK |
| 9322 | CROSS KEYS BANK | 9340 | FARMERS&MERCHANTS STB NY MIL | 9385 | BANK OF BROOKFIELD PURDIN NA |
| 9620 | FIRST BANK | 9632 | FORTRESS BANK | 9871 | ST HENRY BANK |
| 9889 | MID PENN BANK | 9932 | COMMERCIAL STB OF WAGNER | 9978 | FIRST STATE BANK OF UVALDE |
| 9998 | CAPON VALLEY BANK | 10011 | WOODFORD STATE BANK | 10012 | PORTAGE COUNTY BANK |
| 10159 | FIRST STATE BANK&TRUST | 10185 | ELYSIAN BANK | 10248 | FIRST INTERNATIONAL B&T |
| 10315 | BANK OF HALLS | 10363 | CITIZENS&FARMERS BANK | 10438 | RARITAN STATE BANK |
| 10492 | FARMERS&MERCHANTS SB | 10533 | WINNSBORO STATE B&T CO | 10595 | COMMUNITY FIRST BANK |
| 10672 | EXCHANGE BANK | 10709 | CROSSROADS BANK | 10719 | PEOPLES STB OF HALLETTSVILLE |
| 10788 | ILLINI STATE BANK | 10831 | FIRST T&SB OF WATSEKA IL | 10886 | LEGENCE BANK |
| 10973 | ALLIANCE BANK | 11002 | ST CLAIR COUNTY STATE BANK | 11063 | FIRST-CITIZENS BANK&TRUST CO |
| 11124 | S&T BANK | 11352 | SANBORN SAVINGS BANK | 11358 | CUMBERLAND SECURITY BANK INC |
| 11446 | UNITY BANK OF MISSISSIPPI | 11475 | THAYER COUNTY BANK | 11477 | AUBURN STATE BANK |
| 11492 | INBANK | 11527 | FIRST STB OF POND CREEK OK | 11580 | BANK OF SOUTHSIDE VIRGINIA |

*Table S.10.* CERT Identifiers and Bank Names for the $N = 300$ Banks (continued)

| CERT | NAME | CERT | NAME | CERT | NAME |
|---|---|---|---|---|---|
| 11680 | FARMERS STB OF WESTERN IL | 11686 | STATE BANK OF CHERRY | 11777 | COMMERCIAL BANK |
| 12283 | FIRST STATE BANK | 12309 | CITIZENS BANK NATIONAL ASSN | 12372 | FOWLER STATE BANK |
| 12384 | JERSEY STATE BANK | 12395 | LENA STATE BANK | 12410 | LA SALLE STATE BANK |
| 12422 | SECURITY STATE BANK | 12444 | STATE BANK OF COLD SPRING | 12511 | ALLIANCE BANK |
| 12531 | PEOPLES BANK | 12591 | CONVERSE COUNTY BANK | 12594 | PUEBLO BANK&TRUST CO |
| 12614 | ORIGIN BANK | 12758 | FARMERS STB ALLEN OKLAHOMA | 12858 | FARMERS TRUST&SAVINGS BANK |
| 13109 | JACKSON COUNTY BANK | 13112 | OWEN COUNTY STATE BANK | 13113 | PEOPLES TRUST&SAVINGS BANK |
| 13264 | BANK OF LUXEMBURG | 13297 | SOLVAY BANK | 13339 | STATE BANK&TRUST CO |
| 13397 | BANK OF UTICA | 13413 | KAW VALLEY STATE BANK | 13662 | FARMERS&MERCHANTS STB BLOOMF |
| 13774 | BENTON COUNTY STATE BANK | 13813 | FIRST NB IN PINCKNEYVILLE | 13879 | PEOPLESBANK CODORUS VALLEY |
| 13883 | FARMERS STATE BANK | 13951 | COMMUNITY BANK OF LOUISIANA | 14017 | MIDDLETOWN VALLEY BANK |
| 14036 | MISSOURI BANK | 14079 | GNB BANK | 14269 | CITIZENS STATE BANK |
| 14359 | JERSEY SHORE STATE BANK | 14392 | FARMERS TRUST&SAVINGS BANK | 14507 | CNB BANK INC |
| 14581 | CLARKSON BANK | 14660 | BANK OF PRAIRIE DU SAC | 14679 | INDUSTRIAL BANK |
| 14692 | PEOPLES BANK | 14695 | MA BANK | 14772 | DANVILLE STATE SAVINGS BANK |
| 14843 | MIDWESTONE BANK | 14853 | FIRST NB AT ST JAMES | 14952 | LYON COUNTY STATE BANK |
| 15010 | COLUMBUS BANK&TRUST CO | 15092 | RICHLAND COUNTY BANK | 15136 | BANK OF WISCONSIN DELLS |
| 15271 | FIRST STATE BANK | 15478 | UNITED VALLEY BANK | 15590 | TAYLOR COUNTY BANK |
| 15593 | IPSWICH STATE BANK | 15651 | STATE BANK&TRUST CO | 15771 | NORTHEAST GEORGIA BANK |
| 15908 | UNION STATE BANK | 15932 | CITIZENS BANK | 15945 | FOREST PARK NB&T CO |
| 15995 | PCSB BANK | 16039 | SAWYER SAVINGS BANK | 16075 | RAYNE STATE BANK&TRUST CO |
| 16161 | CAMPBELL COUNTY BANK INC | 16177 | FIRST STATE BANK | 16231 | DAKOTA HERITAGE BANK |
| 16413 | BANK OF STRONGHURST | 16600 | FIRST STATE BANK | 16723 | CAROLINA BANK&TRUST CO |
| 16840 | RIVER FALLS STATE BANK | 16895 | FIRST STATE BANK OF BEDIAS | 16908 | SPIRO STATE BANK |
| 17145 | HERITAGE BANK | 17230 | ALBANY B&T CO NA | 17350 | FIRST STB OF SAN DIEGO |
| 17491 | PLAINSCAPITAL BANK | 17513 | FISHER NATIONAL BANK | 17534 | KEYBANK NATIONAL ASSN |
| 17741 | MAINE COMMUNITY BANK | 17749 | BATH SAVINGS INSTITUTION | 17769 | BANK OF HAYS |
| 17811 | FIRST NB IN PORT LAVACA | 17832 | KAW VALLEY STATE B&T CO | 17906 | AMERICAN BANK&TRUST CO |
| 17960 | MERRIMACK COUNTY SB | 18019 | PATRIOTS BANK | 18132 | PEARLAND STATE BANK |
| 18198 | ION BANK | 18258 | THOMASTON SAVINGS BANK | 18384 | NORTHEAST SECURITY BANK |
| 18568 | COMMUNITY BANK OF EASTON | 18677 | VICTORY BANK | 18922 | TRI CITY NATIONAL BANK |
| 18930 | BANK OF BELLE GLADE | 18983 | RIVER CITY BANK | 19101 | HARFORD BANK |
| 19184 | HILLTOP NATIONAL BANK | 19220 | ANB BANK | 19230 | COMMUNITY NB IN MONMOUTH |
| 19333 | REPUBLIC BANK OF CHICAGO | 19396 | UNISON BANK | 19459 | UNITED COMMUNITY BANK |
| 19603 | HOMETOWN BANK NATIONAL ASSN | 19689 | STATE BANK OF WHITTINGTON | 19904 | CENTINEL BANK OF TAOS |
| 20040 | MERCHANTS&MANUFACTURERS BANK | 20215 | BCBANK INC | 20241 | AMERICAN BANK NATIONAL ASSN |
| 20268 | SCALE BANK | 20369 | HEARTLAND BANK&TRUST CO | 21083 | UNITED MISSISSIPPI BANK |
| 21379 | FIRST NATURALSTATE BANK | 21752 | FIRST STATE BANK | 21848 | QUAIL CREEK BANK |
| 22090 | COFFEE COUNTY BANK | 22551 | FIRST TEXAS BANK | 22597 | BANK OF THE SIERRA |
| 22621 | TOWN COUNTRY UNITED BANK | 23007 | OREGON PACIFIC BK DBA OR PAC | 23286 | FALL RIVER FIVE CENTS SB |
| 23293 | FLORENCE BANK | 23620 | DEDHAM INST FOR SVG | 23764 | FMS BANK |
| 23772 | COMMERCE BANK | 24922 | GRAND VALLEY BANK | 25161 | COMMUNITY NB OF OKARCHE |
| 25249 | C3BANK NATIONAL ASSN | 25291 | BANK OF THE VALLEY | 25665 | COMMUNITY NATIONAL BANK |
| 25883 | FIRST EAGLE BANK | 26181 | TEXAS NB OF JACKSONVILLE | 26333 | GENESEE REGIONAL BANK |
| 26451 | FIDELITY COOP BANK | 26590 | ABINGTON BANK | 26602 | CHARLES RIVER BANK |
| 26620 | READING COOP BANK | 26647 | FIRSTRUST SAVINGS BANK | 27847 | PIONEER SAVINGS BANK |
| 28222 | HADDON SAVINGS BANK | 28729 | ASCENDIA BANK | 28747 | HOYNE SAVINGS BANK |
| 28925 | SECURITY SAVINGS BANK | 29009 | LIBERTY BANK FOR SAVINGS | 29093 | INVESTORS COMMUNITY BANK |
| 29809 | FIRST SB OF HEGEWISCH | 30516 | CROSSBRIDGE COMMUNITY BANK | 30585 | MAYVILLE SAVINGS BANK |
| 30619 | ABBEVILLE FIRST BANK SSB | 31936 | FAYETTE SAVINGS BANK SSB | 32135 | SHELBY SAVINGS BANK SSB |
| 32838 | COMMUNITY FIRST BANK INC | 33440 | CITIZENS FIRST BANK | 33503 | UNITY BANK |
| 34040 | FIRST AMERICAN STATE BANK | 34207 | GREAT PLAINS NATIONAL BANK | 34363 | CORE BANK |
| 34422 | AMERICAN BANK | 34666 | COMMUNITY BANK | 34685 | PEOPLES BANK OF COMMERCE |

