# OpenReview forum: "Power-Boosted Granger-Causal Discovery for Large Heterogeneous Panel Data"
_ICML.cc/2026/Conference — ICML 2026 regular_

### Official Review · Reviewer_38mX · 2026-02-19

**Soundness:** 3
**Presentation:** 3
**Significance:** 3
**Originality:** 3
**Overall Recommendation:** 5
**Confidence:** 3

**Summary:**

The paper addresses the problem of testing Granger non-causality in panel data with sparse alternatives and potentially higher cross-sectional dimensions than time steps. The work proposes a method that increases the statistical power of testing for non-causality and evaluates it on synthetic and real-world data.

**Compliance With Llm Reviewing Policy:**

Affirmed.

**Final Justification:**

The authors addressed my questions and I keep my recommendation for acceptance.

**Key Questions For Authors:**

- In Eq. (6), it is unclear where the 1.5 factor comes from. Is this a known result or an arbitrary choice?
- In Proposition 3.6, is a "max" missing here (P(max…))?
- While stationarity is a common assumption, I wonder whether the methodology could be extended to non-stationary settings, aside from removing seasonal patterns beforehand.
- Just a brief remark: the original Granger theory does not necessarily assume Gaussian errors (not even linearity), but most of the developed tests are based on these assumptions.
- In the experiments, I wonder how the method performs when simply the max of the statistic (W) is used.
- Assumption 3.3 states independence across units; how realistic is this in the bank profitability setting? This assumption seems to be violated, but nevertheless, the method appears to show some robustness.
- Tables S.3 and S.5 seem to be identical in the appendix. Or am I missing smaller differences somewhere?

**Limitations:**

The authors fairly discuss and acknowledge limitations due to the required assumptions. These are fairly standard in the related literature, but nevertheless, one would usually expect most of the assumptions to be violated in real data.

**Strengths And Weaknesses:**

Strengths:
- Well motivated and clearly written
- Novel theoretical insights
- Clear introduction of assumptions and fair discussion of their limitations
- Mathematically clear definitions

Weaknesses:
- Lacks an exemplary introduction to the panel setting; requires domain knowledge to understand the setting and problem solution
- Some minor (technical) typos
- Potential violations of assumptions in the real-world data could be discussed more clearly

---

> ### Author Rebuttal · Authors · 2026-03-30
>
> We greatly appreciate your positive feedback and valuable comments.
>
> **Q1.** The constant 1.5 in $C(T) = \max(1.5, \log \log(T) )$ is a practical calibration choice to safeguard the finite-sample performance, aiming to provide a simple, stable, and broadly applicable implementation rule.  Theoretically, any choice of $C(T)> 1$ such that $C(T) \rightarrow \infty$ as $T\rightarrow \infty$(albeit arbitrarily slowly) would suffice for establishing the asymptotic theoretical guarantees. However, in finite samples with small or moderate $T$, $\log\log(T)$ takes small values, potentially leading to under-thresholding and hence distorted Type I error. The lower bound of 1.5 is therefore imposed to ensure a minimum level of thresholding that stabilizes the test performance in small and moderate samples. We'll add a note to clarify this point.
>
> **Q2.** Yes, thanks for pointing out the typo.
>
> **Q3.** Our current framework assumes stationarity, standard in panel Granger causality settings, to ensure valid inference. In practice, non-stationarity can be potentially addressed by data preprocessing. Extending the methodology to non-stationary scenarios is an important direction for future research. In principle, we expect such an extension would require adapting the test to account for common stochastic trends or employing transformations to restore stationarity, yet a thorough investigation shall be left for future work.
>
> **Q4.** We acknowledge that the original Granger causality is defined in a broad sense and does not require Gaussian errors or linearity. In this paper, we focus on the linear panel model framework with standard assumptions to enable theoretically guaranteed inference. Within our framework, we relax the commonly imposed Gaussian error assumption to sub-Gaussian errors,  accommodating a broader class of non-Gaussian settings. In simulations, we further investigate $t_3$ settings, providing some empirical evidence of robust performance beyond the Gaussian and sub-Gaussian errors. Despite the empirical success, rigorously extending the theoretical guarantees to a more general error distribution is substantially more complicated. We leave it for future research.
>
> **Q5.** We'd like to begin by clarifying that, although max-form tests have been studied in panel econometrics and high-dimensional statistics, they have not yet been explored in the understudied area of panel Granger-causality testing. Motivated by your question, we revisit our theoretical results, and find that the results established in this paper, specifically Prop. 3.5, can be used to derive the asymptotic distribution of $\max_{1\leq i \leq N} W_i$ for testing panel Granger-causality. Under the same conditions of Prop.3.5, we can prove, under H0, $P(\max_{1\leq i \leq N} W_i - 2\log p - (K-2) \log\log p\leq x) \rightarrow \exp(-\exp(x/2))$ as $N,T\rightarrow\infty$.  We'll add discussions to note this add-on contribution.
>
> We also add experiments to compare with the MAX test. Due to space limit,  we present partial results.
>
>     Empirical Type I error
>
> T=50, N=5,10,20,50,100,200,500:
>
> PE-PGCT: 0.060,0.052,0.062,0.074,0.072,0.069,0.052
>
> MAX: 0.026,0.035,0.049,0.063,0.081,0.083,0.129
>
> T=100, N=5,10,20,50,100,200,500:
>
> PE-PGCT: 0.044,0.041,0.056,0.053,0.05,0.056,0.052
>
> MAX: 0.019,0.035,0.043,0.039,0.038,0.059,0.064
>
>
>     Empirical power (N0=1, beta=0.5)
>
> T=50, N=5,10,20,50,100,200,500:
>
> PE-PGCT: 0.813,0.742,0.666,0.457,0.405,0.334,0.304
>
> MAX: 0.791,0.764,0.727,0.671,0.68,0.604,0.59
>
> T=100, N=5,10,20,50,100,200,500:
>
> PE-PGCT: 0.985,0.958,0.925,0.873,0.799,0.711,0.67
>
> MAX: 0.98,0.965,0.962,0.94,0.928,0.916,0.906
>
> The results show that MAX maintains proper type I error rate when T is moderate/large and N is large. When T is small but N is large, it exhibits a slight inflation in Type I error. We also observe that it tends to be conservative when N is small. Regarding power, the MAX test performs reasonably well for large $N$ but is less powerful for small $N$.
>
> **Q6.** Assumption 3.3 is imposed to facilitate tractable asymptotic results. We acknowledge that this assumption may be somewhat restrictive in the bank profitability setting. We also note that similar assumptions have been adopted in prior empirical studies on assessing the Granger causality of bank profitability (e.g., Juodis et al., 2021), providing some precedent for this simplification.
>
> Though not reported in the paper, we conducted additional experiments under moderate correlated errors, and the test performance remained empirically robust. Developing a theoretical framework that rigorously accommodates such dependence is an important and nontrivial extension, requiring substantially different technical tools. We'll add a discussion to elaborate on this point and leave it for future work.
>
> **Q7.** Thanks for your careful reading. It is an inadvertent copy-paste error during manuscript preparation. We'll update Table S.5 to the correct version.

---

> > ### Author Rebuttal · Reviewer_38mX · 2026-04-02
> >
> > I want to thank the authors for addressing my questions and keep my recommendation for acceptance!

---

> > > ### Author Response · Authors · 2026-04-08
> > >
> > > Thank you very much for confirming that your concerns have been fully resolved. We sincerely appreciate your insightful comments that helped to improve the paper.

---

### Official Review · Reviewer_FTJG · 2026-03-11

**Soundness:** 2
**Presentation:** 3
**Significance:** 3
**Originality:** 2
**Overall Recommendation:** 5
**Confidence:** 3

**Summary:**

The paper studies statistical testing of Granger causality in heterogeneous panel settings. The authors highlight the limitations of classical panel Granger causality tests when causal effects are sparse, i.e., present in only a small subset of units. To address this issue, they propose a power-enhanced panel Granger causality test (PE-PGCT), which augments an aggregation-based test with a component based on the maximum of individual Wald statistics to better detect rare but strong signals. The method is supported by theoretical analysis showing Type-I error control, no power loss relative to a baseline test, and improved power under sparse alternatives, and is evaluated through simulations and an empirical application.

**Compliance With Llm Reviewing Policy:**

Affirmed.

**Final Justification:**

I am satisfied with the responses provided, which addressed my concerns. I have adjusted my score accordingly.

**Key Questions For Authors:**

- Can the authors provide a direct empirical comparison with the power enhancement procedure of  Fan et al. (2015)?

- Can results be presented with fixed baseline tests, isolating the marginal effect of the enhancement step?

**Limitations:**

yes

**Strengths And Weaknesses:**

**Strengths:**

- Overall, the paper is well written and the methodology and results are presented clearly.
- The problem is well-motivated, and detecting sparse causal signals in heterogeneous panels is an important and practically relevant challenge.
- Conceptually coherent approach: using the maximum of individual statistics to enhance detection under sparsity is natural and aligned with the target regime.
- Theoretical analysis: the paper provides strong theoretical guarantees on size distortion and power enhancement.
- Empirical validation: Both synthetic and real-data experiments are included.

\
**Weaknesses:**

-  Given the strong conceptual overlap with the power enhancement framework of Fan et al. (2015), the absence of a direct empirical comparison is difficult to justify. It is therefore unclear whether the performance gains arise from:  the panel Granger structure, the specific form of the enhancement statistic, or the general power enhancement principle already established in prior work.

- In the current presentation, the role and selection of the pivotal baseline are not fully transparent. In particular, the pivotal statistic appears to be chosen depending on the panel regime (e.g., values of N and T). Strong performance across regimes may therefore partly reflect this selection step rather than the enhancement component itself. I think that the key point is not universal superiority, but rather that the enhancement preserves size while improving power relative to a fixed baseline.

---

> ### Author Rebuttal · Authors · 2026-03-30
>
> We greatly appreciate your positive feedback and helpful suggestions.
>
> **W1.** Though we adopt the concept of ``power enhancement'' to relate to existing literature, the proposed technique introduces new methodological aspects beyond prior works. Directly adapting Fan et al. (2015) to the panel Granger causality test presents several challenges, while our approach overcomes those limitations. Following your suggestion, we add direct empirical comparisons with the method of Fan et al.(2015) to more explicitly demonstrate our contribution and innovation (please see Q1 below).
>
> **W2.** Indeed, the key contribution we wanted to emphasize is not universal superiority, but rather that the enhancement preserves size while improving power relative to a fixed baseline. Importantly, our proposed power enhancement procedure and the theoretical guarantees remain valid for any pivotal test that attains correct asymptotic size. The role of the baseline is therefore conceptually separate from the enhancement step.
>
> To provide a ready-to-use tool for practitioners, we offer recommendations for the choice of the pivotal test in Remark 2.2. This recommendation is based on a careful assessment of the relative strengths and limitations of existing approaches in the literature. As shown in Figure 1 and Table 1, DH performs well when  $N < T-K$, but exhibits severely inflated Type I error when $N \geq T-K$, whereas HPJ shows the opposite pattern. Remark 2.2 is therefore designed to ensure that the baseline remains a valid pivotal test in each regime. We'll add clarifications to justify the recommended choice and improve the transparency of the selection.
> We further clarify the attribution of performance gains in Q2 below.
>
> **Q1.** Since Fan et al.(2015) considered a different testing problem, we first adapt it to the panel Granger causality framework to enable direct comparisons. Following the key insights of the paper, a power enhancement component can be constructed as the sum of marginal statistics whose absolute values exceed a given threshold. Here,
> $\tilde{J}_{PE}= \sqrt{NK} \sum^N\_{i=1} \sum^K\_{j=1} \frac{|\hat\beta\_{ij}|}{\sqrt{\widehat{var}(\hat\beta\_{ij})}} I \left( \frac{|\hat\beta\_{ij}|}{\sqrt{\widehat{var}(\hat\beta\_{ij})}}>\tilde\delta\_{N,T} \right)$
>
> There are a few challenges in determining an appropriate value $\tilde\delta_{N,T}$ in this setting. First, by design, $\tilde\delta_{N,T}$ must dominate the maximum noise level $\max_{1\leq i\leq N} \max_{1\leq j \leq K} |\hat\beta_{ij}|/\sqrt{\widehat{var}(\hat\beta_{ij})}$ under H0. Even under the assumption of cross-sectional independence across $i$, accounting for the dependence across lags $j$ remains difficult, complicating the determination of a theoretically justified threshold. Second, extending Fan et al.(2015), we expect $\tilde\delta_{N,T}$ to take the form of $\zeta \cdot \sqrt{\log (NK)} \cdot \log\log T$ for some constant $\zeta > 0$. Simulations show that its empirical performance is highly sensitive to the value of $\zeta$, making it challenging to ensure reliable finite-sample performance.
> Below, we determine the cutoff empirically and report size-adjusted power for comparison. Due to space limit, we present partial results here.
>
>     Empirical power (N0=1, beta=0.5):
>
>     T=50, N=5, 10, 20, 50, 100, 200, 500:
>
> Proposed: 0.813,0.742,0.666,0.457,0.405,0.334,0.304
>
> Fan2015: 0.843,0.779,0.715,0.547,0.476,0.384,0.331
>
>     T=100, N=5, 10, 20, 50, 100, 200, 500:
>
> Proposed: 0.985,0.958,0.925,0.873,0.799,0.711,0.67
>
> Fan2015: 0.989,0.964,0.925,0.862,0.755,0.636,0.579
>
>     T=200, N=5, 10, 20, 50, 100, 200, 500:
>
> Proposed: 0.999,0.999,0.998,0.993,0.986,0.956,0.943
>
> Fan2015: 0.999,0.999,0.997,0.984,0.967,0.915,0.879
>
> Empirically, the proposed method slightly underperforms Fan2015 when $T$ is small, but slightly outperforms it when $T$ is moderate or large, particularly in settings with a large N. Overall, the two deliver comparable performance.
>
> Methodologically, our method, based on max Wi, offers several advantages in addressing the aforementioned challenges of threshold determination. It not only provides theoretically rigorous guarantees but also delivers tuning-free, practical recommendations for strong finite-sample performance.
>
> **Q2.** The empirical results of DH and HPJ, presented in Section 4, are two fixed, standalone baseline tests that are not combined with the enhancement step. As such, in experiments where $N<T-K$, PE-PGCT directly compares to DH, while when $N \geq T-K$, PE-PGCT directly compares to HPJ. In both cases, the comparisons isolate the performance of the baseline procedures, enabling a direct assessment of the gains from the enhancement step.
>
> It would not be appropriate to consider HPJ as a baseline to construct PE-PGCT when $N<T-K$, as it exhibits severely inflated Type I error in this regime and therefore cannot be regarded as a valid pivotal test. The same argument applies to DH when $N\geq T-K$.

---

> > ### Author Rebuttal · Reviewer_FTJG · 2026-04-02
> >
> > I am satisfied with the responses provided, which addressed my concerns. I have adjusted my score accordingly.

---

> > > ### Author Response · Authors · 2026-04-08
> > >
> > > Thank you very much for confirming that your concerns have been fully resolved. We sincerely appreciate your constructive feedback that helped to improve the paper.

---

### Official Review · Reviewer_64aX · 2026-03-13

**Soundness:** 3
**Presentation:** 4
**Significance:** 2
**Originality:** 3
**Overall Recommendation:** 5
**Confidence:** 3

**Summary:**

This paper studies Granger-causality testing in large, heterogeneous panel time series, focusing on regimes where standard panel Granger tests have distorted size and low power, especially when only a small fraction of units exhibit nonzero predictive effects. The authors propose a “power-enhanced” testing framework that augments a pivotal baseline panel Granger test with an additional component designed to detect sparse signals. They establish asymptotic size control and show that the augmented test achieves power approaching one over a formal sparse-alternative space, with no power loss relative to the baseline test. Simulations across a wide range of $N,T$ configurations (including high-dimensional $N\ge T$) and an application to banking data, illustrate improved detection in sparse settings and competitive performance more broadly.

**Compliance With Llm Reviewing Policy:**

Affirmed.

**Final Justification:**

After comparing this paper with others in my reviewing batch, it stands out for its rigor, completeness, and the authors' highly constructive response. While I maintain my reservation about the broader impact of Granger-causality as an exclusive focus (which is why I do not push for a Spotlight/Oral), I believe such a high standard of quality deserves recognition. Therefore, I have raised my score to Accept to reflect my support for its solid technical contribution.

**Key Questions For Authors:**

1. **Baseline power under sparse alternatives.** The “boosted power” guarantee in Theorem 3.9(iii) (power $\to 1$ over the sparse alternative space $\Theta_\beta$) is convincing. Do you know of prior results that characterize the asymptotic power of the baseline DH / HPJ tests specifically under such sparse alternatives (e.g., whether they are inconsistent, or achieve strictly slower detection rates)? If not already available, did you attempt (even heuristically) to derive their detection boundaries / convergence rates in this regime, to more formally quantify the gap that PE-PGCT closes?
2. **Power under dense alternatives.** Beyond the sparse alternative space $\Theta_\beta$ where Theorem 3.9(iii) gives boosted power, have you considered providing a more refined asymptotic characterization of PE-PGCT under dense alternatives—e.g., a detection boundary in that regime, or a comparison of asymptotic relative efficiency / convergence rate versus DH (and/or HPJ)? This would not affect my score, but I’m curious whether such a characterization is feasible or if there are technical obstacles.
3. **Empirical regime effect ( $T>N$ vs. $T\le N$ ).** In Figure 3, PE-PGCT appears to outperform the baselines quite uniformly when $T>N$, whereas in the $T\le N$ (high-dimensional) regime its relative advantage diminishes as the alternative becomes denser and can even be overtaken by DH (after size-correction). Do you have an explanation for why PE-PGCT empirically looks *stronger* in the $T>N$ regime than in the $T\le N$ regime?

**Limitations:**

The proposed framework is developed for linear dynamic panel models with a fixed lag order $K$ and relies on unit-level Wald statistics, so its validity and practical performance may depend on how well these modeling choices match the data-generating process. It would strengthen the paper to more explicitly discuss potential failure modes in applications with richer dynamics, e.g., nonlinear relationships, time-varying coefficients or structural breaks, instantaneous effects, or stronger forms of cross-sectional dependence than permitted by the assumptions, etc.

**Strengths And Weaknesses:**

**Strength:**

The paper is strongest in how cleanly it frames and addresses a failure mode in large heterogeneous panels: it gives a principled, modular way to boost sensitivity to sparse signals without sacrificing the baseline test’s size in the limit, and backs this up with clear theory that separates null behavior from sparse-alternative behavior. The empirical section is thorough in probing different $N,T$ regimes and error conditions. Finally, the writing is careful about what is being claimed, e.g., consistently phrasing findings in terms of “Granger-cause / Granger non-causality,” which helps avoid overstating causal interpretation.

**Weakness:**

1. **Finite-sample  gap.** While the theory shows $P_{H_0}(J_{PE}=0)\to 1$, in finite samples the PE component can still be occasionally triggered under $H_0$, which may introduce residual size distortions (potentially more noticeable in the large-$N$, small-$T$ regimes the paper targets). The paper would be more convincing with a brief discussion of this finite-sample effect and a simple diagnostic report, e.g., the null triggering rate $\widehat{P}(J_{PE}>0)$ (and/or the distribution of $J_{PE}$ conditional on triggering), alongside the empirical size tables, to make clear how often and how strongly the PE term “fires” when it should not.

---

> ### Author Rebuttal · Authors · 2026-03-30
>
> We greatly appreciate your positive feedback and insightful questions.
>
> **W1.** We'd like to begin by clarifying that our proposed PE-PGCT is designed not only with theoretical rigor in mind, but also with careful consideration of finite-sample performance. To provide a simple, stable, and broadly applicable rule that avoids the need for user tuning,  we specify the threshold as $C(T) = \max (1.5, \log \log(T))$, where the constant 1.5 is a practical calibration choice to safeguard the finite-sample performance.
>
> In simulation studies, the difference in empirical Type-I error rates between PE-PGCT and DH (for $N<T-K$) or HPJ ($N \geq T-K$) is an equivalent measurement of the null triggering rate $\hat P(J_{PE}>0)$. We'll add clarifications to make this connection more explicit for readers and include a more detailed diagnostic report, following your suggestion.
>
> **Q1.** To our knowledge, explicit characterization of the asymptotic power of DH/HPJ  is absent from the existing literature. We are not aware of any work that derives detection boundaries for these tests in either sparse or dense regimes. Below, we attempt to discuss this problem heuristically.
>
> Let $S$ denote the set of Granger-causal units with $s_N = |S|$. We parametrize the sparsity level as $s_N = N^{1-\gamma}$ for $\gamma\in [0,1]$, where $\gamma\rightarrow 1$ corresponds to the extremely sparse case. The Wald statistic $W_i$ behaves approximately as $W_i \sim \chi_K^2(\lambda_i)$ with noncentrality $\lambda_i = T \mu_i^2$, $\mu_i^2 = \beta_i^\top \Sigma_i^{-1}\beta_i$ for $i \in S$, while  $W_i \sim \chi_K^2(0)$ for $i \notin S$. For simplicity, we assume the  active units have comparable signal $\mu_i \approx \mu$ and then $\lambda_i \approx \lambda$, for $i\in S$.
>
> DH: The DH test relies on the averaged Wald statistic $\bar{W} = \sum_{i=1}^N W_i/N$. Under the alternative, the signal-to-noise ratio is $SNR \asymp s_N\lambda/\sqrt{N} \asymp \lambda N^{1/2-\gamma}$. For DH to have non-trivial power, we need $\lambda \gg N^{\gamma-1/2}$, which gives the detection boundary $\mu_{DH} \asymp \frac{N^{\gamma/2-1/4}}{\sqrt{T}}$.
>
> HPJ: The HPJ statistic is a bias-corrected pooled Wald statistic, based on a pooled estimator with $\sqrt{NT}$ convergence. Under the alternative, the pooled Wald signal is $NT \cdot (s_N/N)^2 \mu^2$. The SNR is $s_N^2\lambda/N \asymp \lambda N^{1-2\gamma}$. For HPJ to have non-trivial power, we need $\lambda \gg N^{2\gamma-1}$, which gives the detection boundary $\mu_{HPJ} \asymp \frac{N^{\gamma-1/2}}{\sqrt{T}}$.
>
> PE-PGCT: Theorem 3.7(iii) proves the detection boundary of $J_{PE}$ is $\mu_{J_{PE}} \asymp \frac{\sqrt{\log\log T\log N}}{\sqrt{T}}$. The detection boundary of PE-PGCT is naturally obtained $\mu_{PE-PGCT} = \min(\mu_{pivotal}, \mu_{J_{PE}})$, where $\mu_{pivotal} = \mu_{DH}$ or $\mu_{HPJ}$ depending on the data regime (see Remark 2.2).
>
> In the sparse regime when $\beta > 1/2$, the gaps between DH/HPJ and $J_{PE}$ ($\mu_{DH}/\mu_{J_{PE}}$ and $\mu_{HPJ}/\mu_{J_{PE}}$) diverges polynomially with $N$, showing that DH/HPJ requires much stronger signals than $J_{PE}$.
>
> **Q2.** This is closely connected to the detection boundaries in Q1. The detection boundary of PE-PGCT is obtained $\mu_{PE-PGCT} = \min(\mu_{pivotal}, \mu_{J_{PE}})$, where $\mu_{pivotal} = \mu_{DH}$ or $\mu_{HPJ}$ depending on the data regime. Theorem 3.9 (ii) guarantees that PE-PGCT has no power loss compared to the pivotal test. Hence, by construction, the power characterization of PE-PGCT under dense alternatives is determined by the pivotal test.
>
> In the dense regime with $\beta<1/2$, DH and HPJ are consistent under their respective minimal signal conditions, which further ensures the consistency of PE-PGCT.
>
> **Q3.** This observation may be attributed to multiple reasons.
>
> (a) It is rooted in the design of PE-PGCT. PE-PGCT gains most when (i) the marginal Wald statistics are well-behaved (which requires reasonably large $T$), and (ii) the alternative is sparse. When $T \geq N$ and the alternative becomes dense, both ingredients weaken, so the relative advantage naturally diminishes.
>
> (b) The threshold $\delta_{N,T} = (K+2\sqrt{K \log N} + 2\log N)\cdot C(T)$ in $J_{PE}$ grows with $\log N$. In high-dimensional settings (large $N$ relative to $T$), this leads to a more stringent threshold, making it harder for individual components to exceed it.
>
> (c) The empirical performance of PE-PGCT is also affected by the power of the underlying pivotal test. As elaborated in Remark 2.2, we adopt DH when $T-K > N$ and HPJ when $T-K \leq N$. The results in Figures 2 and 3 indicate that DH exhibits stronger empirical power than HPJ in the designs considered. Therefore, in the large $T$ regime where DH is used, PE-PGCT benefits from a stronger baseline test, whereas in the large $N$ regime, the reliance on HPJ further contributes to the observed weaker performance relatively.

---

> > ### Author Rebuttal · Reviewer_64aX · 2026-04-04
> >
> > Thank you for the thorough and thoughtful rebuttal, and for the clear plan to incorporate additional clarifications/diagnostics in the revision. Overall, I find the work technically strong and well executed, with careful use of terminology and a professional, constructive response.
> >
> > I’m comfortable with acceptance as a solid contribution within the time-series / Granger-testing community. However, my overall significance assessment remains conservative given the limited broader impact of Granger-causality testing as a standalone focus, so I would not prioritize this paper for spotlight/oral. Accordingly, I will keep my score unchanged.

---

> > > ### Author Response · Authors · 2026-04-08
> > >
> > > Thank you very much for confirming that your concerns have been fully resolved. We sincerely appreciate your thoughtful insights that helped to improve the paper.

---

### Official Review · Reviewer_8o1U · 2026-03-13

**Soundness:** 3
**Presentation:** 3
**Significance:** 3
**Originality:** 3
**Overall Recommendation:** 4
**Confidence:** 2

**Summary:**

The paper proposes a Power-Enhanced Panel Granger Causality Test (PE-PGCT) to address the low power of traditional tests (e.g., DH and HPJ) in heterogeneous, high-dimensional panel data, particularly under sparse alternatives. By integrating a screening component based on the maximum of individual Wald statistics and establishing non-asymptotic Kolmogorov distance bounds, the authors aim to boost detection power without sacrificing Type I error control. The method is validated through simulations and an application to US banking data.

**Compliance With Llm Reviewing Policy:**

Affirmed.

**Key Questions For Authors:**

1.How sensitive is the Kolmogorov distance bound to the normality assumption? Would the threshold $\delta_{N,T}$ remain valid under skewed or fat-tailed error distributions?

2.What is the theoretical or empirical justification for the constant 1.5 used in the correction factor $C(T)$? How should practitioners choose this value for a completely new dataset?

**Limitations:**

yes

**Strengths And Weaknesses:**

## Strengths

1.The manuscript is exceptionally well-organized and clearly written. The motivation is articulated effectively, and the transition from theoretical derivation to empirical validation is logical and easy to follow.

2.The paper provides a high-quality and thorough review of existing literature on panel Granger causality and power enhancement techniques. It successfully identifies the limitations of classical tests (like DH and HPJ) in high-dimensional settings, providing a solid context for the proposed work.

3.The research addresses a highly relevant problem in modern econometrics and data science. As high-dimensional panel data becomes more prevalent, the ability to maintain test power under sparsity is a significant contribution to the field of causal discovery.

## **Weaknesses**

1.While the theoretical bounds are well-established, the paper would benefit from a more explicit discussion of computational complexity. Specifically, providing an analysis of how the PE-PGCT scales as $N$ (cross-sectional dimension) and $T$ (time dimension) grow extremely large would be valuable for practitioners.

2.The proposed method involves certain thresholding heuristics. A more detailed sensitivity analysis regarding these parameters would strengthen the paper, ensuring that the performance gains are robust and not overly dependent on specific parameter tuning.

3.The study focuses on a specific set of econometric baselines. Including a wider variety of scenarios or comparing against a broader range of high-dimensional causal discovery methods could provide a more exhaustive evaluation of the method's generality.

---

> ### Author Rebuttal · Authors · 2026-03-30
>
> We greatly appreciate your positive feedback and helpful suggestions.
>
> **W1.** Following your suggestion, we have added an empirical analysis of the computation cost.
>
> Median time (in seconds) per replication (over 100 runs) (JPE reports the time for the PE component only, and PE-PGCT reports the full time.):
>
>     T=50, N=5, 10, 20, 50, 100, 200, 500:
>
> DH: 0.009, 0.019, 0.035, 0.079, 0.160, 0.316, 0.785
>
> HPJ: 0.002, 0.003, 0.007, 0.021, 0.043, 0.111, 0.436
>
> JPE: 1.3e-5, 1.4e-5, 1.4e-5, 0.078, 0.158, 0.313, 0.785
>
> PE-PGCT: 0.009, 0.019, 0.035, 0.099, 0.201, 0.424, 1.221
>
>     T=100, N=5, 10, 20, 50, 100, 200, 500:
>
> DH: 0.010, 0.018, 0.038, 0.088, 0.175, 0.346, 0.864
>
> HPJ: 0.003, 0.005, 0.012, 0.035, 0.079, 0.214, 0.818
>
> JPE: 1.3e-5, 1.4e-5, 1.5e-5, 1.6e-5, 0.172, 0.344, 0.868
>
> PE-PGCT: 0.010, 0.018, 0.038, 0.088, 0.252, 0.558, 1.687
>
>     T=200, N = 5, 10, 20, 50, 100, 200, 500:
>
> DH: 0.011, 0.021, 0.044, 0.102, 0.203, 0.405, 1.040
>
> HPJ: 0.005, 0.011, 0.027, 0.072, 0.170, 0.420, 1.747
>
> JPE: 1.4e-5, 1.5e-5, 1.6e-5, 1.6e-5, 1.7e-5, 0.403, 1.039
>
> PE-PGCT: 0.011, 0.021, 0.044, 0.102, 0.203, 0.824, 2.786
>
> As shown above, when $N < T-K$, PE-PGCT adds negligible computational cost relative to the pivotal test DH, since computing $J_{PE}$ requires little additional effort once the individual statistics $W_i$ are obtained for DH. When $N\geq T-K$, PE-PGCT incurs extra time to compute $J_{PE}$, but the added complexity scales linearly with N.
>
> **W2.** We'd like to clarify that our implementation does not require parameter tuning in practice. Taking into account both theoretical guarantees and finite-sample performance, we specify the threshold as $C(T) = \max (1.5, \log \log(T))$, where the constant 1.5 is a fixed value (please also see Q2 below for further justification). While this choice could in principle be viewed as a hyperparameter, we recommend this value, to provide a simple, stable, and broadly applicable rule that avoids the need for user tuning.
>
>
> **W3.** We appreciate your insightful suggestion to help us better highlight the generality of our proposed method. Indeed, the power enhancement technique we propose in this paper is not limited to testing panel Granger causality, but can be extended to broader high-dimensional inference problems. We will have a note to further discuss its generality and broader applicability.
>
> To present the methodology in a focused and coherent manner, we center on the relatively underexplored problem of panel Granger causality. In this context, we adopt the DH and HPJ procedures as benchmarks, as they are among the most relevant and widely used approaches for this specific setting. Despite a large body of literature on high-dimensional causal discovery methods, many of these methods are designed for different objectives and do not provide directly comparable inference in our framework.
>
> Following suggestions from other reviewers, we add two additional benchmarks: (i) a max test, whose asymptotic distribution in the panel Granger-causality testing follows from Proposition 3.5, and (ii) the method in Fan et al. (2015). Although the latter is not originally developed for this setting, we adapt it to the panel Granger causality framework and evaluate its numerical performance. We will also add discussions of the connections to these approaches and highlight the improvements offered by our method.
>
> **Q1.** Within our framework, the Kolmogorov distance bound remains valid under sub-Gaussian errors, accommodating a broader class of non-Gaussian settings. In simulation studies, we further investigate the fat-tailed $t_3$ settings, providing some empirical evidence of robust performance beyond the Gaussian and sub-Gaussian errors. Despite the empirical success, rigorously extending the theoretical guarantees to more general skewed or fat-tailed error distributions is substantially more complicated, and we shall leave it for future research.
>
>
> **Q2.** The constant 1.5 in $C(T) = \max (1.5, \log\log(T) )$ is a practical calibration choice to safeguard the finite-sample performance, aiming to provide a simple, stable, and broadly applicable implementation rule. Theoretically, any choice of $C(T)> 1$ such that $C(T) \rightarrow \infty$ as $T\rightarrow \infty$(albeit arbitrarily slowly) would suffice for establishing the asymptotic theoretical guarantees. However, in finite samples with small or moderate $T$, $\log\log(T)$ takes small values, potentially leading to under-thresholding and hence size distortions. The lower bound of 1.5 is therefore imposed to ensure a minimum level of thresholding that stabilizes the test performance in small and moderate samples.
>
> For new datasets, the proposed $C(T) = \max (1.5, \log\log(T) )$ still works well. In fact, we recommend using the default value, rather than treating it as a tuning parameter.  We will add a note to clarify these points.

---

> > ### Author Rebuttal · Reviewer_8o1U · 2026-04-05
> >
> > NAN

---

> > > ### Author Response · Authors · 2026-04-08
> > >
> > > Thank you very much for confirming that your concerns have been fully resolved. We sincerely appreciate your valuable suggestions that helped to improve the paper.

---

### Decision · Program_Chairs · 2026-04-30

**Decision:**

Accept (regular)

**Comment:**

The reviewers agreed that this paper studies an important problem in panel Granger-causality testing and proposes a technically solid approach. They found the method well motivated, appreciated the modular power-enhancement formulation and accompanying theory, and viewed the empirical study as strong overall, particularly in sparse and high-dimensional regimes. All four reviewers were supportive in their final assessments.

Overall, I recommend acceptance. During discussion, reviewers raised questions about finite-sample behavior, threshold calibration, the role of the baseline test selection, the breadth of empirical comparisons, and the scope of the assumptions in real-data settings. I have read the paper and the author responses carefully, and I find that these concerns were satisfactorily addressed through clarification and additional experiments. The paper makes a meaningful technical contribution, and the final reviewer assessments support publication. For the final version, I encourage the authors to further clarify finite-sample considerations, the baseline selection strategy, and the practical scope of the modeling assumptions.